# FiGuRO: Intrinsic Dimension Estimation for Multi-Modal Data

Viktoria Schuster [1 2 3]   Sana Tonekaboni [1 2 4]   Caroline Uhler [1 2]

## Abstract

Determining the complexity, or *Intrinsic Dimension (ID)*, of data is fundamental to efficient and interpretable representation learning. This is particularly challenging in multi-modal settings when trying to learn disentangled representations for shared and private information. Existing techniques leave a critical gap: they are often static, uni-modal, or in the case of contrastive methods, adapt only to the shared ID implicitly. We introduce Fidelity-Guided Rank Optimization (FiGuRO), a framework for approximating the ID of uni- and multi-modal data under constraints of model capacity and hyperparameters. FiGuRO learns the dimensions of low-rank projections using truncated singular value decomposition and an algorithm that determines *when* to reduce or increase dimension and in *which* latent space. Disentanglement of shared and private information arises as an emergent property of this optimization, eliminating the need for complex auxiliary loss functions. We demonstrate that FiGuRO outperforms existing ID estimation techniques and is more robust to hyperparameter changes. Across simulations and real-world data, FiGuRO captures distinct ID scales and varying subspace ratios, and decomposes shared and private information successfully. Furthermore, we show that FiGuRO can be applied to modern uni-modal pretrained models, enabling efficient, post-hoc disentanglement of multi-modal representations.

## 1. Introduction

Representation learning aims to find low-dimensional representations able to describe complex, high-dimensional data with minimal information loss (Vincent et al., 2008). It is deeply rooted in the Manifold Hypothesis, which posits that real-world data, despite being embedded in a high-dimensional ambient space, is concentrated near a low-dimensional manifold (Fefferman et al., 2016). Approximating this manifold and thus optimal compression requires good estimates of the data complexity, which is a fundamental challenge in representation learning. A critical component of data complexity is its *Intrinsic Dimension (ID)*, defined as the minimum number of variables needed to describe the data without significant information loss. The ID quantifies the true degrees of freedom and complexity of the underlying manifold, and the performance of deep neural networks has been shown to depend on the intrinsic rather than ambient dimension (Nakada & Imaizumi, 2020). The challenge of ID estimation is amplified in the context of multi-modal data. In this setting, the problem transforms from estimating a single ID to a more complex one of disentangling the latent space into shared and modality-specific or "private" subspaces, each with their own unknown ID. Quantifying these distinct IDs is especially important in fields like biology and medicine, where we i) need interpretable models and ii) want to know whether expensive or difficult-to-obtain modalities are relevant (Zhang et al., 2026; Uhler & Shivashankar, 2022; Tonekaboni et al., 2025; Gliozzo et al., 2025).

Despite its importance, determining the ID of multi-modal data and its subspaces has remained a fundamental challenge. Traditional ID estimation methods struggle with the curse of dimensionality, often underestimating the true ID and failing to scale to complex, high-dimensional data (Campadelli et al., 2015; Binnie et al., 2025). While more advanced neural network approaches exist, they are often static or designed only for uni-modal data (Bahadur & Paffenroth, 2020; Bonheme & Grzes, 2022; Saha et al., 2025; Potapov & Ali, 2002). In multi-modal learning, most methods either treat latent dimensions as fixed hyperparameters requiring extensive tuning (Bousmalis et al., 2016; Gonzalez-Garcia et al., 2018; Lee & Pavlovic, 2021) or, in the case of state-of-the-art contrastive models, adapt only to a shared ID implicitly as an emergent property (Gui et al., 2025). To

---

[1]Laboratory for Information and Decision Systems, Massachusetts Institute of Technology, Cambridge, MA, USA [2]Eric and Wendy Schmidt Center, Broad Institute of MIT and Harvard, Cambridge, MA, USA [3]Technical University of Denmark, Lyngby, Denmark [4]Vector Institute, Toronto, Canada. Correspondence to: Viktoria Schuster <vschu211@mit.edu>, Caroline Uhler <cuhler@mit.edu>.

*Proceedings of the 43rd International Conference on Machine Learning*, Seoul, South Korea. PMLR 306, 2026. Copyright 2026 by the author(s).

the best of our knowledge, while initial theoretical identifiability results have recently been obtained (Sturma et al., 2023), no existing neural network-based approach provides explicit estimates of the distinct IDs of both shared and private subspaces in multi-modal data.

In this work, we introduce a technique for uni- and multi-modal ID estimation called **Fidelity-Guided Rank Optimization (FiGuRO)**. Our approach estimates the effective IDs for each subspace under a number of constraints. We combined two powerful principles to create a method that approximates intrinsic dimensions of decoupled subspaces: Firstly, latent spaces are learned via low-rank decomposable layers using truncated Singular Value Decomposition inspired by adaptive rank reduction (Mounayer et al., 2025) and LoRA (Hu et al., 2021). Secondly, our algorithm to govern the rank optimization is based on Rate-Distortion Theory, using relative reconstruction fidelities of all modalities to decide when to increase or decrease subspace ranks. FiGuRO is the first method to achieve multi-modal ID estimation and information decoupling in a single training pass. We demonstrate FiGuRO's superiority over both uni- and multi-modal methods in ID estimation on various simulated datasets. On multi-modal simulations, we further show that our method picks up on differences in scale and shared-to-private ratios of ground truth IDs. Lastly, we apply FiGuRO to real-world multi-modal datasets and pretrained models, demonstrating the utility of learned latent representations and its efficiency for interpretable multi-modal representation learning. FiGuRO is available on GitHub: https://github.com/viktoriaschuster/FiGuRO.

## 2. Related Work

A common line of inquiry of neural network-based ID estimation has focused on making autoencoders aware of the data's geometric structure. Early approaches were often computationally expensive, such as training multiple models with different bottleneck sizes to identify a sharp increase in reconstruction loss, or "loss cliff" (Bahadur & Paffenroth, 2020). More adaptive solutions have been proposed within the VAE (Kingma & Welling, 2022) framework. These include one-time estimation algorithms like FONDUE, which identifies the optimal fixed dimensionality by detecting the collapse of posteriors for irrelevant latent dimensions (Bonheme & Grzes, 2022), and continuous Bayesian methods like ARD-VAE, which adds a hierarchical prior to automatically prune dimensions during training (Saha et al., 2025). However, these approaches inherit the known optimization challenges of the ELBO (Alemi et al., 2018). Rank reduction autoencoders (RRA) present a deterministic counterpart and learn latent representations as matrix decompositions, allowing for dynamic pruning via truncated singular value decomposition (Mounayer et al., 2025). Matrix decomposi-

tion has also successfully been used for parameter-efficient fine-tuning frameworks like LoRA (Hu et al., 2021) and LoReFT (Wu et al., 2024), with a recent extension of distributing a fixed rank budget based on explained variance of activations (Paischer et al., 2025). Another recent direction in neural ID estimation connects manifold dimension and score-based generative models. Kamkari et al. (2024) introduced FLIPD for efficient local ID estimation. Yeats et al. (2025) took this further and established a formal theoretical link, showing that the denoising score matching loss provides a lower bound for a manifold's ID.

In the multi-modal setting, methods from classical multi-view data decomposition such as JIVE (Lock et al., 2013), AJIVE (Feng et al., 2018), SLIDE (Gaynanova & Li, 2017), ShIndICA (Pandeva & Forré, 2023), DIVAS (Prothero et al., 2024), and PPD (Sergazinov et al., 2025) provide estimates for both joint and individual subspaces, but they inherently rely on the assumption of linear mixing of the underlying variables and lack scalability. To the best of our knowledge, a generalizable neural network-based technique enabling disentanglement *and* explicitly estimating the complete ID structure is still lacking. Several lines of work address the related challenge of disentangling multi-modal information. In causal representation learning, Sturma et al. (2023) identify shared causal variables from unpaired data, but under restrictive assumptions of linearity and non-Gaussianity. Many multi-modal representation learning approaches propose to fuse modalities by learning joint or aligned representations (Wang et al., 2015; Wu & Goodman, 2018; Shi et al., 2019; Sutter et al., 2024). Others have addressed the problem of "modality laziness", which describes a model's tendency to solve a task by relying on the most dominant modality while neglecting the rest (Huang et al., 2021). Proposed solutions include uni-modal pretraining (Ismail et al., 2020), gradient balancing (Peng et al., 2022), and alternate modality training (Zhang et al., 2024). Some advanced methods learn separate representations for shared and modality-specific information, but treat the dimensions as hyperparameters (Bousmalis et al., 2016; Gonzalez-Garcia et al., 2018; Lee & Pavlovic, 2021). Recently, it has been proposed that multi-modal contrastive methods like CLIP (Radford et al., 2021) implicitly adapt to the shared intrinsic dimension of the data as an emergent property of optimizing the contrastive loss (Gui et al., 2025; Wang et al., 2025). To summarize, related work either focuses on alignment, treats dimensions as fixed hyperparameters, or yields an implicit, uninterpretable estimate of only the shared ID. In contrast, our work provides an explicit approach to estimate the IDs of both shared and modality-specific subspaces, as well as a general training framework for multi-modal models to improve disentanglement and thus interpretability.

# 3. Methods

Our method, Fidelity-Guided Rank Optimization (FiGuRO), dynamically estimates the IDs of uni- and multi-modal data. FiGuRO learns low-rank projections for each latent subspace inspired by adaptive rank reduction (ARR) (Mounayer et al., 2025), allowing dimensions to be both reduced and increased in order to converge to the ID. Our proposed algorithm decides whether to decrease or increase ranks guided by principles from Rate-Distortion Theory, using simple reconstruction fidelity metrics.

## 3.1. Preliminaries

**Rate-Distortion Theory and Autoencoders.** Rate-Distortion Theory can be used to describe the trade-off between the complexity of a representation (rate) and its fidelity (distortion). The rate-distortion function, $R(D)$, defines the minimum rate R required to transmit data such that it can be reconstructed with an expected distortion less than or equal to $D$ (Berger, 1975):

$$R(D) = \min_{p(\hat{\boldsymbol{x}}|\boldsymbol{x}) \text{ s.t. } \mathbb{E}[d(\boldsymbol{x},\hat{\boldsymbol{x}})] \leq D} I(\boldsymbol{x};\hat{\boldsymbol{x}}) \qquad (1)$$

with $p(\hat{\boldsymbol{x}}|\boldsymbol{x})$ as the conditional probability of the reconstruction, $\mathbb{E}[d(\boldsymbol{x},\hat{\boldsymbol{x}})]$ as the expected distortion under a chosen metric $d$, and $I(\boldsymbol{x};\hat{\boldsymbol{x}})$ as the mutual information. An autoencoder learns a representation $\boldsymbol{z}$ of a data sample $\boldsymbol{x}$ by training an encoder $f(\boldsymbol{x};\phi)$ with parameters $\phi$ and a decoder $g(\boldsymbol{z};\theta)$ with parameters $\theta$ to reconstruct the input. The reconstruction loss can be seen as a direct measure of distortion. The rate is implicitly controlled by the dimensionality of the bottleneck with dimension $k$. The Minimum Description Length principle formalizes this by showing that the description length $L$ of a latent code $\boldsymbol{z} \in \mathbb{R}^k$ is linearly proportional to its dimension (Grünwald, 2007). The bottleneck dimension $k$ is thus a direct, architectural proxy for the rate.

**Multi-modal autoencoders.** A common approach to improve interpretability in multi-modal representation learning is to split the latent representation into distinct subspaces learning shared and modality-specific information (Tsai et al., 2018; Lee & Pavlovic, 2021). For two modalities $\mathbf{X}_1$ and $\mathbf{X}_2$, latent subspaces are generated either directly or, in deep models, from intermediate representations. We denote these latent subspaces as the shared latent space $\boldsymbol{h}_s$ and private latent spaces $\boldsymbol{h}_1$ and $\boldsymbol{h}_2$. The decoders reconstruct each modality from a combination of its private space and the shared space, where $\oplus$ denotes concatenation.

$$\begin{aligned} \hat{\boldsymbol{x}}_1 &= g_1(\boldsymbol{h}_s \oplus \boldsymbol{h}_1; \theta_1) \\ \text{with } \boldsymbol{h}_1 &= f_1(\boldsymbol{x}_1; \phi_1), \ \boldsymbol{h}_s = f_s(\boldsymbol{x}_1, \boldsymbol{x}_2; \phi_s) \end{aligned} \qquad (2)$$

The formulation for $\hat{\boldsymbol{x}}_2$ is analogous. This architecture encourages $\boldsymbol{h}_s$ to capture modality-invariant information, while $\boldsymbol{h}_1$ and $\boldsymbol{h}_2$ capture modality-specific details.

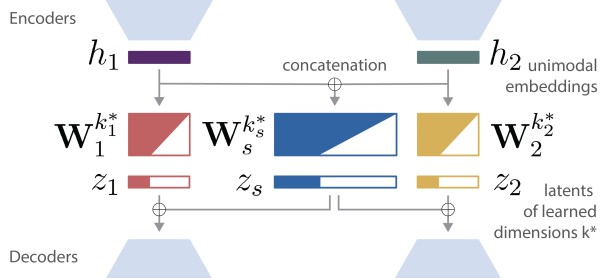

*Figure 1.* **FiGuRO's multi-modal adaptive fusion architecture.** Uni-modal embeddings $h_m$ are decomposed into shared ($z_s$) and private ($z_m$) representations via pruned weight matrices $\mathbf{W}_s^{(k_s^*)}$ and $\mathbf{W}_m^{(k_m^*)}$. When using frozen pretrained models, concatenated representations ($z_s, z_m$) are passed through a fusion layer to reconstruct $h_m$.

**Adaptive rank reduction.** ARR is a technique for dynamically reducing the dimensionality of the latent space in a neural network (Mounayer et al., 2025). Instead of learning a full latent matrix $\mathbf{Z} \in \mathbb{R}^{N \times k}$, one can learn its low-rank decomposition. Building on truncated Singular Value Decomposition (SVD), the unique optimal rank-$k^*$ approximation of $\mathbf{Z}$ is given by $\mathbf{Z} \approx \mathbf{Z}^{(k^*)} = \mathbf{U}^{(k^*)}\mathbf{S}^{(k^*)}\mathbf{V}^{T(k^*)}$, as established by the Eckart-Young-Mirsky Theorem (Eckart & Young, 1936; Mirsky, 1960). While Mounayer et al. (2025) used a unit step of 1 to reduce ranks, a cumulative energy threshold $\gamma$ is a more flexible and scalable choice (Zhang et al., 2023): Dimensions above index $\min(k : \sum_j^k \mathbf{E}_j \geq 1 - \gamma)$ with energy $\mathbf{E}_j = \mathbf{S}_j^2 / \sum_j^k \mathbf{S}^2$ of singular values $S$ are discarded all at once.

## 3.2. Fidelity-Guided Rank Optimization (FiGuRO)

The core of our contribution is an algorithm that optimizes the dimension of latent subspaces (converges toward the ID) under a user-defined "acceptable" level of lossy compression inspired by ARR and Rate-Distortion Theory. Unlike standard ARR, which applies SVD on latent batches, we decompose the weight matrices similar to LoRA (Hu et al., 2021) to learn a global low-rank structure independent of batch size. Furthermore, we explicitly prune the irrelevant dimensions from the projection, giving low-dimensional representations. We also enable the increase of ranks if compression has become too lossy. We introduce a low-rank decomposable layer with maximum rank $k_{max}$ into the bottleneck of an arbitrary, sufficiently capable autoencoder (see theoretical guarantees in A.1.3) with weight matrix $\mathbf{W}$ as follows:

$$\mathbf{W} \approx \mathbf{W}^{(k^*)} = \mathbf{U}^{(k^*)}\mathbf{S}^{(k^*)}\mathbf{V}^{T(k^*)} \qquad (3)$$

with $\mathbf{U}^{(k^*)} \in \mathbb{R}^{l \times k^*}$ and $\mathbf{V}^{T(k^*)} \in \mathbb{R}^{k^* \times l}$ as trainable parameters. $l$ denotes the upstream hidden dimension in the autoencoder. Rank reduction to $k^*$ is governed by the energy threshold $\gamma$ as in ARR. Following Equations 2 and 3,

for a given modality $\boldsymbol{x}_m$, its reconstruction is generated as

$$\hat{\boldsymbol{x}}_m = g_m(\boldsymbol{z}_s \oplus \boldsymbol{z}_m; \theta_m)$$

$$\text{with } \boldsymbol{z}_m = \mathbf{W}_m^{(k_m^*)}\boldsymbol{h}_m, \ \boldsymbol{h}_m = f(\boldsymbol{x}_m; \phi_m) \quad (4)$$

$$\boldsymbol{z}_s = \mathbf{W}_s^{(k_s^*)}(\boldsymbol{z}_{m=1} \oplus \boldsymbol{z}_{m=2})$$

where $\mathbf{W}_1^{(k_1^*)}$ and $\mathbf{W}_s^{(k_s^*)}$ are the dynamically adjusted low-rank weight matrices. Ranks $k_s$ and $k_m$ are determined via Algorithm 1. After an initial training phase, we compute the baseline fidelity/distortion metric $D_0$. The distortion budget $\lambda$ defines how "lossy" we allow the compression to become. We mainly use the coefficient of determination ($R^2 \in [0, 1]$, see Appendix B.4) as our fidelity metric. $R^2$ is scale-invariant, which allows us to define the minimum acceptable fidelity as $D_0 - \lambda$ determining when to stop rank reduction. This makes the use of FiGuRO with different data and loss functions more intuitive.

Every $\tau$ epochs, we compute $D$ and evaluate which ranks should be reduced or increased based on the distortion budget $\lambda$. Reduction is performed via cumulative energy reduction (using energy threshold $\gamma$, See Section 3.1) and dimensions are additionally masked out from the weight matrix to produce low-dimensional representations. Increasing dimensions is achieved by unmasking. An additional hyperparameter of FiGuRO is patience $\pi$. $\pi$ determines after how many steps $\tau$ without rank changes the algorithm stops. FiGuRO also only increases ranks if $D < D_0 - \lambda$ was true for at least $\pi/2$ steps. The full procedure is detailed in Algorithm 1.

Our approach can be framed as a greedy algorithm for finding an efficient operating point on the rate-distortion curve. It works by minimizing the reconstruction loss given model parameters $\phi$, $\theta$, and rank $k^*$ under the condition that the change in distortion $\Delta\mathbb{E}[d(\mathbf{X}, \hat{\mathbf{X}})]$ is below or equal to the distortion budget $\lambda$.

$$R(D) = \min_{\phi, \theta, k^* \text{ s.t. } \Delta\mathbb{E}[d(\mathbf{X}, \hat{\mathbf{X}})] \leq \lambda} \mathcal{L}_{\text{recon}} \quad (5)$$

The essential components of FiGuRO are the explicit shared/subspace separation (in the multi-modal case), low-rank decomposable layers, and the bi-directional rank optimization logic described in Algorithm 1, including the use of a distortion metric $D$ and threshold $\lambda$. Using an interval with step size $\tau$ and patience $\pi$, the choice of distortion metric, and updating ranks based on cumulative energy threshold $\gamma$ are implementation choices.

FiGuRO promotes convergence toward the ID in finite time (A.1.6, A.1.7) under a number of assumptions. As suggested in Equation 5, our estimation is dependent on models $f_\phi$ and $g_\theta$, and relies on the assumption that they act as sufficient function approximators to model the data manifold (A.1.3). We further require the chosen distortion metric to be a suitable and stable proxy for reconstruction fidelity (A.1.4).

---

**Algorithm 1** Fidelity-Guided Rank Optimization

1: **Input:** Multi-modal data $\{\mathbf{X}_1, \ldots, \mathbf{X}_M\}$, distortion budget $\lambda$, reduction frequency $\tau$, patience $\pi$, energy threshold $\gamma = 0.01$.
2: **Initialize:** Multi-modal autoencoder with full-rank adaptable layers: $k_{\min} \leftarrow 1$, $k_{\max} \leftarrow l$, $k_0 = k_{\max}$
3: **Phase 1: Pre-training**
4: Train the full-rank model to loss convergence
5: Compute initial fidelity: $D_{0,m} \leftarrow D(\mathbf{X}_m, \hat{\mathbf{X}}_m) \ \forall \ m$
6: **Phase 2: Rank Optimization**
7: **for** each epoch $t$ **do**
8:     Train model on reconstruction loss for one epoch.
9:     **if** $t \bmod \tau = 0$ **then**
10:         Identify ranks for adjustment
11:         $K_{decrease} \leftarrow \{k_{t,m} \mid D_{t,m} > (D_{0,m} - \lambda)\}$
12:         $K_{increase} \leftarrow \{k_{t,m} \mid D_{t,m} \leq (D_{0,m} - \lambda)\}$
13:         Assign shared rank $k_s$ if all modalities agree
14:         **if** $K_{increase} = \emptyset$ **then**
15:             $K_{decrease} \leftarrow K_{decrease} \cup \{k_s\}$
16:         **else if** $K_{decrease} = \emptyset$ **then**
17:             $K_{increase} \leftarrow K_{increase} \cup \{k_s\}$
18:         **end if**
19:         Update ranks
20:         **for** $k_{t,i}$ in $K_{decrease}$ **do**
21:             $k_{t+1,i} \leftarrow \min(k : \sum_j^k \mathbf{E}_j \geq 1 - \gamma)$
22:         **end for**
23:         **for** $k_t$ in $K_{increase}$ **do**
24:             $k_{t+1,i} \leftarrow \max(k_t + 1, int(1.1k_t))$
25:         **end for**
26:         **if** ranks have not changed for $\tau \times \pi$ epochs **then**
27:             **break**
28:         **end if**
29:     **end if**
30: **end for**
31: **Return:** Final ranks $\{k_s^*, k_1^*, \ldots, k_M^*\}$ and trained model.

---

Our theoretical guarantees are elaborated in Appendix A.1. Appendix A.2 discusses how information disentanglement emerges as the most rank-efficient solution in multi-modal settings.

## 4. Experimental Setup

### 4.1. Datasets

**Simulated Data.** Evaluation of ID estimation techniques requires data with known ground truth dimensions. For this purpose, we generated simulated datasets of varying complexity in terms of generative process, IDs, and ambient dimensionality. A summary is given in Table 1. Datasets **A** and **B** present simple simulations which we refer to as "parametric" as they can be generated for a variety of data

*Table 1.* **Simulated datasets.** A summary of the datasets we simulated, including the ambient data dimensions, ground truth hidden dimensions for shared and private subspaces (IDs), and the number of samples.

| Dataset | Dimension | IDs | Samples |
|---------|-----------|-----|---------|
| | | $z_s, z_1, z_2$ | |
| **A** | 50 | 5 | 10K |
| $\mathbf{B}_s$ ("small") | 200, 200 | 2, 3, 5 | 10K |
| $\mathbf{B}_{i1}$ ("imbalanced 1") | 200, 200 | 20, 2, 2 | 10K |
| $\mathbf{B}_{i2}$ ("imbalanced 2") | 200, 200 | 2, 2, 20 | 10K |
| $\mathbf{B}_l$ ("large") | 200, 200 | 20, 20, 20 | 10K |
| **C** | 20K, 20K | 6, 5, 6 | 30K |

characteristics. Uni-modal dataset **A** was designed to evaluate hyperparameter sensitivity and robustness to data characteristics. Multi-modal simulation **B** was created with four variations that differ in the ratios and scales of the shared and private hidden dimensions in order to test whether our method can pick up on different subspace IDs. Dataset **C** is a more complex and higher-dimensional simulated dataset inspired by biological measurements of gene expression and protein abundance. It allows us to validate our method in a setting closer to real data. Details on the simulation processes are provided in Appendix B.1. Since our method and some baselines require sufficiently capable autoencoders, we performed an architecture and hyperparameter search for the most complex dataset, **C**. Full details on the objectives, search space, training, and final selected parameters are provided in Appendix B.3.1.

**Real-world Data.** To demonstrate applicability to real-world data and performance on downstream tasks, we used three multi-modal datasets: Audio MNIST (Becker et al., 2023), So2Sat (Zhu et al., 2020), and NYU Depth V2 (Nathan Silberman & Fergus, 2012). Audio MNIST comprises images of handwritten digits and audio recordings of these digits. So2Sat contains synthetic aperture radar (SAR) and optical image data used to classify climate zones. NYU Depth V2 consists of paired RGB and depth images captured from indoor scenes. See the Appendix for details on data preprocessing (B.2).

## 4.2. Robustness

**Hyperparameters and Data Characteristics.** To evaluate the stability of our estimator, we conducted a comprehensive grid search on dataset **A** ($N = 1080$ runs) varying the distortion budget $\lambda$, rank reduction frequency $\tau$, energy threshold $\gamma$, and patience $\pi$ (Appendix B.5). We further tested robustness to data characteristics by systematically varying sample sizes, latent distribution types, nonlinearity functions and depths, the sparsity of the transformation matrix (connectivity), sparsity, and noise (Appendix B.7.1). Finally, we compared different distortion metrics ($R^2$, Ex-

plained Variance Score, MSE, and RMSE) under varying levels of nonlinearity and sparsity (Appendix B.7.3).

**Ablation and Additive Studies.** To verify the necessity of our fidelity-guided optimization, we compared FiGuRO against a naive "rank-reduction-only" baseline. This variant performs singular value pruning based on the energy threshold $\gamma$ but lacks our bidirectional rank adjustment and distortion constraints. Furthermore, we tested whether auxiliary loss terms (Frobenius norm, L1, L2) are needed for disentanglement. All experiments were performed on the subsets of **B**.

## 4.3. Benchmarking

**Uni-modal Estimation.** We benchmarked ID estimation against a wide range of traditional and neural network-based estimators using both modalities of **C**, which have the same high ambient dimension and similar ground truth IDs. However, $\mathbf{C_1}$ is corrupted by high noise, while $\mathbf{C_2}$ features high nonlinearity, representing common failure modes for classic estimators. For estimators with tunable parameters, we report the full range of results to demonstrate sensitivity. Baseline methods and their implementations and parameterization are detailed in Appendix B.8.

**Multi-modal Estimation and Decomposition.** To evaluate subspace ID estimation and disentanglement, we compared FiGuRO against multi-view decomposition methods JIVE (Lock et al., 2013), AJIVE (Feng et al., 2018), SLIDE (Gaynanova & Li, 2017), and ShIndICA (Pandeva & Forré, 2023) using all four subsets of **B**, which have variable ID scales and ratios for the different subspaces. We implemented these methods with default settings except for sparsity penalty in SLIDE and the rank grid in ShIndICA (tuning required for convergence). Details can be found in Appendix B.8.

## 4.4. Application to Real Data

We applied FiGuRO to three multi-modal datasets described above. For Audio MNIST and So2Sat, we pre-trained custom uni-modal autoencoders to establish baseline representations. For the high-resolution NYU Depth V2 dataset, we evaluated FiGuRO's ability to probe the latent spaces of modern foundation models. We used the frozen variational autoencoder from Stable Diffusion (Rombach et al., 2021) and the Depth Anything V2 estimator (Yang et al., 2024) as feature extractor (plus a custom decoder). We report on the domain-specific meaningfulness of the learned subspaces as well as the effect of dimensionality reduction on reconstruction performance. As an example of downstream utility of the learned subspaces, we evaluated classification performance on Audio MNIST (digit recognition) and So2Sat (climate zone classification). Architecture and training details are provided in Appendices B.3 and B.6.

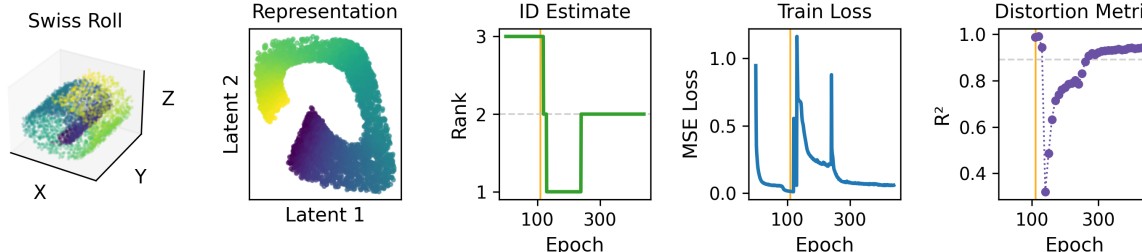

*Figure 2.* **FiGuRO's ID estimation over time on the 3D Swiss Roll Manifold.** From left to right: 3D data space, learned latent representation with 2 dimensions (seed 19), rank over epochs as the ID estimate (with a dashed line on true ID 2), training mean squared error (MSE) loss over epochs, and distortion metric ($R^2$) over epochs with the dashed line depicting the minimum acceptable $R^2 = R_0^2 - \lambda$. The orange vertical lines show the end of the warmup period after which rank optimization started.

## 5. Results

We validate FiGuRO's ability to estimate IDs and disentangle subspaces, progressing from controlled simulations to real-world applications with large pretrained models. Figure 2 illustrates an example of the training dynamics of FiGuRO on the Swiss Roll dataset, tracking the convergence of the rank estimate to the true ID alongside the reconstruction loss, distortion metric, and the resulting low-dimensional latent topology.

### 5.1. Robustness to Hyperparameters and Data Characteristics

Across our comprehensive hyperparameter sweep, FiGuRO arrived at a robust average estimate of $4.89 \pm 0.01$ (SEM) for a true ID of 5. Results in Supplementary Table 1 indicate a mild dependence on the distortion budget $\lambda$ and negligible sensitivity to $\gamma$, validating our fidelity-guided stopping criterion. While the method proved robust to diverse data characteristics, we observed that high levels of dropout or noise (signal-to-noise ratio SNR < 7) could lead to over-

estimation (Supplementary Figure 1). Experiments on 3D manifolds (Supplementary Figure 3) further confirmed the necessity of our bidirectional update mechanism, as ranks were frequently observed dropping too low before being corrected upwards to the true ID. Our hyperparameter sweep allowed us to select a single configuration balancing performance and speed $\{\lambda = 0.05, \tau = 10, \gamma = 0.01, \pi = 10\}$ for subsequent experiments unless stated otherwise. Investigating the robustness of different distortion metrics to nonlinearities and sparsity in the data revealed that $R^2$ was more robust to sparsity than MSE and RMSE, and more robust to nonlinearities than the Explained Variance Score.

### 5.2. ID Estimation Benchmarks

Existing uni-modal ID estimators displayed distinct failure modes for different data characteristics (Supplementary Table 2). Global linear methods like PCA severely overestimated the dimension of the noisy data $\mathbf{C_1}$, likely because they measure the linear embedding dimension rather than the manifold dimension. Most local estimators (e.g. MLE

*Table 2.* **Comparison of estimated ranks for joint and individual components across multi-modal decomposition methods for datasets of B.** Columns 4-7 present baseline methods from multi-view data decomposition. Best estimates (closest to ground truth) are highlighted in bold, second best underlined. We report average deviation from ground truth at the end including the standard error of the mean SEM. For our method, we used a distortion budget of $\lambda = 0.05$.

| Dataset | Ground Truth | Subspace | JIVE | AJIVE | SLIDE | ShIndICA | FiGuRO (ours) |
|---------|-------------|----------|------|-------|-------|----------|---------------|
| $\mathbf{B}_s$ | 2 | Shared | **1** | **3** | **1** | **1** | $\underline{3.6 \pm 0.6}$ |
| | 3 | Modality 1 | $\underline{11}$ | 197 | 10 | 10 | $\mathbf{1.0 \pm 0.0}$ |
| | 5 | Modality 2 | 9 | **6** | $\underline{8}$ | $\underline{8}$ | $\mathbf{6.2 \pm 1.8}$ |
| $\mathbf{B}_{i1}$ | 20 | Shared | 1 | **21** | 1 | 1 | $\underline{13.4 \pm 0.4}$ |
| | 2 | Modality 1 | 23 | **2** | 22 | 22 | $\underline{4.4 \pm 0.2}$ |
| | 2 | Modality 2 | 23 | **3** | 22 | 22 | $\underline{4.0 \pm 0.5}$ |
| $\mathbf{B}_{i2}$ | 2 | Shared | **1** | **3** | **1** | **1** | $\underline{7.0 \pm 0.0}$ |
| | 2 | Modality 1 | 12 | 196 | $\underline{11}$ | $\underline{11}$ | $\mathbf{1.0 \pm 0.0}$ |
| | 20 | Modality 2 | 23 | **20** | 22 | 22 | $\underline{19.4 \pm 1.5}$ |
| $\mathbf{B}_l$ | 20 | Shared | 1 | $\underline{21}$ | 1 | 10 | $\mathbf{19.2 \pm 0.7}$ |
| | 20 | Modality 1 | 41 | 40 | 40 | $\underline{31}$ | $\mathbf{12.8 \pm 1.0}$ |
| | 20 | Modality 2 | 41 | 42 | 40 | $\underline{31}$ | $\mathbf{15.2 \pm 1.3}$ |
| Avg deviation from ground truth | | | 12.4 | 36.3 | 11.8 | $\underline{9.5}$ | **2.9** |
| SEM | | | 2.5 | 21.7 | 2.4 | $\underline{2.1}$ | **0.7** |

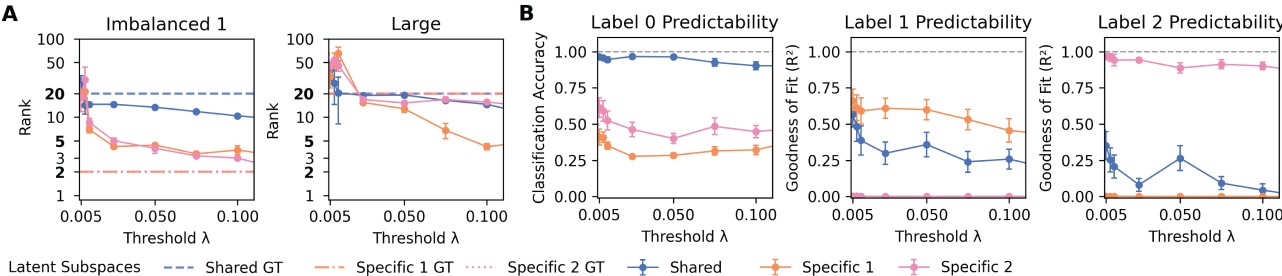

*Figure 3.* **Multi-modal ID estimation and disentanglement of B with varying true IDs.** The x axis presents the distortion threshold $\lambda$. **(A)** Log-scale estimated ranks (mean $\pm SEM$, $N = 5$ seeds) for shared (blue) and modality-specific (orange, pink) subspaces of $\mathbf{B}_{i1}$ and $\mathbf{B}_l$. Ground truth (GT) IDs are depicted as dashed lines. Values closer to the GT lines of the same color indicate better performance. Results for the remaining subsets of $\mathbf{B}$ are shown in Supplementary Figure 4. **(B)** Average predictability of shared and private information over all random seeds and datasets ($N = 20$). The class of the shared space (label 0) is evaluated as classification accuracy. Labels 1 and 2 represent the mean value of modality-specific generative hidden vectors. Their predictability is evaluated with $R^2$. Metrics are defined in B.4.

(Levina & Bickel, 2004), TwoNN (Facco et al., 2017)) systematically underestimated the ID of the non-linear $\mathbf{C_2}$. Other neural network approaches, such as Rank Reduction Autoencoders (Mounayer et al., 2025), proved unstable, with estimates varying strongly depending on the choice of the rank reduction threshold $\gamma$. Bayesian alternatives like ARD-VAE (Saha et al., 2025) offered a better estimation range than RRA but still performed significantly worse than our method, frequently suffering from posterior collapse. In contrast, FiGuRO was the only framework to provide consistent, close-to-accurate estimates across both regimes.

In the multi-modal case, most baselines showed a high average deviation from the ground truth (GT) over the four subsets of $\mathbf{B}$ (Table 2). FiGuRO, on the other hand, demonstrated a strong ability to recover the underlying dimensional structure of multi-modal data. Table 2 shows that the estimated ranks for the shared and specific subspaces largely preserve the scales and relationships between the ground truth dimensions, albeit sometimes underestimating the true ID ($\mathbf{B}_{i1}$ shared and $\mathbf{B}_l$ private subspace 1). The $\mathbf{B}_{i1}$ shared generative matrix exhibited an effective rank of 6.61 in contrast to its 20 variables (Supplementary Table 3), and we

also see some information leakage into the private subspaces for this case (Supplementary Figure 6). In Figure 3A, we see that these estimates are robust over a large range of the main hyperparameter $\lambda$. Our ablation study (Table 3) showed that a naive *rank-reduction-only* approach universally collapsed to the minimum ranks on all multi-modal datasets $\mathbf{B}$, demonstrating the relevance of our proposed algorithm.

### 5.3. Disentanglement

In many multi-modal settings, we want to separate shared and modality-specific information for interpretability. FiGuRO is designed to learn these decomposed subspaces along with their dimensionalities. We evaluate this emergent disentanglement through measuring the predictability of labels derived from the generative variables of datasets $\mathbf{B}$. Figure 3B shows that predictability per label was highest in its corresponding subspace, demonstrating that disentanglement is an emergent property of FiGuRO. Adding explicit regularizers, such as Frobenius norm for orthogonality or L1 penalties, provided no improvement in disentanglement and often degraded rank estimation accuracy (Supplemen-

*Table 3.* **Ablation study.** This table compares the average estimated ranks $(\mathbf{k_s}, \mathbf{k_1}, \mathbf{k_2})$ of a naive SVD-only rank reduction model against the full FiGuRO implementation (SVD + R(D) algorithm) and ground truth (GT) across datasets $\mathbf{B}$. We report mean $\pm$ SEM on three random seeds.

| Method | Subspace | $\mathbf{B_s}$ | $\mathbf{B_{i1}}$ | $\mathbf{B_{i2}}$ | $\mathbf{B_l}$ |
|---|---|---|---|---|---|
| GT | $k_s$ | 2 | 20 | 2 | 20 |
|  | $k_1$ | 3 | 2 | 2 | 20 |
|  | $k_2$ | 5 | 2 | 20 | 20 |
| FiGuRO | $k_s$ | $1.0 \pm 0.0$ | $1.0 \pm 0.0$ | $1.0 \pm 0.0$ | $1.0 \pm 0.0$ |
| (SVD) | $k_1$ | $1.0 \pm 0.0$ | $1.0 \pm 0.0$ | $1.0 \pm 0.0$ | $1.0 \pm 0.0$ |
|  | $k_2$ | $1.0 \pm 0.0$ | $1.0 \pm 0.0$ | $1.0 \pm 0.0$ | $1.0 \pm 0.0$ |
| FiGuRO | $k_s$ | $3.6 \pm 0.06$ | $13.4 \pm 0.4$ | $7.0 \pm 0.0$ | $19.2 \pm 0.7$ |
| (SVD + | $k_1$ | $1.0 \pm 0.0$ | $4.4 \pm 0.2$ | $1.0 \pm 0.0$ | $12.8 \pm 1.0$ |
| R(D)) | $k_2$ | $6.2 \pm 1.8$ | $4.0 \pm 0.5$ | $19.4 \pm 1.5$ | $15.2 \pm 1.3$ |

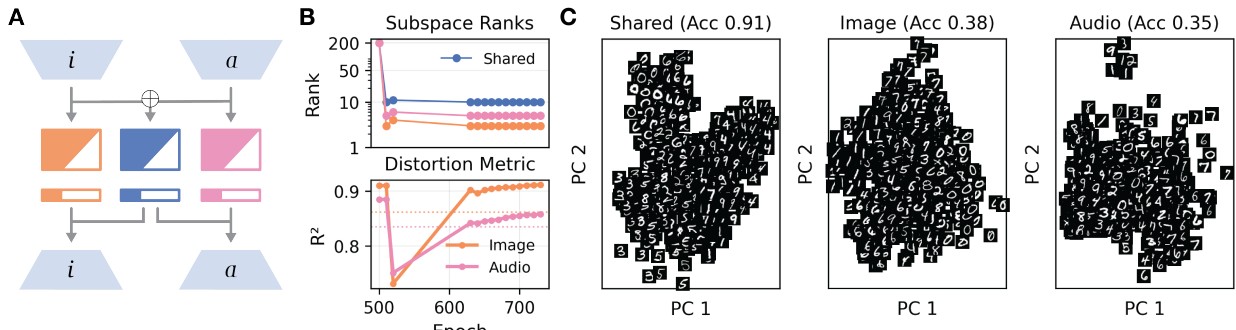

*Figure 4.* **Disentanglement of semantic content from modality-specific style on Audio MNIST. (A)** FiGuRO's multi-modal adaptive fusion architecture. Uni-modal embeddings are decomposed into shared and private representations of learned dimensionality. Colors of matrices and decomposed representations $z$ match the legend in B. **(B)** Rank estimation and distortion metric ($R^2$) over epochs. Colors indicate the subspaces in the rank plot and modalities in the distortion plot. Dashed lines indicate the distortion thresholds per modality. **(C)** Latent representations per subspace. We show the first two components of PCAs, with MNIST images instead of dots. Titles indicate the subspace and show classification accuracy of digits for seed 0.

tary Tables 4-7). While L2 regularization assisted in the edge case of fully independent modalities (Supplementary Table 8), the standard FiGuRO objective sufficed for disentanglement in general multi-modal settings. Crucially, baseline decomposition methods frequently failed to correctly assign information, often 'swapping' shared and private signals (see Supplementary Figures 5-8).

### 5.4. Application to Real-World Data

We validated FiGuRO's capacity to handle complex real-world data by analyzing its compression efficiency, ID estimation, and downstream utility. The datasets we used were Audio MNIST, So2Sat, and NYU Depth V2.

**Scalability and Compression.** FiGuRO reduced the total combined ranks by factors of 5-36, retaining high performance with test losses increasing only marginally (Supplementary Table 13). While FiGuRO is more computationally expensive than multi-modal baselines on the small simulations **B**, Supplementary Table 9 demonstrates how the relationship starts to shift it on real-world, larger data. Unlike the baselines that scale quadratically with $\max(N, d)$ ($N$ being the sample size and $d$ the ambient dimension), FiGuRO scales linearly with the sample size and dimension (in the case of an MLP as the base model). By operating on the fixed embeddings of the NYU Depth V2 dataset extracted from Latent Diffusion (Rombach et al., 2021) and Depth Anything V2 (Yang et al., 2024), rather than training end-to-end, we demonstrate FiGuRO's potential as a scalable latent probe and interpretable fusion method for large pretrained, uni-modal models.

**ID Estimation.** Differences in modality-specific IDs aligned with domain expectations, for example allocating more dimensions to image modalities than radar or depth (Supplementary Table 14). For MNIST in particular, we

have existing estimates from traditional uni-modal methods, assuming the ID to be around 7-25 (Pope et al., 2020). Supplementary Table 14 shows that multi-modal baselines were far off from this expectation using Audio-MNIST. FiGuRO, however, converged to an average rank ($k_s + k_{image}$) of $11.4 \pm 0.6$, fitting right into the previously reported range. A small sensitivity analysis to hyperparameters (Supplementary Table 15) revealed that FiGuRO was most sensitive to $\lambda$, the choice of distortion metric, and $\gamma$ (in this order). The sensitivity to $\gamma$ stemmed from an overestimation of ranks due to small $\gamma$ values not allowing for enough rank reduction given $\lambda$. As a result, we believe using $\gamma \geq 0.01$ is most robust. Average total ID estimates were lowest for Explained Variance Score and highest for MSE. The latter matches our previous robustness results on simulations.

**Downstream Utility and Disentanglement.** Beyond structural analysis, the decomposed subspaces proved highly effective for downstream tasks. We evaluated classification accuracy for labels expected to be shared across modalities (Audio MNIST: digit identity, So2Sat: climate zone). FiGuRO successfully isolated these signals in the shared subspace, even achieving higher accuracies than in uni-modal embeddings (Table 4). It also outperformed most multi-modal decomposition baselines, which again tended to "swap" shared and private information. Our results confirm that FiGuRO's emergent disentanglement preserves semantic utility while eliminating redundancy.

## 6. Conclusion

In this work, we address the fundamental challenge of estimating the intrinsic dimensions (IDs) of multi-modal data. We introduce **Fidelity-guided Rank Optimization (FiGuRO)**, a framework that explicitly and dynamically learns the dimensionalities of shared and private subspaces.

*Table 4*. **Multi-modal downstream task performance evaluation (classification).** We report classification accuracies (Appendix B.4.2) of the main prediction task per subspace., $Acc_s$ refers to the accuracy on the shared subspace for balanced datasets, $Acc_{all}$ refers to the accuracy on all concatenated subspaces. For the class-imbalanced So2Sat, we report balanced Accuracy. Arrows indicate whether a higher or lower accuracy is better, based on the assumption that the predicted classes are expected to be part of the shared information. Bold numbers highlight the best performance of the shared representations, underlined second best. $i$: image modality, $a$: audio, $r$: radar. For any predictions from models we trained, we report the mean $\pm$ SEM. "n.a." indicates the computation failed. "−" indicates the the subspace does not exist. For our method, we used a distortion budget of $\lambda = 0.05$.

| Dataset | Metric | Unimodal | JIVE | AJIVE | SLIDE | ShIndICA | **Ours** |
|---|---|---|---|---|---|---|---|
| Audio | $Acc_s \uparrow$ | – | 0.42 | **0.99** | 0.28 | 0.23 | $\underline{0.94 \pm 0.01}$ |
| MNIST $(i, a)$ | $Acc_1 \downarrow$ | $0.66 \pm 0.04$ | 0.63 | **0.41** | 0.77 | $\underline{0.46}$ | $0.49 \pm 0.06$ |
| | $Acc_2 \downarrow$ | $0.65 \pm 0.03$ | 0.81 | $\underline{0.65}$ | 0.72 | 0.71 | **0.44 ± 0.02** |
| | $Acc_{all} \uparrow$ | $0.80 \pm 0.04$ | 0.64 | **0.99** | 0.45 | 0.36 | $\underline{0.97 \pm 0.01}$ |
| So2Sat $(r, i)$ | $Acc_s \uparrow$ | – | 0.15 | 0.20 | $\underline{0.24}$ | n.a. | **0.55 ± 0.03** |
| | $Acc_1 \downarrow$ | $0.34 \pm 0.01$ | $\underline{0.18}$ | **0.15** | 0.21 | 0.25 | $0.25 \pm 0.02$ |
| | $Acc_2 \downarrow$ | $0.38 \pm 0.01$ | 0.35 | $\underline{0.29}$ | **0.26** | n.a. | $0.44 \pm 0.01$ |
| | $Acc_{all} \uparrow$ | $\underline{0.41 \pm 0.01}$ | 0.26 | 0.24 | 0.27 | n.a. | **0.55 ± 0.02** |

FiGuRO successfully recovered subspace ratios and isolated shared signals where classical decomposition methods failed. Beyond standard estimation, we demonstrate that the disentanglement of shared and modality-specific information arises as an emergent property of our multi-modal adapter architecture and rank optimization algorithm. It is not guaranteed, as we saw in some failure modes in which dimensions were underestimated likely due to generative variable correlation, information leakage, or a conservative distortion metric. However, a high degree of disentanglement occurs without the need for auxiliary orthogonality or sparsity constraints.

Our findings are further supported by experiments on real-world datasets with diverse modalities, including images, radar, depth, and audio. We demonstrated that FiGuRO can learn low-dimensional decompositions successfully separating shared and specific information without significant performance loss. Our results on the NYU Depth V2 dataset highlight FiGuRO's potential as a scalable tool for modern representation learning. We show that FiGuRO can serve as a lightweight latent probe applied to the frozen latent spaces of pre-trained models. This decouples the decomposition mechanism from the feature extraction backbone, allowing researchers to leverage state-of-the-art uni-modal encoders off-the-shelf to disentangle multi-modal data without the prohibitive cost of end-to-end retraining. We acknowledge that FiGuRO, like all neural network approaches, relies on an underlying architecture sufficiently expressive to model the data manifold and that ID estimates are dependent on the model's parameters and nonlinearity. However, unlike prior neural estimators that proved unstable, extensive hyperparameter sweeps confirm FiGuRO's robustness across diverse data characteristics and distortion metrics. For robust results, we recommend using FiGuRO to estimate bounds under low and high distortion budgets.

FiGuRO provides a critical missing piece to multi-modal

learning: a principled method to quantify informational complexity. Low-rank bottlenecks provide meaningful regularization and may prove useful for generalization and improved interpretability, which are crucial for machine learning in biology and medicine. Looking forward, the dynamic nature of our framework also makes it uniquely suited for continual learning scenarios, where models must adapt their latent capacity online in response to heterogeneous data sources.

## Acknowledgments

V.S. and S.T. were supported by the Eric and Wendy Schmidt Center at the Broad Institute. V.S. was supported by the Novo Nordisk Foundation grant NNF24OC0089461. C.U. was partially supported by NCCIH/NIH (1DP2AT012345), ONR (N00014-24-1-2687), the United States Department of Energy (DE-SC0023187), and the Eric and Wendy Schmidt Center at the Broad Institute.

## Impact Statement

This work advances interpretable, multi-modal representation learning by quantifying data complexity, enabling efficient modeling and lightweight probing of foundation models. While separating shared signals helps mitigate sensitive attributes in high-stakes domains, this capability carries dual-use risks for malicious profiling. Furthermore, imperfect disentanglement can foster a false sense of fairness. We therefore advocate that these tools be used strictly within a broader, human-in-the-loop auditing process.

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

# A. Theoretical Guarantees

## A.1. ID estimation

Our empirical results are supported by a theoretical framework grounded in the Manifold Hypothesis and Rate-Distortion Theory. We formalize this by first stating our core assumptions, followed by theorems that provide guarantees for the convergence, correctness, and sample complexity of FiGuRO. Our theoretical results hold under the following assumptions:

**Assumption A.1.1** (Data Generation). Let the "clean" data $\mathbf{Y}$ be sampled from a distribution $p(x)$ with support on an $r$-dimensional compact manifold $\mathcal{M} \subset \mathbb{R}^n$. Let the observed data be corrupted by i.i.d. additive noise

$$X = Y + \epsilon$$

where the noise $\epsilon$ is drawn from a distribution with zero mean ($\mathbb{E}[\epsilon] = 0$) and a covariance matrix $\Sigma_\epsilon = \sigma^2 I_D$, with $\sigma^2$ representing the noise variance. The variance of the noise is smaller than the geometric variance of the Manifold.

**Assumption A.1.2.** For any multi-modal data, we further assume that the number of generative variables is greater than $0$, such that the modalities are not fully independent.

**Assumption A.1.3** (Sufficient Capacity). Given Assumption A.1.1, we assume our autoencoder architecture, defined by the encoder family $\mathcal{F}$ (parameterized by $\phi$) and decoder family $\mathcal{G}$ (parameterized by $\theta$), constitutes a **sufficient function approximator**. This holds if, for any arbitrarily small precision $\epsilon > 0$, there exists a set of parameters $(\phi, \theta)$ and a latent dimension $l \geq r$ such that the expected reconstruction distortion $\mathbb{E}[d(X, \hat{X})]$ can be made arbitrarily close to the irreducible noise floor:

$$\mathbb{E}_{p(x)} [d(x, g_\theta(f_\phi(x)))] \leq \sigma^2 + \epsilon$$

**Assumption A.1.4** (Fidelity Proxy). We assume the coefficient of determination, $R^2$, or any other metric chosen, is a suitable and stable proxy for reconstruction fidelity. Specifically, for a given distortion budget $\lambda > 0$, the condition $R^2(X, \hat{X}) \geq R_0^2 - \lambda$ implies that the true reconstruction distortion $\mathbb{E}[d(X, \hat{X})]$ on the manifold $\mathcal{M}$ is bounded.

**Assumption A.1.5** (Compactness). We assume that the $r$-dimensional manifold $\mathcal{M}$ is compact and its generative variables are $i.i.d.$. Compactness implies that there is no sparsity among the generative variables of the data.

**Theorem A.1.6.** *Under Assumptions A.1.3 and A.1.4, the rank optimization phase of the FiGuRO algorithm is **guaranteed to converge** in a finite number of epochs, returning a final set of ranks $\{k_s^*, k_1^*, ..., k_m^*\}$.*

*Proof.* Together with the discrete rank updates and a tightening bound $[k_{min}, k_{max}]$, continuous weight optimization ensures that reconstruction fidelity for any fixed rank is non-decreasing over training time. This contractive dynamic ensures the model cannot cycle indefinitely between failure states and sets for the minimum stable rank. The patience parameter $\pi$ ensures that if the ranks remain unchanged for $\pi$ consecutive checks, the optimization process terminates. As a result, ranks cannot oscillate indefinitely and the algorithm is guaranteed to halt. $\square$

**Theorem A.1.7.** *Let the data be generated from a distribution supported on a dense $r$-dimensional manifold $\mathcal{M}$ (Assumption A.1.5). Under Assumptions A.1.3 and A.1.4, and given a sufficient number of samples $N$ (as defined in Corollary A.1.8), there exists a sufficiently small distortion budget $\lambda$ such that the rank $k^*$ returned by FiGuRO satisfies $\mathbb{E}[k^*] \approx r$.*

*Proof.* The proof relies on the connection to Rate-Distortion theory. If the estimated rank $k^*$ is less than the true ID $r$, the autoencoder is forced to discard information essential for describing the manifold, causing the reconstruction distortion to violate the budget $\lambda$. Conversely, if $k^* > r$, the model is still over-parameterized. The rank reduction mechanism can then prune redundant dimensions without violating the fidelity budget. The algorithm thus converges to a rank that balances model capacity (rate) with reconstruction quality (distortion), which corresponds to the intrinsic dimension of the manifold. This aligns with recent work showing that globally optimized autoencoders can provably learn the correct manifold dimension (Zheng et al., 2023). $\square$

**Corollary A.1.8.** For the guarantee in Theorem A.1.7 to hold, the number of training samples $N$ must satisfy the sample-to-capacity ratio:

$$N \geq \frac{\alpha \cdot \mathcal{C}_e}{k^*} \text{ with } \alpha \geq 10$$

where $k^*$ is the estimated ID, $\mathcal{C}_e$ is the number of parameters in the encoder, and $\alpha$ is a constant factor termed the load (Schuster & Krogh, 2021).

*Proof.* If $N$ is insufficient, the autoencoder will overfit to the training data. This leads to an unreliable fidelity signal ($R^2$), as the model's performance on the validation set (which guides the fidelity check) will diverge from its performance on the training set. A stable and generalizable fidelity signal is necessary for the rank optimization mechanism to correctly probe the data's true dimensionality. $\square$

**Theorem A.1.9.** *Let $\mathcal{M}$ be a $d$-dimensional compact Manifold embedded in an ambient space $\mathbb{R}^n$. Under Assumption A.1.1 the top $k^*$ ranks represent the geometric variance of the Manifold and discard noise.*

*Proof.* The principal components of the data are a combination of geometric and noise variance. As long as the noise level $sigma$ is not excessively high relative to the manifold's curvature, the singular values associated with the manifold's structure are expected to be significantly larger than those associated purely with noise. Then, by penalizing reconstruction error via the distortion metric, the network is incentivized to first capture the geometric variance. The rank reduction process then acts as a form of denoising, progressively discarding the smaller singular values that correspond to the noise subspace. $\square$

## A.2. Disentanglement

While our primary objective $\mathcal{L}_{\text{recon}}$ (Eq. 5 in main paper) does not include an explicit term for minimizing mutual information between subspaces, we posit that disentanglement emerges as an optimal solution from the interplay between our model architecture (Eq. 4 in main paper) and the rate minimization objective of FiGuRO (Algorithm 1). We formalize this using the principles of Rate-Distortion Theory and Information Bottlenecks (IB).

**Assumption A.2.1.** We assume that the rank $k^*$ of a latent subspace, as estimated by FiGuRO, serves as a proxy for its information content, or *Rate $R$*. This is an extension of the Minimum Description Length (MDL) principle, where the rank $k^*$ is the minimal number of parameters (degrees of freedom) required to describe the data, such that $k^* \propto R \approx I(Z; X)$. Given this, the FiGuRO algorithm can be interpreted as solving the following optimization problem for latent representations $H_s, H_1, H_2$ derived from inputs $X_1, X_2$:

1. **Minimize Total Rate (Rank):** $\min_{k_s, k_1, k_2} R_{\text{total}} \approx k_s + k_1 + k_2$

2. **Subject to Distortion Constraint (Fidelity):** The reconstruction error for each modality must remain bounded by the distortion budget $\lambda$, which implies sufficient information preservation:

$$I(H_s, H_1; X_1) \geq H(X_1) - \mathcal{D}_1$$
$$I(H_s, H_2; X_2) \geq H(X_2) - \mathcal{D}_2$$

where $H(X)$ is the entropy (total information) of a modality and $\mathcal{D}$ is the information loss permitted by $\lambda$.

We further define the "true" information components of the data (assuming no synergistic information):

- **Shared Information:** $S = I(X_1; X_2)$

- **Private Information (Modality 1):** $P_1 = H(X_1|X_2) = H(X_1) - S$

- **Private Information (Modality 2):** $P_2 = H(X_2|X_1) = H(X_2) - S$

The total information required to reconstruct both modalities is $H(X_1, X_2) = S + P_1 + P_2$.

**Theorem A.2.2** (Optimality of the Disentangled Solution). *Under the FiGuRO optimization objective (minimizing total rank s.t. reconstruction fidelity) and given the architectural constraints (Eq. 4), the unique optimal (lowest-rank) solution that satisfies the reconstruction constraint is the disentangled solution, where:*

$$I(H_s) \rightarrow S$$
$$I(H_1) \rightarrow P_1$$
$$I(H_2) \rightarrow P_2$$

*This implies $I(H_1; H_s) \rightarrow 0$, $I(H_2; H_s) \rightarrow 0$, and $I(H_1; H_2) \rightarrow 0$.*

*Proof.* The total information $S + P_1 + P_2$ must be captured by the latent subspaces $H_s, H_1, H_2$ to satisfy the reconstruction fidelity constraint. The total rate (rank) to be minimized is $R_{\text{total}} \approx k_s + k_1 + k_2 \approx I(H_s) + I(H_1) + I(H_2)$. We compare two possible solutions that both satisfy reconstruction: □

1. **Solution 1: The Disentangled (Optimal) Allocation**

   - $H_s$ is allocated the shared information: $I(H_s) \approx S$.
   - $H_1$ is allocated the private information for $X_1$: $I(H_1) \approx P_1$.
   - $H_2$ is allocated the private information for $X_2$: $I(H_2) \approx P_2$.
   - **Check Constraints:**
     - Decoder $g_1$ receives $H_s \oplus H_1$, which contain $(S, P_1)$. This is sufficient to reconstruct $X_1$ (as $H(X_1) = S + P_1$).
     - Decoder $g_2$ receives $H_s \oplus H_2$, which contain $(S, P_2)$. This is sufficient to reconstruct $X_2$ (as $H(X_2) = S + P_2$).
   - **Total Rate:** $R_{\text{optimal}} \approx S + P_1 + P_2$.

2. **Solution 2: A Non-Disentangled (Suboptimal) Allocation** Consider a solution where the model relies only on the private pathways and $H_s$ is pruned (i.e., $k_s \to 0$).

   - $H_s$ is allocated no information: $I(H_s) = 0$.
   - To reconstruct $X_1$, $H_1$ *must* now capture all of $X_1$'s information: $I(H_1) \approx H(X_1) = S + P_1$.
   - To reconstruct $X_2$, $H_2$ *must* now capture all of $X_2$'s information: $I(H_2) \approx H(X_2) = S + P_2$.
   - **Check Constraints:**
     - Decoder $g_1$ receives $(0, S + P_1) \to$ Reconstructs $X_1$. (Satisfied)
     - Decoder $g_2$ receives $(0, S + P_2) \to$ Reconstructs $X_2$. (Satisfied)
   - **Total Rate:** $R_{\text{suboptimal}} \approx 0 + (S + P_1) + (S + P_2) = (S + P_1 + P_2) + S$.
   - **Solution 3: The "All-Shared" (Entangled) Allocation** Consider a solution where all information is forced into the shared subspace $H_s$, and private spaces are pruned ($k_1 \to 0, k_2 \to 0$).
     - $H_s$ captures all information: $I(H_s) \approx S + P_1 + P_2$.
     - $H_1, H_2$ are empty.
     - **Check Constraints:**
       * Decoder $g_1$ receives $(S + P_1 + P_2, 0)$. To reconstruct $X_1$, it must extract $S + P_1$ and *suppress* $P_2$ (which acts as high-variance noise to $X_1$).
       * Decoder $g_2$ receives $(S + P_1 + P_2, 0)$. To reconstruct $X_2$, it must extract $S + P_2$ and *suppress* $P_1$.
     - **Total Rate:** $R_{\text{all-shared}} \approx (S + P_1 + P_2) + 0 + 0$.

**Conclusion:** We compare the three scenarios:

- **Solution 2 (All-Private)** is strictly suboptimal in terms of rank ($R \approx R_{\text{optimal}} + S$) due to the redundant encoding of shared information $S$. The rank minimization objective $R_{\text{total}}$ actively penalizes this.

- **Solution 3 (All-Shared)** is rank-equivalent to the optimal solution ($R \approx S + P_1 + P_2$) but optimizationally unstable. For this solution to hold, Decoder $g_1$ must learn to output distinct zeros for the dimensions of $H_s$ corresponding to $P_2$, effectively treating them as noise. Learning to actively suppress high-variance signal via "hard zeros" is difficult for gradient descent.

- **Solution 1 (Disentangled)** allows Decoder $g_1$ to ignore $P_2$ by architectural design (since $g_1$ is not connected to $H_2$).

Therefore, while Solutions 1 and 3 are global minima for the rank objective, Solution 1 is the *stable* minimum. The interplay between the rank objective (eliminating Sol 2) and the split-decoder architecture (favoring Sol 1 over Sol 3) drives the model toward the disentangled representation.

# B. Methods and Materials

## B.1. Data simulation

### B.1.1. UNI-MODAL PARAMETRIC $\mathbf{A}$

We created a simple but highly flexible simulation for generating a single data modality controlled by data characteristics as parameters. This allows for rapid testing of model performance across a wide range of data structures and complexities. The simulation generates an observed data matrix $\mathbf{A} \in \mathbb{R}^{N \times n}$ from a latent representation matrix $\mathbf{Z} \in \mathbb{R}^{N \times k}$, where $N$ represents the number of samples, $n$ the ambient dimension, and $k$ the latent dimension (the ID). We first sample $\mathbf{Z}$ from a chosen distribution $P$ with options {Beta, Gaussian, Poisson, Binomial, Gumbel, Uniform, Weibull}. The latent variables undergo no, one, or more rounds (controlled by the nonlinearity level) of nonlinear transformations. The specific function is determined by the nonlinearity type parameter $f(x) \in \{x^2, \max(0, x), \frac{1}{1+e^{-x}}, \sin(x)\}$. The transformed latent variables $z'$ are linearly projected into the higher-dimensional data space using a sparse weight matrix $\mathbf{W} \in R^{k \times n}$ controlled by a connectivity parameter. Finally we add noise $\epsilon$ with a desired variance and apply dropout.

$$\mathbf{A} = z'\mathbf{W} + \epsilon \text{ with } z' = f(z), z \sim P, \epsilon \sim \mathcal{N}(0, \sigma) \tag{1}$$

### B.1.2. MULTI-MODAL PARAMETRIC $\mathbf{B}$

To evaluate our model's ability to handle multi-modal data, we extend the uni-modal framework to generate two data matrices, $\mathbf{B}_1$ and $\mathbf{B}_2$. The key difference is the use of a composite latent structure comprising both shared variables $\mathbf{Z}_s \in \mathbb{R}^{N \times k_s}$ that influence both modalities and modality-specific variables $\mathbf{Z}_m \in \mathbb{R}^{N \times k_m}$ that affect only one. Based on our experimental configurations, the shared variables $\mathbf{Z}_s$ are sampled from either a Binomial distribution (when $k_s = 2$) or a Gaussian Mixture Model for $k_s > 2$. The modality-specific variables $\mathbf{Z}_{m1}$ and $\mathbf{Z}_{m2}$ are sampled from Poisson and Weibull distributions, respectively. For each modality m, a complete latent representation is formed by concatenating the shared and modality-specific variables. This combined representation is then projected into the data space using a modality-specific sparse weight matrix $\mathbf{W}_m$. For simplicity, the nonlinear transformation step was omitted in these experiments. As in the uni-modal case, each data matrix is independently corrupted by adding Gaussian noise and applying a dropout mask.

### B.1.3. MULTI-OMICS $\mathbf{C}$

To generate a realistic multi-modal dataset with a known causal structure, we designed a multi-stage simulation inspired by a limited set of known processes in gene expression and translation. The result is the multi-omics data simulation $\mathbf{C}$. The full procedure and parameterization are detailed in Algorithm 2.

**Genomic and cellular architecture:** First, we establish a fixed genomic architecture where 20,000 genes are organized into co-regulated gene clusters and programs, including a core set of housekeeping genes. We then simulate a cell lineage tree originating from three distinct stem cell lines. Cellular differentiation is modeled as the progressive and stochastic silencing of gene programs, which defines each cell type's identity and its baseline gene expression potential.

**Simulation flow:** With the genomic architecture fixed, we simulate the molecular state for each cell in a causal cascade. We determine a ground-truth chromatin accessibility profile for each cell. This profile is then perturbed by cell-specific variables simulating cell cycle phase and a general stress response. The resulting accessibility profile, along with gene-specific transcription efficiencies and cell-specific DNA damage, dictates the "true" mRNA counts. In turn, these mRNA counts determine protein abundance via translational efficiencies (dependent on amino acid composition and ribosome availability) and protein degradation rates. Finally, each modality is subjected to a separate technical noise model that simulates artifacts like capture efficiency, batch effects, and stochastic dropout to produce the final observed data matrices.

---

**Algorithm 2** Multi-Omics Data Simulation $\mathbf{C}$

---

1: **Inputs:** $N_{genes}$, $N_{cells}$, $N_{stemcells}$
2: **Outputs:** Modality matrices $\mathbf{C_1}$, $\mathbf{C_2}$, and causal variables $\mathbf{Z}$.
3:
4: **Part 1: Generate Fixed Genomic and Cell Type Architecture (seed=0)**
5: **Define Gene Properties:**
6:     For each gene $g \in \{1, ..., N_{genes}\}$:
7:         Gene length $L_g \sim \lfloor (\text{NegativeBinomial}(300, 0.02))^{3.9} \rfloor + 150$
8:         Base expression $\mu_{base,g} \sim \text{NegativeBinomial}(1000, 0.01) + 1$
9:         Transcription probability $\tau_{trans,g} = f_{trans}(L_g) + \mathcal{N}(0, 0.05^2)$
10:        mRNA degradation prob. $\delta_{RNA,g} = f_{deg,RNA}(L_g) + \mathcal{N}(0, 0.05^2)$
11:        Protein length $L'_g = \lfloor L_g/3 \rfloor$
12:        Translation ease $\phi_{AA,g} = f_{ease}(\text{AA composition of } g)$
13:        Translation probability $\tau_{prot,g} = f_{trans}(L'_g)$
14:        Protein degradation prob. $\delta_{PROT,g} = f_{deg,PROT}(L'_g)$
15: **Define Regulatory Structure:**
16:     Generate gene cluster assignment matrix $\mathbf{M}_{C \to G}$
17:     Generate gene program matrix $\mathbf{M}_{P \to C}$ where $M_{P \to C}[p, c] \sim \text{Bernoulli}(0.1)$
18:     Generate cell type hierarchy matrix $\mathbf{M}_{CT \to P}$ via iterative program silencing
19:     Compute cell type to gene activity: $\mathbf{M}_{CT \to G} = \text{threshold}(\mathbf{M}_{CT \to P}\mathbf{M}_{P \to C}\mathbf{M}_{C \to G})$
20: **Define Perturbation Effects:**
21:     Stress closure vector (by gene) $\mathbf{c}_{stress} = \text{cluster\_closure} \cdot \mathbf{M}_{C \to G}$
22:     Cell cycle amplification matrix $\mathbf{E}_{cc}$ and openness matrix $\mathbf{M}_{open,cc}$
23:
24: **Part 2: Simulate All Cells**
25: **Sample Cell-Specific Latent Variables** for $i \in \{1, ..., N_{cells}\}$:
26:     Cell type $ct_i \sim \text{Categorical}(\mathbf{1}/N_{CT})$
27:     Stress level $s_i \sim \text{Bernoulli}(0.05)$
28:     Cell cycle phase $cc_i \sim \text{Categorical}([0.1, 0.2, 0.3, 0.4])$
29:     Transcription activity $a_{trans,i} \sim \text{Clamp}(\text{Poisson}(4), 0, 9) + 1$
30:
31:     Baseline open chromatin: $\mathbf{O}_{base,i,:} = a_{trans,i} \cdot \mathbf{M}_{CT \to G}[ct_i, :]$
32:     Cell cycle modulation: $\mathbf{O}_{cc,i,:} = (\mathbf{O}_{base,i,:} + \mathbf{E}_{cc}[cc_i, :]) \odot \mathbf{M}_{open,cc}[cc_i, :]$
33:     Final ground truth open chromatin: $\mathbf{O}_{i,:} = \mathbf{O}_{cc,i,:} \odot (1 - s_i \cdot \mathbf{c}_{stress})$
34:
35: **Transcription (RNA-seq)**
36:     DNA damage $p_{dmg,i} \sim \text{Beta}(1, 2) \cdot \text{level}(ct_i)/10$
37:     Potential transcription: $\mathbf{R}_{pot} = \mathbf{O} \odot \boldsymbol{\mu}_{base}$
38:     Real mRNA counts: $\mathbf{R}_{RNA} = \lfloor \mathbf{R}_{pot} \odot (1 - \mathbf{p}_{dmg}) \odot \boldsymbol{\tau}_{trans} \odot e^{-\boldsymbol{\delta}_{RNA}} \rfloor$
39:     Technical group assignment: $b_{RNA,i} \sim \text{Categorical}(\mathbf{1}/3)$
40:     Observed mRNA: $\mathbf{C_{RNA,i}} = \lfloor \mathbf{R}_{RNA,i} \cdot \epsilon_{RNA}(b_{RNA,i}) \rfloor \odot (1 - d_{RNA}(b_{RNA,i}))$
41:
42: **Translation (Proteomics)**
43:     Sample protein machinery variables for cell $i$:
44:         Ribosome availability $f_{ribo,i} \propto \sum_{g \in \text{rDNA}} R_{RNA,i,g}$
45:         tRNA availability $a_{tRNA,i} \sim \text{Beta}(2, 1)$
46:         Proteasome activity $a_{prot,i} \sim \text{Beta}(1, 2)$
47:     Ribosome efficiency: $\boldsymbol{\eta}_{ribo,i} = (\boldsymbol{\phi}_{AA}/0.05 \odot \boldsymbol{\tau}_{prot}) \cdot (f_{ribo,i} / \sum_g R_{RNA,i,g} \cdot a_{tRNA,i})$
48:     Proteins translated: $\mathbf{P}_{trans} = \mathbf{R}_{RNA} \odot \boldsymbol{\eta}_{ribo}$
49:     Real protein counts: $\mathbf{P}_{real} = \mathbf{P}_{trans} \odot e^{-(\boldsymbol{\delta}_{PROT} \odot \mathbf{a}_{prot})}$
50:     Technical group assignment: $b_{PROT,i} \sim \text{Categorical}(\mathbf{1}/2)$
51:     Observed protein: $\mathbf{C_{Protein,i}} = \lfloor \mathbf{P}_{real,i} \cdot \epsilon_{PROT}(b_{PROT,i}) \rfloor \odot (1 - d_{PROT}(b_{PROT,i}))$
52:
53: **Store** final matrices $\mathbf{C_1}$, $\mathbf{C_2}$ and all latent variables $\mathbf{Z}$.

---

## B.2. Real Datasets

### B.2.1. NINFEA DATA

**Dataset access and description:** The "Non-Invasive Multimodal Foetal ECG-Doppler Dataset for Antenatal Cardiology Research (NInFEA)" dataset (Sulas et al., 2021) is available under Open Data Commons Attribution License on PhysioNet (doi: 10.13026/c4n5-3b04). It consists of 60 recordings from 39 pregnant women between the 21st and 27th week of gestation. Each record contains multiple synchronized signals. For our experiments, we utilized the following key modalities described in the dataset :

- Abdominal Electrophysiology (fECG view): 24 unipolar channels recorded from the maternal abdomen and back at 2048 Hz. These signals contain the target fetal ECG (fECG) signal heavily mixed with the maternal ECG (mECG) and other noise.

- Thoracic Electrophysiology (mECG view): 3 bipolar channels from the maternal thorax, primarily capturing the maternal ECG for reference.

- Maternal Respiration: A single channel from a piezo-resistive respiration belt, sampled at 2048 Hz.

- Pulsed-Wave Doppler (PWD): Provided as a single wide image in bitmap (.bmp) format, representing the Doppler velocity spectrum of the mechanical fetal heart activity.

**Data preprocessing:** The recordings were of different lengths, so we cropped ECG and respiration signals to the first 15351 data points per channel. Before cropping PWD images, we aligned them to be centered around the baseline (horizontal middle line of the ultrasound) and cropped them to $263 \times 2128$ pixel with three channels. We normalized all modalities to a range of $[0, 1]$ per channel.

**Ethical Considerations:** As stated in the original publication, the dataset was collected with approval from the Independent Ethics Committee of the Cagliari University Hospital (AOU Cagliari) and all volunteers provided signed informed consent.

### B.2.2. AUDIO MNIST

We used the Audio MNIST dataset from Becker et al. (2023), which provided 5000 train and 1000 test samples per digit derived from pairing MNIST (Deng, 2012) and FSDD (Jackson et al., 2018). MNIST images are black and white images of size $24 \times 24$ and audio samples are spectrograms of size $112 \times 112$.

### B.2.3. SO2SAT

We utilized the So2Sat LCZ42 benchmark dataset (Version 2) (Zhu et al., 2020), a large-scale multi-modal dataset designed for Local Climate Zone (LCZ) classification. The dataset consists of co-registered image patches acquired from the Sentinel-1 (Synthetic Aperture Radar) and Sentinel-2 (Multispectral Optical) satellite constellations. The data was obtained from the official repository hosted by the Technical University of Munich: https://mediatum.ub.tum.de/1454690.

The dataset contains 400,673 labeled image pairs with pixel dimensions $32 \times 32$ corresponding to 17 distinct Local Climate Zones (LCZ), covering 42 urban agglomerations across diverse cultural and environmental regions. We utilized the standard "Culture-10" split provided in Version 2, where the training set consists of patches from 42 cities, and the validation set consists of patches from the western half of 10 independent cities to test generalization.

From the raw 18-channel tensor provided by the dataset, we extracted two distinct physical views (modalities) for our experiments:

- Modality 1 (SAR Imagery): We utilized the full 8-channel Sentinel-1 data, which includes both the real and imaginary parts of the VH and VV polarizations, as well as intensity and covariance features. This modality captures structural and geometric properties such as surface roughness and material density.

- Modality 2 (Optical Imagery): We extracted the 3 visible spectral bands (Red, Green, Blue) from the Sentinel-2 data (channels 8–10 of the combined tensor). This modality captures visual semantic features such as vegetation color and building textures.

To address the disparate statistical properties of the two modalities, we applied specific preprocessing pipelines prior to training. The SAR modality follows a heavy-tailed distribution with extreme outliers. To handle negative values in the raw complex channels and compress the dynamic range, we first shifted the data to be non-negative by subtracting the global minimum, followed by a log-transformation: $x' = \log(1 + (x - \min(x)))$. We computed the 99th percentile of the training data and clipped values above this threshold to mitigate the impact of extreme specular reflections (outlier clipping). Finally, we applied channel-wise Z-normalization using statistics $(\mu, \sigma)$ calculated from the whole set to both modalities.

### B.2.4. NYU DEPTH V2

We utilized the NYU Depth V2 dataset (Nathan Silberman & Fergus, 2012), which consists of 1,449 dense pairs of aligned RGB and depth images captured from 464 diverse indoor scenes and 27 unique scene types. For our experiments, we utilized a random 90/10 split, allocating 90% of the samples for training and 10% for validation. To align the data with the requirements of the pre-trained autoencoders, we applied specific preprocessing pipelines. Both modalities were resized to a spatial resolution of $256 \times 256$ pixels, utilizing bilinear interpolation for RGB images and nearest-neighbor interpolation for depth maps to preserve edge distinctness.

- **RGB Imagery:** Pixel values were normalized to the range $[0, 1]$ by dividing by 255.

- **Depth Imagery:** Raw depth values were converted from millimeters to meters. To mitigate sensor noise and focus on the relevant indoor range, we clipped the maximum depth distance to 10 meters. The resulting maps were then linearly normalized to the $[0, 1]$ range, where 0 represents the camera plane and 1 represents a distance of 10 meters.

### B.3. Architectures

All autoencoders trained in this work consist of symmetric encoder-decoder architectures.

### B.3.1. BASE AUTOENCODER HYPERPARAMETER SEARCH

To ensure our method's performance was not confounded by a suboptimal base model, we conducted a hyperparameter search for the autoencoder architecture and training parameters using the Optuna (Akiba et al., 2019) framework. The goal was to identify a sufficiently deep and wide architecture capable of achieving high reconstruction fidelity (i.e., near-lossless compression) when using a large, fixed bottleneck dimension. The search was performed on the RNA modality of dataset **C** (N=30,000). We ran a multi-objective optimization over 100 trials, aiming to simultaneously maximize the reconstruction goodness-of-fit (mean $R^2$) and minimize the autoencoder's validation loss (MSE). The $R^2$ is explained in B.4. The search space for the optimization included key architectural and training parameters: network depth ($\{2, 3, 4, 6\}$), width as a fraction of input dimension ($\{0.25, 0.5, 0.75, 1.0\}$), learning rate (log-uniform between $10^{-5}$ and $10^{-3}$), batch size ($\{64, 128, 256, 512\}$), weight decay (log-uniform between $10^{-6}$ and $10^{-4}$), dropout ($\{0.0, 0.1, 0.2\}$), and early stopping patience ($\{10, 50\}$). From the resulting Pareto front of optimal solutions, we selected a final configuration that offered the best balance between the two objectives. This balanced solution was identified by normalizing both objective scores across the Pareto front and selecting the trial with the highest geometric mean. The final parameters were a depth of 2, width factor 0.5, dropout 0.1, batch size 512, learning rate $10^{-5}$, weight decay $2 \times 10^{-5}$, and early stopping of 50 epochs.

### B.3.2. UNI-MODAL SIMULATIONS

For the uni-modal parametric simulations, we employed a fully-connected autoencoder. Both the encoder and decoder consisted of 2 hidden layers. The width of these layers was set to be equal to the input data dimension (50 features), corresponding to a width factor of 1.0. A dropout rate of 0.1 was applied to all hidden layers for regularization. The central layer was our low-rank adaptable layer, initialized with a rank of 20. For the more complex and high-dimensional uni-modal omics simulation benchmark, the architecture was scaled appropriately. We used an autoencoder of the same depth (2 layers), but with a width factor of 0.5 relative to the input dimension. The adaptable layer was initialized with a highly overcomplete rank of 1000. The dropout rate was kept at 0.1.

### B.3.3. 3D SHAPES

For the 3D manifold experiments, we use the same 2-layer autoencoder with a hidden width ratio of 1.0 and dropout 0.1. The central adaptable layer was initialized with a rank of 3, matching the ambient dimension of the data.

### B.3.4. MULTI-MODAL SIMULATIONS

For the multi-modal parametric simulations, the model consisted of modality-specific encoders feeding into three adaptable bottleneck layers representing the shared, private 1, and private 2 subspaces as described in the Methods section. Each modality's encoder and decoder had 2 hidden layers with a width factor of 1.0 relative to the input dimension (200 features) and a dropout rate of 0.1. All three adaptable layers were initialized with a rank of 100. For the higher-dimensional multi-omics simulation, a similar architecture was used but with a width factor of 0.5. To accommodate the greater complexity, the initial rank for each of the three adaptable layers was increased to 500.

### B.3.5. NINFEA

Due to the heterogeneous data types in the NInFEA dataset, which includes both time-series (ECG, respiration) and image-based (PWD) modalities, we utilized a hybrid architecture. Each modality was first processed by a dedicated encoder pathway consisting of convolutional layers to extract relevant features. These features were then passed to a fully-connected network with 2 hidden layers, each containing 512 units, with a dropout rate of 0.1 applied for regularization. The adaptable bottleneck layers for the shared and private subspaces were all initialized with a rank of 100.

### B.3.6. AUDIO MNIST

For the Audio MNIST dataset, we processed the image modality by a convolutional encoder with two Conv2d layers (16 and 64 channels, respectively), each with a 5x5 kernel and followed by 2x2 max-pooling. This was connected to two fully-connected layers of width 800. The audio modality was similarly processed with 10x10 kernels and hidden layers of width 1600. The adaptable bottleneck layers for the shared, image-specific, and audio-specific subspaces were each initialized with a rank of 200.

### B.3.7. SO2SAT

We use Convolutional Neural Network (CNN) branches for both the SAR and Optical modalities. The encoders consist of a stack of strided 2D convolutional layers (kernel $3 \times 3$, stride 2) with ReLU activation and dropout, which progressively downsample the input tensors before flattening (2048) and linearly projecting them into a lower-dimensional latent vector of dimension 200. The decoders mirror this architecture, taking the concatenated shared and specific latent vectors (400 total) and passing them through a linear layer followed by a series of transposed convolutions (deconvolutions) to upsample the features back to the original $32 \times 32$ spatial resolution for reconstruction.

### B.3.8. NYU DEPTH V2

For the NYU Depth V2 dataset, we leveraged the pretrained KL-regularized variational autoencoder provided by https://huggingface.co/stabilityai/sd-vae-ft-mse based on latent diffusion (Rombach et al., 2021) as a feature extractor for the RGB modality. For depth, we used the Depth Anything V2 (Yang et al., 2024) for feature extraction from https://huggingface.co/depth-anything/Depth-Anything-V2-Small-hf and built a simple residual convolutional decoder that we trained for reconstruction. The decoder utilizes three stages of residual blocks and sub-pixel convolutions with Group Normalization and GELU activations to progressively upsample features, reducing channel depth from 256 to 32 before a final Sigmoid projection. Both the RGB and Depth inputs were resized to $256 \times 256$ and encoded by the frozen encoders into high-dimensional feature maps. These maps were flattened and projected via trainable linear adapters into the disentangled subspaces. The adaptable bottleneck layers were initialized with a rank of 500.

## B.4. Metrics

### B.4.1. DISTORTION METRICS

**Goodness of fit $R^2$**

The coefficient of determination $R^2$ is a measure of reconstruction goodness of fit. We measure it for each feature $j$ over $10\%$ of training samples $i$ and report the average over features (observables in the data) $n$.

$$R^2 = \frac{1}{n} \sum_{j=1}^{n} \left( 1 - \frac{\sum_{i=1}^{N} (x_{ij} - \hat{x}_{ij})^2}{\sum_{i=1}^{N} (x_{ij} - \bar{x}_j)^2} \right) \tag{2}$$

## MSE and RMSE

The Mean Squared Error (MSE) is a direct measure of distortion, calculating the average of the squared differences between the original data $x$ and the reconstructed data $\hat{x}$. It is highly sensitive to large errors due to the squaring operation. The Root Mean Squared Error (RMSE) is the square root of the MSE and is often preferred as it returns the error metric to the same scale as the original data.

$$\text{MSE} = \frac{1}{ND} \sum_{i=1}^{N} \sum_{j=1}^{D} (x_{ij} - \hat{x}_{ij})^2 \tag{3}$$

$$\text{RMSE} = \sqrt{\text{MSE}} \tag{4}$$

## Explained Variance Score

The Explained Variance Score measures the proportion of the variance in the original data that is accounted for by the model's reconstructions. A score of 1.0 indicates that the model perfectly explains the variance of the data. It is calculated as:

$$\text{Explained Variance} = 1 - \frac{\text{Var}(x - \hat{x})}{\text{Var}(x)} \tag{5}$$

### B.4.2. EVALUATION METRICS

### Classification accuracy

We use a 5-fold cross-validation logistic regression classifier to assess how well representations linearly predict ground truth class labels. We used the sklearn implementation with max_iter=1000, class_weight='balanced', solver='liblinear', and random_state=42 to predict class labels $y_i$. The accuracy is the number of correct predictions divided by the total number of predictions.

$$\text{Accuracy} = \frac{1}{N} \sum_{i=1}^{N} \mathbb{I}(y_i = \hat{y}_i) \tag{6}$$

We also report balanced accuracy to account for potential class imbalance, defined as the arithmetic mean of the recall scores per class:

$$\text{Balanced Accuracy} = \frac{1}{K} \sum k = 1^K \frac{1}{N_k} \sum_{i \in C_k} \mathbb{I}(y_i = \hat{y}_i) \tag{7}$$

where $K$ is the number of classes, $C_k$ is the set of indices belonging to class $k$, and $N_k$ is the total number of samples in class $k$.

### B.5. FiGuRO hyperparameter search

All tested hyperparameters are reported in Table 1 along with their estimated ranks and sample sizes. Architecture is described in B.3.2, training is described in B.6.1.

### B.6. Training

In this section we describe general training frameworks and the specific hyperparameter sets used per experiment. We used the Adam optimizer and mean squared error (MSE) loss for all training runs.

### B.6.1. UNI-MODAL SIMULATIONS

Models for the parametric simulations were trained using a learning rate of $10^{-4}$ and a weight decay of $2 \times 10^{-5}$. We used a batch size of 128 and trained for a maximum of 5000 epochs. The FiGuRO rank optimization procedure began after an initial pre-training phase with 50 epochs early stopping. For the high-dimensional omics simulations, the training configuration was adapted for the larger dataset. We used a lower learning rate of $10^{-5}$ with the same weight decay and a larger batch size of 512.

### B.6.2. 3D SHAPES

The models were trained with a learning rate of $10^{-3}$ and a weight decay of $2 \times 10^{-5}$. We used a batch size of 512 and trained for a maximum of 5000 epochs with early stopping 50.

### B.6.3. MULTI-MODAL SIMULATIONS

The parametric simulation models were trained for a maximum of 5000 epochs (50 early stopping) with a learning rate of $10^{-4}$, a weight decay of $10^{-5}$, and a batch size of 128. For the larger multi-omics simulation, the training parameters were adjusted to a learning rate of $10^{-5}$, a weight decay of $2 \times 10^{-5}$, and a batch size of 1024.

### B.6.4. NINFEA

The models were trained for a maximum of 5000 epochs with a batch size of 8, a learning rate of $10^{-4}$, and no weight decay. The rank adaptation process was initiated after an initial training phase, which was run with an early stopping patience of 100 epochs. Once optimization began, the final rank convergence was determined with a patience of 10 epochs.

### B.6.5. AUDIO MNIST

The training process consisted of two distinct phases. In the first phase, we performed rank optimization for a maximum of 5000 epochs using a batch size of 512 and a learning rate of $10^{-3}$ with a linear decay schedule. The FiGuRO algorithm was initiated after an initial pre-training phase, which ran until the validation loss did not improve for 50 epochs. Rank convergence was then determined with a patience of 10 epochs and based on the Explained Variance Score, which was more robust for the audio modality. In the second phase, the model with its now-fixed ranks was fine-tuned for an additional up to 1000 epochs using a lower learning rate of $10^{-5}$. This fine-tuning stage utilized early stopping with a patience of 50 epochs based on the validation loss to ensure optimal performance.

### B.6.6. SO2SAT

We used the same training setup as for Audio MNIST, pretraining the full model for up to 1000 epochs with early stopping 50 and a learning rate decay schedule from $10^{-4}$ to $10^{-7}$. FiGuRO was initiated after the pre-training phase with default parameters for another maximum 1000 epochs. We again used Adam optimizer with weight decay $2 \times 10^{-5}$ and a batch size of 512.

### B.6.7. NYU DEPTH V2

We trained the multimodal fusion and adaptable bottleneck layers for a fixed duration of 1000 epochs, while keeping the pre-trained uni-modal VAE backbones frozen. We used the Adam optimizer with a learning rate of $10^{-4}$, weight decay of $10^{-6}$, and a batch size of 16. The FiGuRO rank optimization was initiated after a warmup phase of 100 epochs. Rank reduction was performed every 5 epochs with an energy threshold of $0.01$, a patience of 10, and a distortion budget of $0.05$ based on the $R^2$ score. We further employed mixed-precision training to accommodate the memory requirements of the $256 \times 256$ inputs.

## B.7. Robustness experiments

### B.7.1. ROBUSTNESS TO DATA CHARACTERISTICS

We tested the robustness of our approach with respect to key properties of the simulated data: number of training samples, generative variable distribution, connectivity of generative variables, depth and function of nonlinearity, and noise and dropout. To conduct these experiments, we utilized our parametric uni-modal simulation **A** to generate a suite of synthetic datasets. Starting from a default data configuration, we systematically varied one parameter at a time while holding all others constant to isolate its effect on our model's performance. We trained each configuration with the default simulation training setup described in B.6 for 5 random seeds. All tested values and results are shown in Figure 1.

### B.7.2. SAMPLE SIZE REQUIREMENTS

The number of samples (N) required for FiGuRO to produce a reliable ID estimate is not an absolute value but is instead dependent on the complexity of the autoencoder's encoder network. This relationship can be understood through a sample-

to-capacity ratio, $\alpha$, which relates the number of samples to the number of encoder parameters $\mathcal{C}_e$ per latent dimension $k$ as $\alpha = \frac{kN}{\mathcal{C}_e}$ (Schuster & Krogh, 2021). In our experiments, we observed that robust estimates for the ground truth ID were achieved for sample sizes of N$\geq$5000. For the specific encoder architecture used in these tests, this corresponds to a sample-to-capacity ratio $\alpha \approx 20$. Since this autoencoder had a width ratio of 1 instead of 0.5 due to the small ambient dimension, we expect the $\alpha$ to be closer to 10. This finding is consistent with prior work on autoencoders, which suggests that a ratio of $\alpha \geq 10$ is generally required to sufficiently constrain the model (Schuster & Krogh, 2021).

### B.7.3. DISTORTION METRIC COMPARISON

We also tested the robustness of different general options for distortion metrics under varying nonlinearities and dropout levels. We tested $R^2$, MSE, RMSE, Explained Variance Score, and McFadden$R^2$ as potential distortion metrics. $R^2$, Explained Variance Score, and McFadden$R^2$ could be used with the absolute distortion threshold $\lambda = 0.05$ we introduced in the methods since they are supported in the range of $[0, 1]$. The remaining metrics we used with relative distortion thresholds, where we compute the maximum distortion as $1.05$ times the initial metric value. The nonlinearity types we tested were $x^2, \max(0, x), \frac{1}{1+e^{-x}}, \sin(x)$. We trained each configuration with the default simulation training setup described in B.6 for 5 random seeds. Figure 2 shows the average deviation from the true ID per metric and varied parameter.

### B.8. Baseline method implementation

### B.8.1. CLASSICAL METHODS

We compare our approach against several non-neural network methods for ID estimation. These methods range from global linear techniques to local, neighbor-based estimators. We primarily utilized implementations from the `scikit-learn` and `skdim` python libraries, testing each method across a range of its key hyperparameters to ensure a fair and robust comparison.

- **Linear, Global**:

  - **Principal Component Analysis (PCA)**: ID is estimated as the number of components needed to explain a certain amount of variance. We used the `scikit-learn` implementation with variance `threshold` values of $\{0.8, 0.9, 0.95\}$ for the variance parameter $\sigma$.
  - **Singular Value Decomposition (SVD)**: The ID is estimated directly as the matrix rank, computed via `PyTorch` for explained energy ratios $EE \in \{100, 95, 90, 80\}\%$.

- **Random Matrix Theory**:

  - **BEMA (Bulk Edge Marchenko-Pastur Analysis)** (Ke et al., 2021): ID is the number of eigenvalues ("spikes") that exceed the theoretical upper bound of the Marchenko-Pastur distribution. We tested `bulk_percentiles` $\%ile \in \{80, 90, 95, 99\}$ to define the noise region.

- **Local, Neighbor-Based**:

  - **Local PCA (lPCA)** (Kambhatla & Leen, 1997): A variant of PCA that averages estimates from local neighborhoods. Tested with `alphaFO` values of $\{0.0001, 0.001, 0.01, 0.05, 0.1, 0.5, 0.9\}$ for parameter $\alpha$.
  - **Correlation Integral (CorrInt)** (Camastra & Vinciarelli, 2002): A fractal dimension estimator. We performed a grid search over neighbor parameters $k_1'$ and $k_2'$ from the set $\{2, 5, 10, 20, 50, 100\}$.
  - **FisherS** (Albergante et al., 2019): An estimator based on the separability of classes, run with its default parameters.
  - **MiND_ML** (Rozza et al., 2012): A method based on the statistics of distances between nearest neighbors. Tested with the number of neighbors `k'` set to $\{2, 5, 10, 20, 50, 100\}$.
  - **Maximum Likelihood Estimator (MLE)** (Levina & Bickel, 2004): A widely used estimator based on nearest-neighbor distances. We performed a grid search over the noise parameter `sigma` in $\{0, 0.001, 0.01, 0.1\}$ and the number of neighbors `K` in $\{2, 5, 10, 20, 50, 100\}$.
  - **TLE** (Amsaleg et al., 2022): An estimator based on the two-sample log-likelihood of nearest neighbor distances. Tested with `epsilon` values of $\{10^{-10}, 10^{-5}, 10^{-4}, 10^{-3}, 10^{-2}, 0.1, 1.0\}$.
  - **TwoNN** (Facco et al., 2017): An estimator based on the ratio of distances to the first and second nearest neighbors. Tested with `discard_fraction` $\frac{r_2}{r_1}$ values of $\{0.0, 0.1, 0.2, 0.3, 0.5, 0.7, 0.9\}$.

### B.8.2. NEURAL-NETWORK METHODS

We implemented a small number of NN-based techniques that can be used for ID estimation. Their training procedures are described below.

**Loss cliff** (Bahadur & Paffenroth, 2020): What we refer to as the loss cliff is a standard way of using autoencoders to roughly estimate the minimum bottleneck capacity an autoencoder (AE) needs to effectively reconstruct the data described by Bahadur & Paffenroth (2020). The core idea is that the reconstruction error will remain low as the latent dimension, $k$, is reduced, until $k$ drops below the true ID, at which point the error increases sharply, forming a "cliff" or "elbow". We employ the same autoencoder as four our method except from the decomposition layer. The training objective and training hyperparameters are the same as for our method, described in B.6. To find the cliff, we define an upper and lower limit for $k$ and train autoencoders with those bottlenecks. Instead of testing many dimensions, we search for 20 steps, evaluating the midpoint bottleneck between the lower and upper bound. Bound bottlenecks are dynamically updated based on whether the reconstruction loss was above or below the threshold $\mathcal{L}_{min} \times (1 + \lambda')$. We stop as soon as the range is $\leq 10$ epochs. This way we narrow down the range of the ID, but it requires training up to 21 models. We repeat this for three random seeds.

**Rank Reduction Autoencoder** (Mounayer et al., 2025): We implemented the Rank Reduction Autoencoder (RRA) from Mounayer et al. (2025) with a slight modification learning the weights as matrix decompositions as in our proposed approach. Ranks are pruned based on the rank reduction threshold $\gamma$ until no more singular values are above $\gamma$. We use the same architecture and training hyperparameters as for our approach and test thresholds for $\gamma \in [0.0001, 0.1]$. We again repeat the experiment for the same three random seeds as the other methods.

**ARD-VAE** (Saha et al., 2025): Another NN-based method we compare to is the ARD-VAE, a Variational Autoencoder (Kingma & Welling, 2022) with an Automatic Relevance Determination (ARD) prior (Saha et al., 2025). We implemented it according to the setup described below. However, VAEs collapsed for some random seeds even with low beta terms, potentially due to the high latent dimension and capacity of the networks we needed to train.

We again use the same architecture and training hyperparameters as for our approach, but add additional latent dimensions to model not just $\mu$ but also $\sigma^2$ for the approximate posterior $q(z|x) = \mathcal{N}(z|\mu, \text{diag}(\sigma^2))$, which latents $z$ are sampled from. Our implementation is based on the principles outlined by the original authors but uses a corrected loss function to ensure mathematical validity. The key to this method is the hierarchical ARD prior placed on the latent variables, where $\boldsymbol{\alpha}$ is a vector of learnable precision parameters. During training, the model optimizes the evidence lower bound (ELBO) (Kingma & Welling, 2022), which includes a Kullback-Leibler (KL) divergence term between the approximate posterior and the ARD prior:

$$\mathcal{L} = \mathbb{E}_{q(z|x)}[\log p(x|z)] - D_{KL}(q(z|x)||p(z|\boldsymbol{\alpha})) \tag{8}$$

This objective encourages the model to prune uninformative dimensions by driving their corresponding precisions $\alpha_i$ towards zero (i.e., their variance towards infinity). After training, the intrinsic dimension was determined by the relevance score. The relevance score $\hat{\sigma}^2_{\mathbf{w}} = \mathbf{w}_{\hat{\sigma}} \odot \hat{\sigma}^2$ with weight vector $\mathbf{w}_{\hat{\sigma}}$ based on the Jacobian is better suited to find the relevant dimensions than defining a threshold of variance according to Saha et al. (2025). They determine the relevant dimensions based on $99\%$ explained variance in $\hat{\sigma}^2_{\mathbf{w}}$.

**FLIPD** (Kamkari et al., 2024): We additionally benchmark against the Fokker-Planck-based Local Intrinsic Dimension (FLIPD) estimator, a recently proposed method that leverages deep generative diffusion models. FLIPD estimates the ID by computing the rate of change of marginal log probabilities during the diffusion process, relying directly on the model's learned score function.

To estimate the Local Intrinsic Dimension (LID) using FLIPD, we adapted the base methodology to effectively handle the high dimensionality of our simulated single-cell data. The original authors demonstrated that FLIPD can be tractably computed within the latent space of a pretrained continuous encoder, such as the one used in Stable Diffusion (Rombach et al., 2021). To replicate this latent diffusion environment, we preceded our diffusion model with a pretrained bottleneck autoencoder that compresses the raw feature space into a 1000-dimensional latent space. To ensure comparability, this autoencoder is the same as the other neural network-based approaches in our evaluation. Additionally, we applied a mild Kullback-Leibler (KL) divergence penalty with weight $10^{-6}$ during pretraining to regularize the latent representations to mimic the process with Stable Diffusion. Within this latent space, we trained the diffusion model using a 3-layer multi-layer perceptron equipped with a bottleneck of dimension 500 and skip connections, matching the recommended configuration for high-dimensional manifolds. Rather than predicting the score directly (due to instability experienced), we utilized the epsilon-parameterization standard in Denoising Diffusion Probabilistic Models (DDPMs) to predict the added noise. The

LID was then computed using the exact adapted FLIPD formula for epsilon-predicting networks. Finally, we deviated from the default hyperparameter settings by expanding our evaluation across a broader range of timescale values, specifically $t_0$ in 0.01, 0.05, 0.1, 0.2, 0.3, 0.5, 0.7. While the original work frequently relies on automated knee-detection or small fixed values for lower-dimensional synthetic data, our preliminary experiments indicated that estimates remained artificially inflated at these lower bounds. By sweeping across a wider range of $t_0$ values, we ensure the estimator is given the best possible opportunity to identify the true manifold dimension across varying scales of noise resolution.

We note that our decision to maintain a uniform MLP architecture to ensure a fair algorithmic comparison highlights a fundamental limitation of diffusion-based LID estimators: their high sensitivity to the backbone capacity. As recently demonstrated by Yeats et al. (2025), standard MLPs can underfit highly complex manifolds. This biases estimates toward higher dimensions. More expressive architectures such as Diffusion Transformers, are often necessary to reliably fit the distribution and extract the local intrinsic dimension.

### B.8.3. MULTI-MODAL DATA DECOMPOSITION METHODS

To evaluate the performance of our proposed model in in the multi-modal setting, we compare it against five baseline methods, although these baselines assume linear mixing of the hidden variables and some do not give estimates on the individual spaces (CCA, DIVAS). These methods range from classical correlation techniques to modern spectral decomposition algorithms, each designed to separate shared (joint) and modality-specific (individual) variation from two data views, $X_1 \in \mathbb{R}^{N \times n_1}$ and $X_2 \in \mathbb{R}^{N \times n_2}$. For all methods, data views are first centered by subtracting the feature-wise mean. Where applicable, initial signal ranks are estimated using the Optimal Hard Threshold (OHT) method, which is based on Random Matrix Theory (Gavish & Donoho, 2014) and applied in PPD.

**JIVE (Joint and Individual Variation Explained)** (Lock et al., 2013): This method first estimates the individual signal ranks $(k_1, k_2)$ using OHT. It then performs SVD on the concatenated *signal bases* ($Z_{\text{basis}} = [U_1, U_2]$) and estimates the joint rank $k_J$ from $Z_{\text{basis}}$'s spectrum, again using OHT. The joint basis is used to project the signal matrices, yielding $J_1$ and $J_2$, with the residuals forming the individual components $I_1$ and $I_2$.

**AJIVE (Angle-based JIVE)** (Feng et al., 2018): We use the AJIVE implementation from the `py-jive` library (Feng et al., 2018). AJIVE also performs a full decomposition but uses a more robust procedure for rank estimation. It determines the joint rank $k_J$ by analyzing the principal angles between signal subspaces, using a permutation-based test to distinguish shared from non-shared variation. We utilized the implementation-default permutation test from the published code. This process is stochastic, and we use a fixed random seed for reproducibility.

**SLIDE (Structural Learning and Integrative Decomposition)** (Gaynanova & Li, 2017): SLIDE is a decomposition method that relies on iterative optimization to enforce a sparse, block structure on the factor loading matrix. After estimating the total signal ranks $(k_1, k_2)$ for each modality via OHT and estimating the joint rank $(k_J)$ using the concatenation method (similar to JIVE), SLIDE iteratively updates the projection matrices. This process minimizes the reconstruction error while ensuring the loadings adhere to a predefined, structured sparsity pattern that separates the joint and individual variation. We used a fixed sparsity penalty of 0.1 across all experiments to ensure consitency. This yields explicit reconstructions for the joint components ($\mathbf{J}_1, \mathbf{J}_2$) and individual components ($\mathbf{I}_1, \mathbf{I}_2$).

**ShIndICA (Shared and Individual Independent Component Analysis)** (Pandeva & Forré, 2023): ShIndICA is a method that decomposes the data based on statistical independence (a principle derived from Independent Component Analysis, ICA), rather than orthogonal variance (like SVD-based methods). The goal is to find sources ($\mathbf{Z}$) that are maximally non-Gaussian. The decomposition is achieved by maximizing the non-Gaussianity of the source signals while enforcing a penalty that aligns the shared sources across modalities. Crucially, ShIndICA implements an automatic model selection procedure by testing a range of possible joint ranks and selecting the one that minimizes the Normalized Reconstruction Error (NRE) on a held-out test split. This means that the method needs to be run over various settings. We tested joint rank options $[1, 5, 10, 20]$.

## C. Supplementary Results

### C.1. Robustness to hyperparameter choice and data characteristics

**Experimental setup.** We evaluated the robustness of our method's general ID estimation capability with a comprehensive hyperparameter grid search on dataset **A** over distortion budget $\lambda$, rank reduction frequency $\tau$, rank reduction threshold $\gamma$, and patience $\pi$. We further tested FiGuRO's stability across a range of data characteristics, including varying sample sizes, different generative distributions, levels and types of nonlinearity, and increasing levels of noise and dropout. Details on the experimental design and training are provided in B.5 and B.7. We further tested different distortion metrics with respect to nonlinearity and dropout, which affect data distribution and potentially the appropriateness of the metrics. Metrics are described in B.4. Lastly, we demonstrate the training dynamics on popular 3D Manifold datasets: Sphere, Swiss Roll, and S-curve.

**Results.** Across the entire hyperparameter sweep (N=1080 runs), our method gave a robust average estimate of $4.89 \pm 0.01$ standard error of the mean (SEM) for a true ID of 5. Summary statistics of the sweep are provided in Supplementary Table 1. This sweep revealed a mild dependence on $\lambda$, confirming that the distortion budget is the main driver of the stopping criterion. Dependency on $\gamma$ was negligible. This stability allowed us to select a single configuration balancing performance and speed $\{\lambda = 0.05, \tau = 10, \gamma = 0.01, \pi = 10\}$ for subsequent experiments. The method was also robust to most data characteristics. Accurate estimates were achieved for sample sizes N$\geq$5000 (see B.7.2 for a discussion on general sample size requirements). Certain generative distributions (Beta, Gumbel) and high degrees of nonlinearity led to underestimation. Conversely, high levels of dropout and noise (signal-to-noise ratio $< 7$) caused the method to overestimate the true ID (Supplementary Figure 1). Dropout also had a significant effect on the robustness of different distortion metrics. Increases of ID estimates with dropout were lowest among $R^2$ and the Explained Variance Score. Given different types of nonlinearities, $R^2$, MSE, and RMSE were most robust with smallest deviations from the true ID (Supplementary Figure 2). As a result, we identified $R^2$ as the best overall distortion metric for our experiments. Applied to popular 3D Manifolds, we see the necessity of rank reduction *and* increase. Figure 2 and Supplementary Figure 3 show that ranks often dropped too low before being increased to the true ID.

### C.2. Qualitative evaluation on $> 2$ modalities

**Experimental setup.** The Non-Invasive Multimodal Foetal ECG-Doppler (NInFEA) dataset (Sulas et al., 2021) contains 60 synchronized recordings of maternal and fetal physiology from high-density electrocardiography (ECG) from the maternal abdomen and thorax, a maternal respiration signal, and Pulsed-Wave Doppler (PWD) ultrasound of the fetal heart. While we do not have ground truth IDs for this data, we can evaluate the relative amounts of shared information between modalities relying on clinical intuition. We applied FiGuRO on all pairs of the four modalities and report the fraction of shared over total ranks. We also trained a model on three modalities at once. Architecture and training are adjusted to the temporal and image modalities (see B.6).

**Results.** Applying FiGuRO to the NInFEA dataset allowed us to quantitatively assess information overlap between modalities. Our estimated shared-to-total rank ratios (Supplementary Table 11) confirm established clinical and physiological knowledge: the fetal ECG (fECG) and Pulse-Wave Doppler (fPWD) show the largest fraction of shared information (0.54), reflecting their common underlying fetal cardiac process (Sulas et al., 2021). Other pairs, such as the maternal ECG (mECG) and respiration (mR), also showed the expected synchronization overlap (0.47) (Yasuma & Hayano, 2004). Conversely, the low overlap between fPWD and the maternal modalities (mECG/mR) was also consistent with expectations. Results from a three-modality version (fECG, mR, and fPWD) again highlighted the strongest overlap between fECG and fPWD (Supplementary Table 12), suggesting the feasibility of applying FiGuRO to settings with more than two modalities.

# D. Supplementary Tables

*Table 1.* **FiGuRO Hyperparameter sensitivity analysis.** We report the mean estimated rank and mean deviation from the ground truth (GT) ID ($\pm$ SEM) across all simulated datasets and random seeds for the given number of samples (N). Bold parameter values indicate chosen hyperparameters, bold ranks show best results based on rank and deviation (if applicable).

| Hyperparameter | Value | Estimated Rank | Deviation from GT | Sample size (N) |
|---|---|---|---|---|
| Distortion threshold ($\lambda$) | 0.005 | $6.56 \pm 0.04$ | $1.65 \pm 0.04$ | 180 |
| | 0.01 | $6.16 \pm 0.04$ | $1.29 \pm 0.04$ | |
| | **0.05** | $\mathbf{5.02 \pm 0.01}$ | $\mathbf{0.50 \pm 0.01}$ | |
| | 0.1 | $4.45 \pm 0.01$ | $0.68 \pm 0.01$ | |
| | 0.15 | $3.94 \pm 0.00$ | $1.06 \pm 0.00$ | |
| | 0.2 | $3.57 \pm 0.01$ | $1.43 \pm 0.01$ | |
| Frequency ($\tau$) | 5 | $5.24 \pm 0.02$ | $1.17 \pm 0.02$ | 360 |
| | **10** | $\mathbf{4.81 \pm 0.02}$ | $\mathbf{1.02 \pm 0.02}$ | |
| | 20 | $4.48 \pm 0.02$ | $1.11 \pm 0.02$ | |
| Energy threshold $\gamma$ | 0.0001 | $\mathbf{4.99 \pm 0.02}$ | $1.29 \pm 0.02$ | 270 |
| | 0.001 | $5.11 \pm 0.03$ | $1.16 \pm 0.02$ | |
| | **0.01** | $4.87 \pm 0.03$ | $1.09 \pm 0.03$ | |
| | 0.1 | $4.57 \pm 0.02$ | $\mathbf{0.85 \pm 0.01}$ | |
| Patience ($\pi$) | 5 | $5.30 \pm 0.04$ | $1.23 \pm 0.04$ | 216 |
| | **10** | $4.73 \pm 0.02$ | $\mathbf{0.88 \pm 0.02}$ | |
| | 20 | $5.06 \pm 0.03$ | $1.08 \pm 0.03$ | |
| | 50 | $\mathbf{5.03 \pm 0.03}$ | $1.30 \pm 0.03$ | |
| | 100 | $4.62 \pm 0.01$ | $1.07 \pm 0.01$ | |

---

[1]Two out of three random seeds resulted in posterior collapse and ID estimates of 1 for $\mathbf{C_2}$. For $\mathbf{C_1}$, KL weights $\beta \leq 0.0001$ resulted in posterior collapse and an ID estimate of 1.

*Table 2.* **Unimodal ID estimation benchmark.** We compare FiGuRO to a range of statistical estimators (Global, Local) and other neural network-based (NN) methods. nn refers to nearest neighbor. Ranges indicate estimates for varying hyperparameters (if applicable, in brackets next to the method). The respective hyperparameters and ranges tested for each method are described in B.8. Neural network-based methods are reported as means from three random seeds with SEM. We round the best estimate for each method and indicate best overall estimates compared to the ground truth ID with bold numbers, second best underlined. Note: The overestimation observed for FLIPD illustrates their sensitivity to architectural capacity, a limitation of diffusion-based LID estimators. Standard MLPs can underfit complex manifolds and bias estimates toward the ambient dimension (Yeats et al., 2025).

| Category | Method | $C_1$ (ID = 11) | $C_2$ (ID = 12) | Best ($C_1$,$C_2$) | Wall time (min) |
|---|---|---|---|---|---|
| Global | SVD ($CE$) | 1522 - 9424 | 1 - 104 | 1522, 1 ($CE = 80\%$) | $21.8 \pm 0.4$ |
| | PCA ($\sigma^2$) | 4040 - 9283 | 12 - 29 | 4040, 12 ($\sigma^2 = 0.8$) | $15.4 \pm 0.2$ |
| | BEMA ($\%ile$) | 570 - 948 | 273 - 1322 | 570, 273 ($\%ile = 99$) | $113.9 \pm 15.5$ |
| Local, proj. | lPCA ($\alpha$) | 4-8862 | 2-86 | **10, 9** ($\alpha = 0.05$) | $40.5 \pm 2.0$ |
| | FisherS | 3.4 | 1.8 | 3, 2 | $29.6 \pm 0.4$ |
| Local, geom. | CorrInt ($k_1', k_2'$) | 5.2 - 5.6 | 2.9 - 3.3 | 6, 3 ($k_1' = 2, k_2' = 5$) | $3.0 \pm 0.0$ |
| | TLE | 44 | 2.9 | 44, 3 | $3.2 \pm 0.3$ |
| Local, nn | MindML ($k'$) | 68 - 172 | 2.6 - 3.0 | 68, 3 ($k' = 100$) | $1.8 \pm 0.0$ |
| | MLE | 53 | 2.4 | 53, 2 | $1.6 \pm 0.0$ |
| | TwoNN ($\frac{r_2}{r_1}$) | 130 - 229 | 3.1 - 3.3 | 130, 3 ($\frac{r_2}{r_1} = 0$) | $3.1 \pm 0.1$ |
| NN | Loss cliff ($\lambda'$) | $512.0 \pm 0.0$ - $728.0 \pm 0.0$ | $68.0 \pm 5.0$ - $258.7 \pm 188.2$ | $512.0 \pm 0.0, 68 \pm 5.0$ ($\lambda' = 0.8$) | $2282.3 \pm 14.5$ |
| | ARD-VAE ($\beta$)[1] | $1.0 \pm 0.0$ - $14.0 \pm 0.0$ | $1.0 \pm 0.0$ - $93.3 \pm 92.3$ | $14.0 \pm 0.6$ , $7.3 \pm 6.3$ ($\beta = 0.001$) | $127.7 \pm 23.0$ |
| | RRA ($\gamma$) | $5.3 \pm 0.2$ - $146 \pm 0.6$ | $1.3 \pm 0.3$ - $138 \pm 6.0$ | $14.0 \pm 1.0, 8.7 \pm 0.7$ ($\gamma = 0.05$) | $42.8 \pm 6.2$ |
| | FLIPD ($t_0$) | $9.2 \pm 0.4$ - $998.1 \pm 0.2$ | $16.0 \pm 0.5$ - $828.8 \pm 1.2$ | $9.2 \pm 0.4, 16.0 \pm 0.5$ (manual, $t_0 = 0.7$) $130.5 \pm 0.6$, $174.3 \pm 3.1$ (knee: $t_0 = 0.4$, $t_0 = 0.15$) | $27.9 \pm 2.2$ |
| Ours | FiGuRO ($\lambda$) | **$10.4 \pm 0.2$ - $20.0 \pm 0.6$** | **$4.3 \pm 0.0$ - $11.7 \pm 0.9$** | $15.3 \pm 2.2, 5.7 \pm 0.3$ ($\lambda = 0.05$) | $153.5 \pm 3.3$ |

*Table 3.* **Effective ranks vs. number of generative variables.** We report the number of generative variables used to create each subspace GT matrix in datasets **B**, as well as the matrices' effective ranks (entropy-based effective dimensionality of the spectrum with a threshold of $10^{-12}$ for non-zero elements) computed from 10000 samples.

| | $B_s$ | $B_{i1}$ | $B_{i2}$ | $B_l$ |
|---|---|---|---|---|
| Number of generative variables | 2, 3, 5 | 20, 2, 2 | 2, 2, 20 | 20, 20, 20 |
| Effective rank of the matrix | 2.00, 2.59, 1.64 | 6.61, 1.94, 1.16 | 2.00, 1.98, 8.20 | 1.63, 18.55, 10.23 |

*Table 4.* **ID estimation with $L_1$ regularization.** Comparison of the average estimated ranks ($k_s, k_1, k_2$) when adding varying weights of $L_1$ regularization to the latent spaces. We report only the mean for three random seeds.

| L1 weight | $B_s$ | $B_{i1}$ | $B_{i2}$ | $B_l$ |
|---|---|---|---|---|
| 0.00 | 3.6, 1.0, 6.2 | 13.4, 4.4, 4.0 | 7.0, 1.0, 19.4 | 19.2, 12.8, 15.2 |
| 0.01 | 1.0, 1.0, 2.3 | 9.7, 6.3, 3.3 | 1.0, 1.0, 6.0 | 14.0, 9.0, 14.3 |
| 0.10 | 1.0, 1.0, 2.0 | 10.3, 5.7, 4.3 | 1.0, 1.0, 17.7 | 14.3, 8.7, 15.3 |
| 1.00 | 1.0, 1.0, 2.0 | 10.0, 6.0, 3.0 | 1.0, 1.0, 9.7 | 14.0, 9.0, 15.0 |
| 10.00 | 1.0, 1.0, 2.7 | 10.3, 6.3, 2.0 | 1.0, 1.0, 10.0 | 14.0, 10.7, 14.3 |

*Table 5.* **Disentanglement with $L_1$ regularization.** We report mean classification accuracy of label 0 accuracy from the shared subspace ($Acc_s$) and predictability ($R^2$) of label 1 and 2 from their respective subspaces in format ($Acc_s$, $R_1^2$, $R_2^2$) per $L_1$ weight and dataset (N=3).

| L1 weight | $\mathbf{B_s}$ | $\mathbf{B_{i1}}$ | $\mathbf{B_{i2}}$ | $\mathbf{B_l}$ |
|---|---|---|---|---|
| 0.00 | 1.00, 0.29, 0.94 | 0.98, 0.99, 1.00 | 1.00, 0.61, 0.98 | 0.99, 0.99, 0.99 |
| 0.01 | 0.74, 0.04, 0.92 | 0.94, 0.92, 0.97 | 0.75, 0.50, 0.77 | 0.91, 0.45, 0.82 |
| 0.10 | 0.75, 0.33, 0.96 | 0.94, 0.91, 0.98 | 0.83, 0.20, 0.83 | 0.90, 0.31, 0.86 |
| 1.00 | 0.82, 0.34, 0.94 | 0.94, 0.93, 0.97 | 0.79, 0.50, 0.96 | 0.90, 0.35, 0.84 |
| 10.00 | 0.88, 0.41, 0.94 | 0.94, 0.92, 0.97 | 0.73, 0.50, 0.64 | 0.91, 0.65, 0.85 |

*Table 6.* **ID estimation with orthogonal loss (Frobenius norm).** Comparison of the average estimated ranks ($\mathbf{k_s}$, $\mathbf{k_1}$, $\mathbf{k_2}$) when adding varying weights to the latent spaces. We report only the mean for three random seeds.

| Orthogonal weight | $\mathbf{B_s}$ | $\mathbf{B_{i1}}$ | $\mathbf{B_{i2}}$ | $\mathbf{B_l}$ |
|---|---|---|---|---|
| 0.00 | 3.6, 1.0, 6.2 | 13.4, 4.4, 4.0 | 7.0, 1.0, 19.4 | 19.2, 12.8, 15.2 |
| 0.01 | 1.0, 1.0, 2.3 | 10.3, 6.3, 2.0 | 1.0, 1.0, 16.3 | 15.0, 8.3, 16.0 |
| 0.10 | 1.0, 1.0, 2.0 | 10.3, 6.3, 2.3 | 1.0, 1.0, 12.7 | 13.7, 7.7, 15.7 |
| 1.00 | 1.3, 1.7, 3.3 | 14.0, 2.0, 3.3 | 1.0, 1.7, 11.0 | 13.7, 9.0, 18.0 |
| 10.00 | 1.3, 1.7, 5.7 | 14.0, 5.3, 5.3 | 1.0, 1.7, 14.3 | 15.0, 12.7, 30.7 |

*Table 7.* **Disentanglement with orthogonal loss (Frobenius norm).** We report mean classification accuracy of label 0 accuracy from the shared subspace ($Acc_s$) and predictability ($R^2$) of label 1 and 2 from their respective subspaces in format ($Acc_s$, $R_1^2$, $R_2^2$) per weight and dataset (N=3).

| Orthogonal weight | $\mathbf{B_s}$ | $\mathbf{B_{i1}}$ | $\mathbf{B_{i2}}$ | $\mathbf{B_l}$ |
|---|---|---|---|---|
| 0.00 | 1.00, 0.29, 0.94 | 0.98, 0.99, 1.00 | 1.00, 0.61, 0.98 | 0.99, 0.99, 0.99 |
| 0.01 | 0.74, 0.01, 0.35 | 0.94, 0.91, 0.96 | 1.00, 0.08, 0.83 | 0.90, 0.18, 0.86 |
| 0.10 | 0.84, 0.08, 0.49 | 0.94, 0.95, 0.95 | 0.77, 0.05, 0.84 | 0.89, 0.22, 0.86 |
| 1.00 | 0.50, 0.15, 0.58 | 0.94, 0.63, 0.97 | 0.25, 0.03, 0.98 | 0.90, 0.47, 0.78 |
| 10.00 | 0.47, 0.07, 0.89 | 0.94, 0.83, 0.98 | 0.44, 0.34, 0.94 | 0.89, 0.14, 0.81 |

*Table 8.* **Edge case: Multi-modal data without shared information. $L_2$ regularization enforces disentanglement.** This table shows the effect of increasing the $L_2$ weight on the decoder's first layer when trained on a simulated dataset with **no true shared ID** ($k_{s,GT} = 0$) and modality-specific IDs of 20 (a modification of $\mathbf{B}_l$). A lower shared rank $k_s$ is better. $k_{tot}$ refers to the total rank (sum of all subspace ranks). Predictability of each modality's label is given by the goodness of fit $R^2$. This was done for three random seeds, showing the mean and SEM.

| L2 weight | $k_s \downarrow$ | $k_{tot}$ | $k_s/k_{tot} \downarrow$ | $R^2$ label 1 $\uparrow$ | $R^2$ label 2 $\uparrow$ |
|---|---|---|---|---|---|
| 0.00 | $9.7 \pm 0.3$ | $23.7 \pm 0.3$ | 0.40 | $0.40 \pm 0.14$ | $0.99 \pm 0.00$ |
| 0.01 | $4.3 \pm 1.9$ | $23.7 \pm 0.7$ | 0.18 | $0.69 \pm 0.20$ | $0.98 \pm 0.01$ |
| 0.10 | $7.0 \pm 5.0$ | $31.5 \pm 8.5$ | 0.22 | $0.74 \pm 0.25$ | $0.99 \pm 0.00$ |
| 1.00 | $2.7 \pm 0.7$ | $26.7 \pm 9.2$ | 0.10 | $0.99 \pm 0.01$ | $0.98 \pm 0.01$ |
| 10.00 | $1.0 \pm 0.0$ | $12.0 \pm 1.7$ | 0.08 | $0.97 \pm 0.01$ | $0.98 \pm 0.01$ |

*Table 9.* **Wall times in minutes for multi-modal baselines and FiGuRO.** We report wall times in minutes for runs with seed 42 (unless stated otherwise) on a system with RTX A5000 and AMD Ryzen Threadripper PRO 3975WX.

| Data | JIVE | AJIVE | SLIDE | ShIndICA | FiGuRO (ours) |
|---|---|---|---|---|---|
| **B** (all 4 sets, 3 runs) | $0.11 \pm 0.04$ | $0.02 \pm 0.00$ | $0.03 \pm 0.00$ | $2.25 \pm 0.36$ | $8.65 \pm 0.60$ |
| Audio MNIST (from embeddings) | 1.70 | 0.68 | 0.25 | 8.54 | 16.79 |
| Audio MNIST (from raw | 299.98 | 19.00 | 14.43 | 507.83 | 60.49 |

*Table 10.* **Comparison of predictability performance per label and subspace across multi-modal decomposition methods for simulated datasets B.** Columns 4-7 present baseline methods from multi-view data decomposition. Our method (FiGuRO) is listed at the end. Predictability performance is reported in triplets as performance on shared, private 1, and private 2 subspace. Rows per dataset indicate the label that was predicted and the metric that was used for evaluation ($Acc$: classification accuracy, $R^2$: Goodness of fit). Highest values for the corresponding space and label are highlighted in bold, second best underlined (if applicable). For baseline methods, we report the average of joint_X and joint_Y for the shared performance on the shared label, and the corresponding joint values for the specific labels. For FiGuRO, we present the mean predictability metrics (ranges shown in Supplementary Figures 4-7).

| Data | Metric | Label | JIVE | AJIVE | SLIDE | ShIndICA | FiGuRO (ours) |
|---|---|---|---|---|---|---|---|
| $\mathbf{B}_s$ | $Acc$ | Shared | 0.93, 1.00, 1.00 | 0.24, 1.00, 1.00 | 0.71, 1.00, 1.00 | 0.95, 0.59, 0.59 | **1.00**, 0.28, 0.38 |
| | $R^2$ | Modality 1 | 1.00, 0.00, 0.00 | 1.00, 0.00, 0.00 | 1.00, 0.00, 0.00 | 0.67, 0.02, 0.02 | 0.66, **0.34**, 0.00 |
| | $R^2$ | Modality 2 | 1.00, 0.00, 0.00 | 1.00, 0.00, 0.00 | 1.00, 0.00, 0.00 | 0.98, 0.00, 0.00 | 0.38, 0.00, **0.93** |
| $\mathbf{B}_{i1}$ | $Acc$ | Shared | **0.96**, 0.24, 0.24 | 0.20, 0.96, 0.96 | 0.51, 0.80, 0.80 | 0.93, 0.28, 0.28 | 0.94, 0.27, 0.34 |
| | $R^2$ | Modality 1 | 1.00, 0.00, 0.00 | 1.00, 0.00, 0.00 | 1.00, 0.00, 0.00 | 0.84, 0.01, 0.01 | 0.00, **0.70**, 0.00 |
| | $R^2$ | Modality 2 | 1.00, 0.00, 0.00 | 1.00, 0.00, 0.00 | 0.97, 0.00, 0.00 | 1.00, 0.00, 0.00 | 0.00, 0.00, **0.97** |
| $\mathbf{B}_{i2}$ | $Acc$ | Shared | **1.00**, 1.00, 1.00 | 0.23, 1.00, 1.00 | 0.56, 1.00, 1.00 | 0.95, 0.52, 0.51 | **1.00**, 0.36, 0.25 |
| | $R^2$ | Modality 1 | 1.00, 0.25, 0.00 | 1.00, 0.00, 0.00 | 1.00, 0.00, 0.00 | 0.77, 0.02, 0.02 | 0.56, **0.61**, 0.00 |
| | $R^2$ | Modality 2 | 1.00, 0.00, 0.00 | 1.00, 0.00, 0.00 | 1.00, 0.00, 0.00 | 0.99, 0.00, 0.00 | 0.75, 0.00, **0.72** |
| $\mathbf{B}_l$ | $Acc$ | Shared | **0.93**, 0.25, 0.25 | 0.19, 0.94, 0.94 | 0.49, 0.98, 0.98 | 0.69, 0.60, 0.60 | 0.92, 0.23, 0.55 |
| | $R^2$ | Modality 1 | 1.00, 0.00, 0.00 | 1.00, 0.00, 0.00 | 92, 0.00, 0.00 | -0.03, 0.02, 0.02 | 0.33, **0.64**, 0.00 |
| | $R^2$ | Modality 2 | 1.00, 0.00, 0.00 | 1.00, 0.00, 0.00 | 0.85, 0.00, 0.00 | 0.86, 0.03, 0.03 | 0.00, 0.00, **0.92** |

*Table 11.* **Information overlap between pairs of NInFEA modalities measured as the ratio of shared over total rank.** The total rank is calculated as the sum of all subspace ranks. "*" denotes a modality pair we initially expected to have a small overlap but showed strong technical bias from one modality onto the other.

| | fECG-mECG | fECG-fPWD | mECG-mR | fECG-mR | mECG-fPWD | mR-fPWD |
|---|---|---|---|---|---|---|
| $k_s$ / $k_{tot}$ | **0.38** | **0.54** | **0.47** | 0.37* | 0.17 | 0.13 |

*Table 12.* **Subspace IDs for FiGuRO with three modalities.** Trained on NInFEA.

| Subspace | ID |
|---|---|
| global shared | 3 |
| fECG + respiration | 3 |
| fECG + PWD | 8 |
| respiration + PWD | 5 |
| fECG | 1 |
| respiration | 1 |
| PWD | 1 |

*Table 13*. **Reconstruction performance before and after rank optimization.** $\mathcal{L}$ indicates the loss (MSE), with $\Delta$ indicating the difference in loss from initial ($t = 0$) to final rank ($\mathcal{L}^t - \mathcal{L}^0$). We report mean and SEM over three random seeds. % indicates the relative increase in loss. For final ranks we only show mean, for losses we show mean +- SEM. $k_0$ refers to the initial sum of max ranks we set for each subspaces, $k_{total}$ to the mean sum of all final ranks over the random seeds.

| Dataset | $k_0$ | $k_{total}$ | $\mathcal{L}_{test}$ | $\Delta\mathcal{L}_{test}$ | $\%_{test}$ |
|---|---|---|---|---|---|
| Audio MNIST $(i,a)$ | 600 | 16.4 | $0.3153 \pm 0.0013$ | $0.0007 \pm 0.0003$ | 0.2% |
| So2Sat $(r,i)$ | 600 | 85.7 | $0.3144 \pm 0.0056$ | $0.0470 \pm 0.0038$ | 1.2% |
| NYU Depth V2 $(i, d)$ | 1500 | 303.4 | $0.0213 \pm 0.0001$ | $0.0004 \pm 0.0001$ | 1.9% |

*Table 14*. **ID estimation on real multi-modal datasets.** We report ID estimates for shared ($k_s$) and modality-specific subspaces on different datasets and methods. Our NN-based method was evaluated on 3 random seeds, reporting the mean $\pm$ SEM. We excluded CCA and DIVAS as they do not estimate individual subspaces, and PPD because it failed to return predictions on Audio MNIST after 2 days of running. For our method, we used a distortion budget of $\lambda = 0.05$.

| Dataset | Subspace | JIVE | AJIVE | SLIDE | ShIndICA | **Ours** |
|---|---|---|---|---|---|---|
| Audio MNIST $(i,a)$ | $k_s$ | 1 | 9 | 1 | 1 | $9.7 \pm 0.3$ |
| | $k_1$ | 133 | 135 | 117 | 117 | $1.7 \pm 0.3$ |
| | $k_2$ | 230 | 485 | 234 | 234 | $5.0 \pm 0.6$ |
| So2Sat $(r,i)$ | $k_s$ | 1 | 221 | 1 | 5 | $32.0 \pm 0.6$ |
| | $k_1$ | 1619 | 1402 | 1618 | 1614 | $1.0 \pm 0.0$ |
| | $k_2$ | 1117 | 999 | 1116 | 1112 | $52.7 \pm 1.9$ |
| NYU Depth V2 $(i, d)$ | $k_s$ | 1 | 27 | 1 | 10 | $149.7 \pm 0.7$ |
| | $k_1$ | 379 | 353 | 403 | 394 | $143.7 \pm 1.3$ |
| | $k_2$ | 472 | 445 | 530 | 510 | $10.0 \pm 0.0$ |

*Table 15*. **AudioMNIST Sweep: Mean $\pm$ SEM rank deviations from base configuration.** This table reports the average deviations and standard error of the mean (SEM) over $n = 3$ seeds across the shared and modality-specific (mod1, mod2) subspaces for various sweep parameters. We highlight the base configuration in **bold**.

| Parameter | Value | Shared | Modality 1 | Modality 2 |
|---|---|---|---|---|
| distortion metric $D$ | **ExVarScore** | $0.0 \pm 0.0$ | $0.0 \pm 0.0$ | $0.0 \pm 0.0$ |
| | MSE | $6.0 \pm 0.6$ | $0.0 \pm 0.0$ | $6.0 \pm 0.6$ |
| | R2 | $3.3 \pm 0.7$ | $-2.0 \pm 0.0$ | $3.3 \pm 0.7$ |
| | RMSE | $4.3 \pm 0.3$ | $0.0 \pm 0.0$ | $4.3 \pm 0.3$ |
| patience $\pi$ | 5 | $2.0 \pm 0.0$ | $0.0 \pm 0.0$ | $2.0 \pm 0.0$ |
| | **10** | $0.0 \pm 0.0$ | $0.0 \pm 0.0$ | $0.0 \pm 0.0$ |
| | 20 | $0.0 \pm 0.0$ | $0.0 \pm 0.0$ | $0.0 \pm 0.0$ |
| distortion budget $\lambda$ | 0.01 | $4.3 \pm 0.3$ | $0.0 \pm 0.0$ | $4.3 \pm 0.3$ |
| | **0.05** | $0.0 \pm 0.0$ | $0.0 \pm 0.0$ | $0.0 \pm 0.0$ |
| | 0.1 | $0.0 \pm 0.0$ | $0.0 \pm 0.0$ | $0.0 \pm 0.0$ |
| reduction frequency $\tau$ | 2 | $4.0 \pm 0.6$ | $0.0 \pm 0.0$ | $4.0 \pm 0.6$ |
| | 5 | $2.0 \pm 0.0$ | $0.0 \pm 0.0$ | $2.0 \pm 0.0$ |
| | **10** | $0.0 \pm 0.0$ | $0.0 \pm 0.0$ | $0.0 \pm 0.0$ |
| | 20 | $0.0 \pm 0.0$ | $0.0 \pm 0.0$ | $0.0 \pm 0.0$ |
| energy threshold $\gamma$ | 0.001 | $16.0 \pm 0.0$ | $0.0 \pm 0.0$ | $0.0 \pm 0.0$ |
| | **0.01** | $0.0 \pm 0.0$ | $0.0 \pm 0.0$ | $0.0 \pm 0.0$ |
| | 0.1 | $-2.7 \pm 0.3$ | $0.3 \pm 0.3$ | $-1.0 \pm 0.0$ |

# E. Supplementary Figures

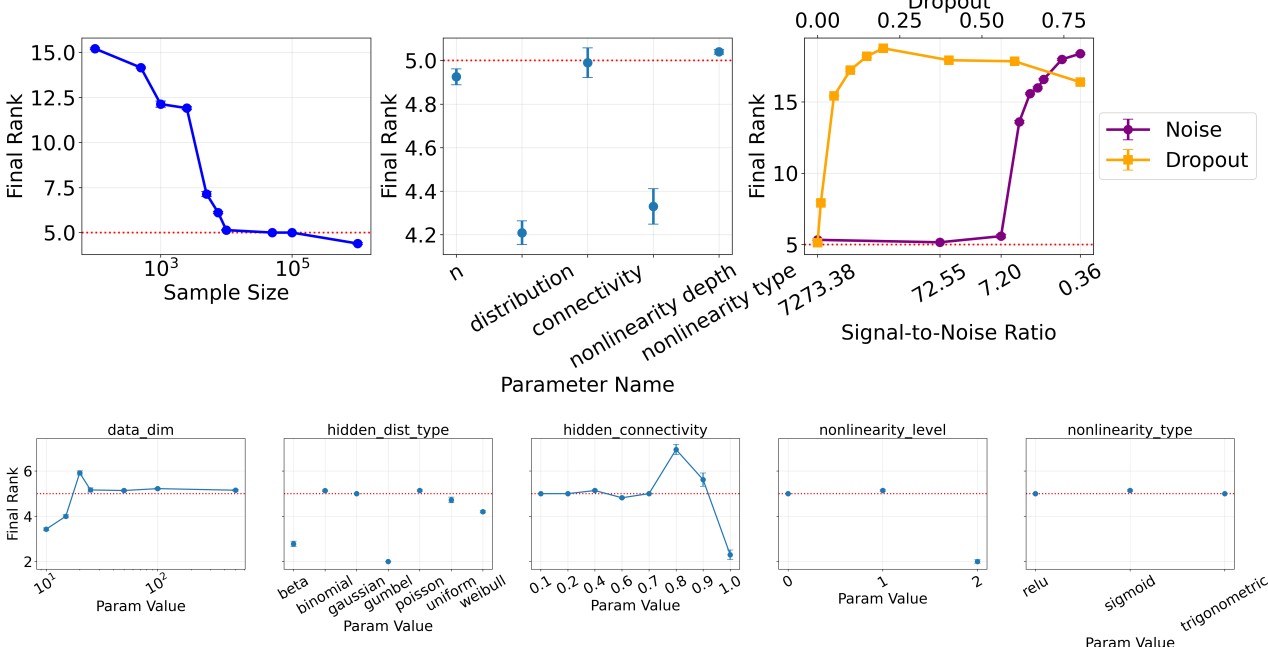

*Figure 1.* **Robustness tests.** We evaluated the method's stability on simulated data (ground truth ID of 5, red dotted lines) by systematically varying key generative parameters and running for three random seeds. **Top row:** The top left plot shows that the estimated rank converges to the true ID as sample size increases, stabilizing around N=5000. The middle plot displays the mean final rank across the different data characteristics we tested. The top right plot demonstrates robustness to noise up to a high signal-to-noise ratio (SNR) but shows overestimation with high levels of dropout. **Bottom Row** These plots provide a more detailed view of the top middle plot. Each subplot represents one of the data characteristics from the x axis of the top left plot. Error bars indicate the SEM from three random seeds.

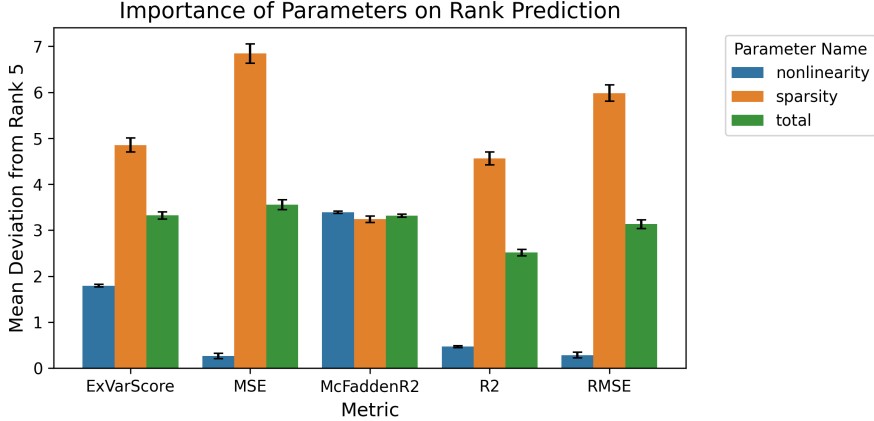

*Figure 2.* **Distortion metrics.** We evaluated five different distortion metrics by measuring the mean deviation of their ID estimates from the ground truth (ID=5) across simulated datasets with varying levels of nonlinearity and sparsity. We report the average deviation from the ground truth ID 5 with error bars indicating the SEM from three random seeds.

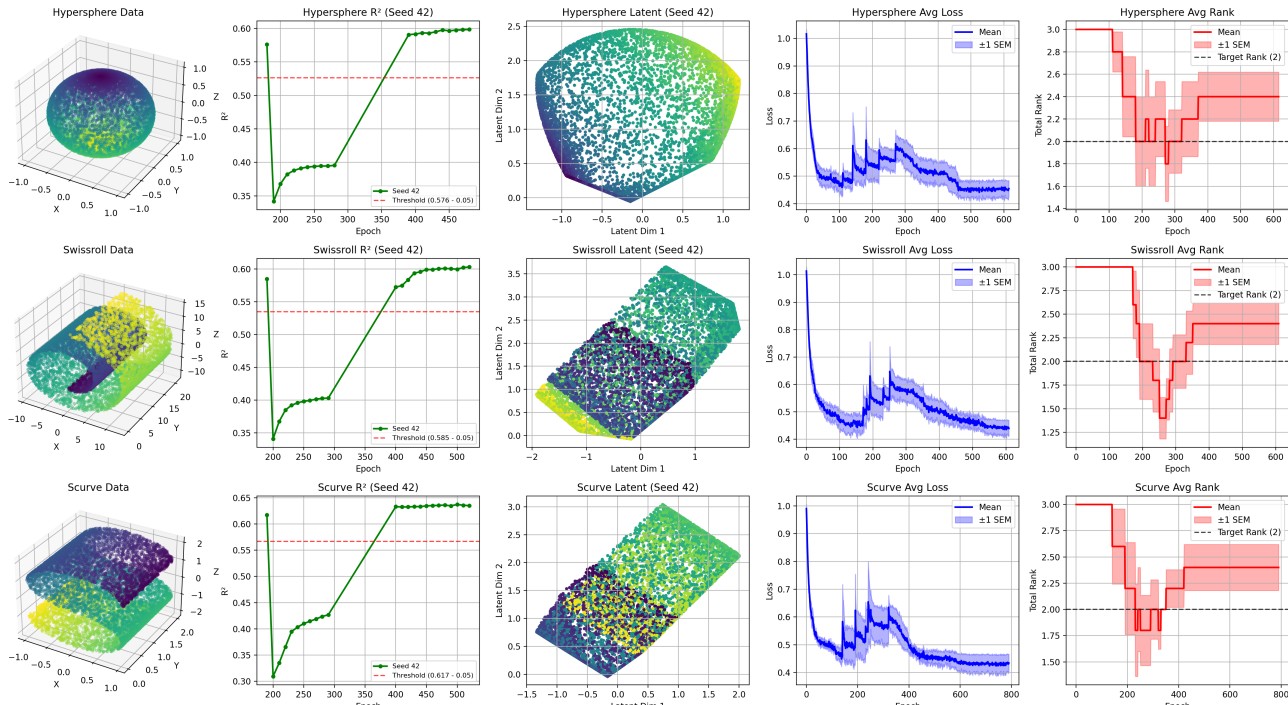

*Figure 3.* **Training dynamics on 3D Manifold datasets.** Each row corresponds to a different dataset: Hypersphere, Swiss Roll, and S-Curve. For each dataset, the columns show: the original 3D data, the R² metric during rank optimization (red dashed line showing the distortion budget), the learned 2D latent representation, the average training loss over 5 seeds, and the average rank convergence over 5 seeds. The target rank of 2 is indicated by the dashed grey line.

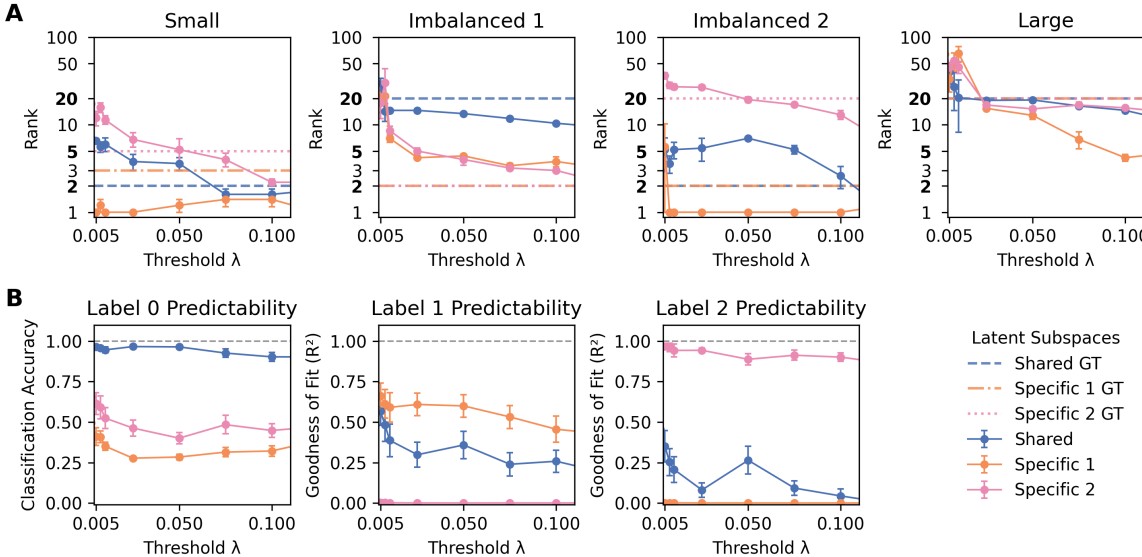

*Figure 4.* **Multi-modal ID estimation and disentanglement of B with varying true IDs.** The x axis presents the distortion threshold $\lambda$. **(A)** Log-scale estimated ranks (mean $\pm SEM$, $N = 5$ seeds) for shared (blue) and modality-specific (orange, pink) subspaces. Ground truth (GT) IDs are depicted as dashed lines. Values closer to the GT lines of the same color indicate better performance. **(B)** Average predictability of shared and private information over all random seeds and datasets ($N = 20$). The class of the shared space (label 0) is evaluated as classification accuracy. Labels 1 and 2 represent the mean value of modality-specific generative hidden vectors. Their predictability is evaluated with the $R^2$. Metrics are defined in B.4.

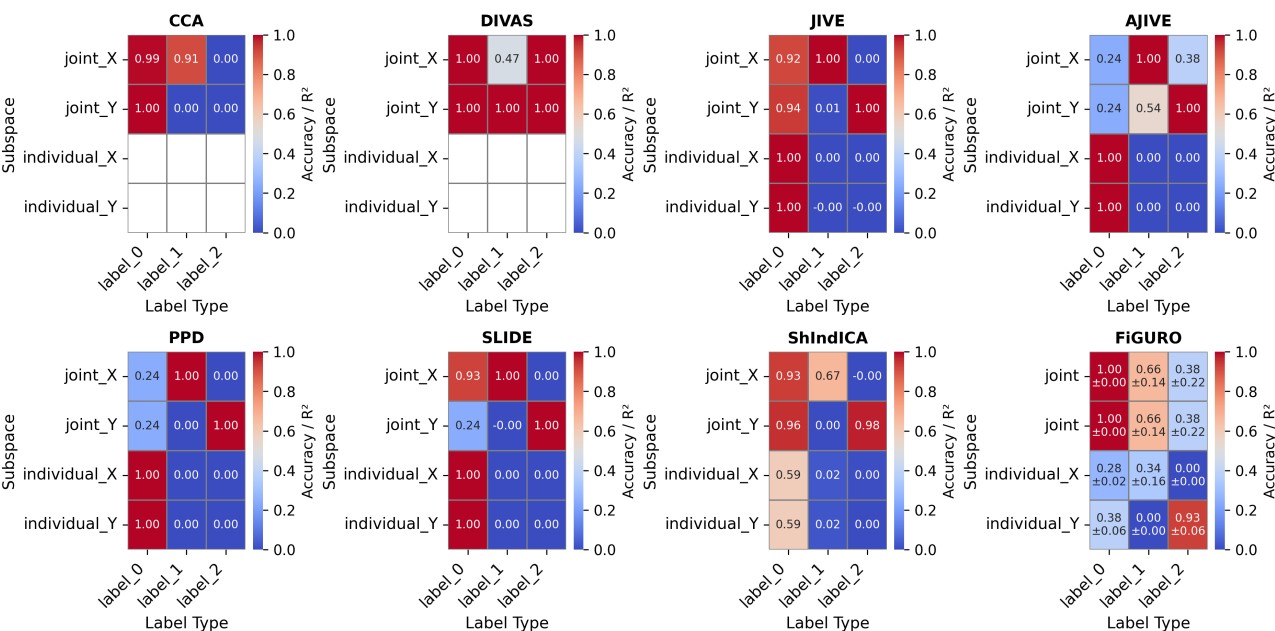

*Figure 5.* **Disentanglement evaluation of multi-modal baselines on dataset B_s.** Each heatmap plots the predictability of the three ground truth labels (0: shared, 1: modality 1, 2: modality 2) from the decomposed joint and individual (private) subspaces per method. Predictability is evaluated as in Figure 3B as classification accuracy and $R^2$. For our method FiGuRO, the joint predictabilities are duplicated as there is only one learned joint subspace, and values are reported as mean $\pm$ SEM from 5 random seeds. Empty cells for the first two methods in individual components indicate that these methods did not estimate individual subspaces.

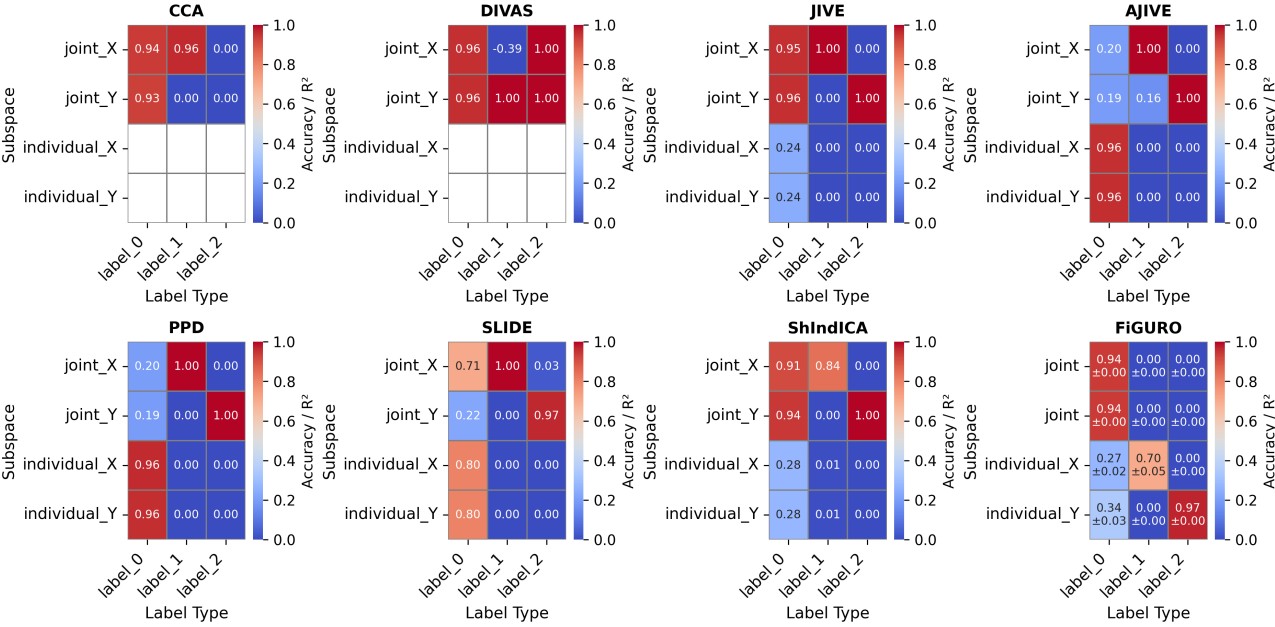

*Figure 6.* **Disentanglement evaluation of multi-modal baselines on dataset B_i1.** For details see caption 5.

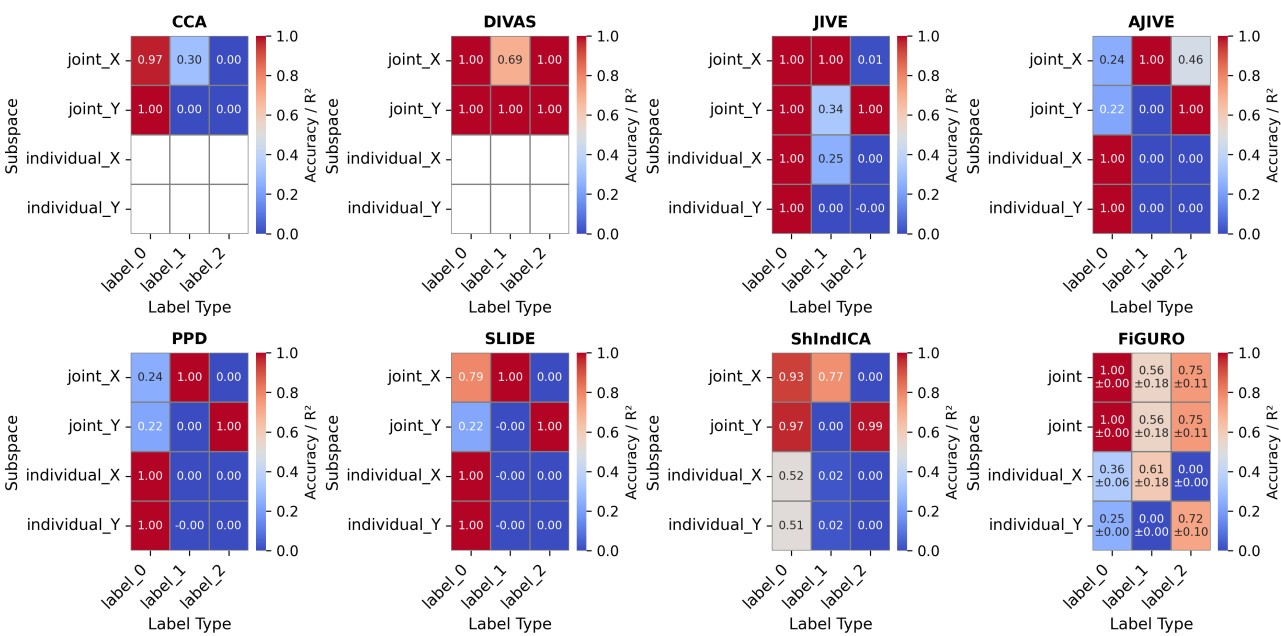

*Figure 7.* **Disentanglement evaluation of multi-modal baselines on dataset $B_{i2}$.** For details see caption 5.

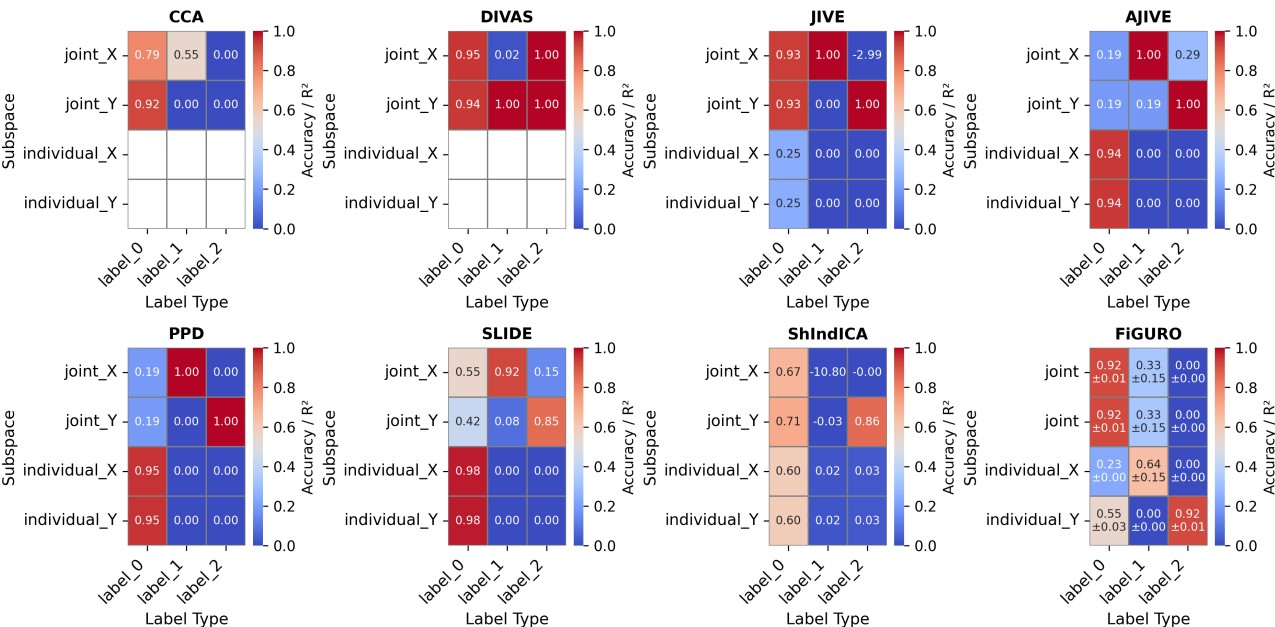

*Figure 8.* **Disentanglement evaluation of multi-modal baselines on dataset $B_1$.** For details see caption 5.

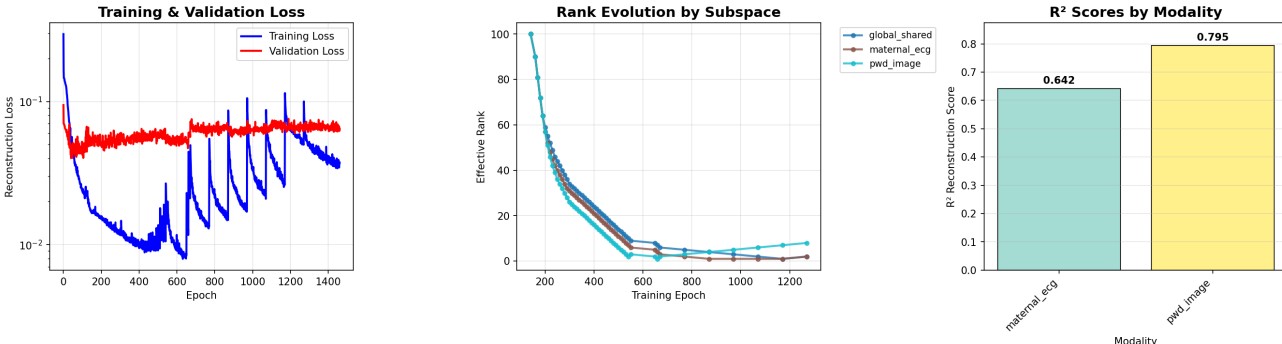

*Figure 9.* **Training dynamics on NInFEA mECG-fPWD.** The left plot shows the training and validation loss. The middle plot depicts the ranks of all three subspaces over epochs. The right plot shows the initial $R^2$ metrics per modality.

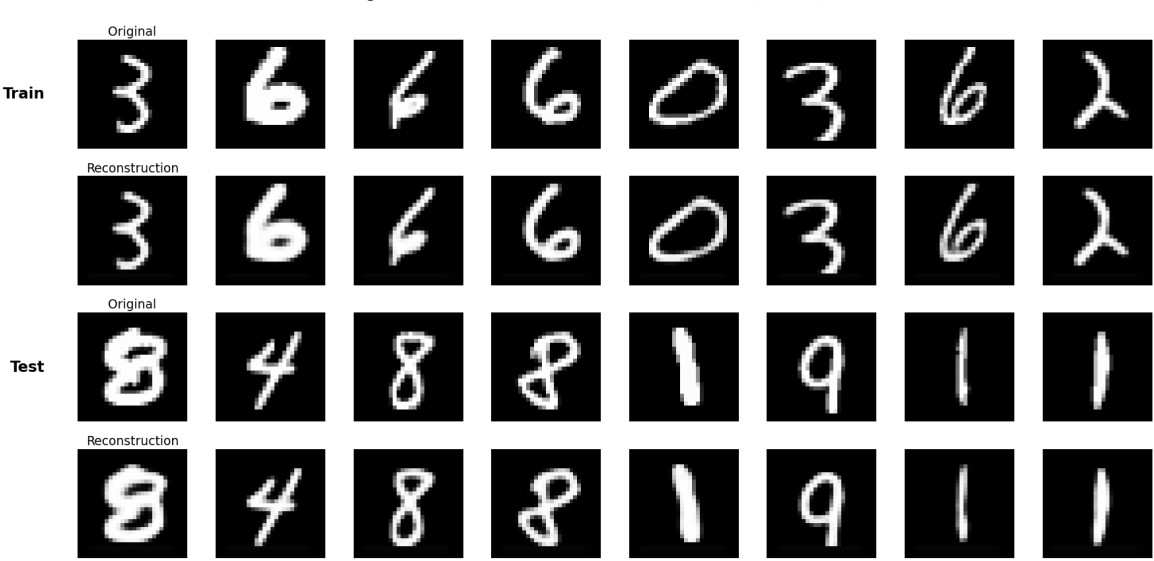

*Figure 10.* **Image reconstruction samples from Audio MNIST.** The top row presents original test samples, the bottom its reconstructions from our pretrained Audio MNIST model (seed 0).

Audio Autoencoder Reconstructions - Full Spectrum, Train & Test (Seed 0)

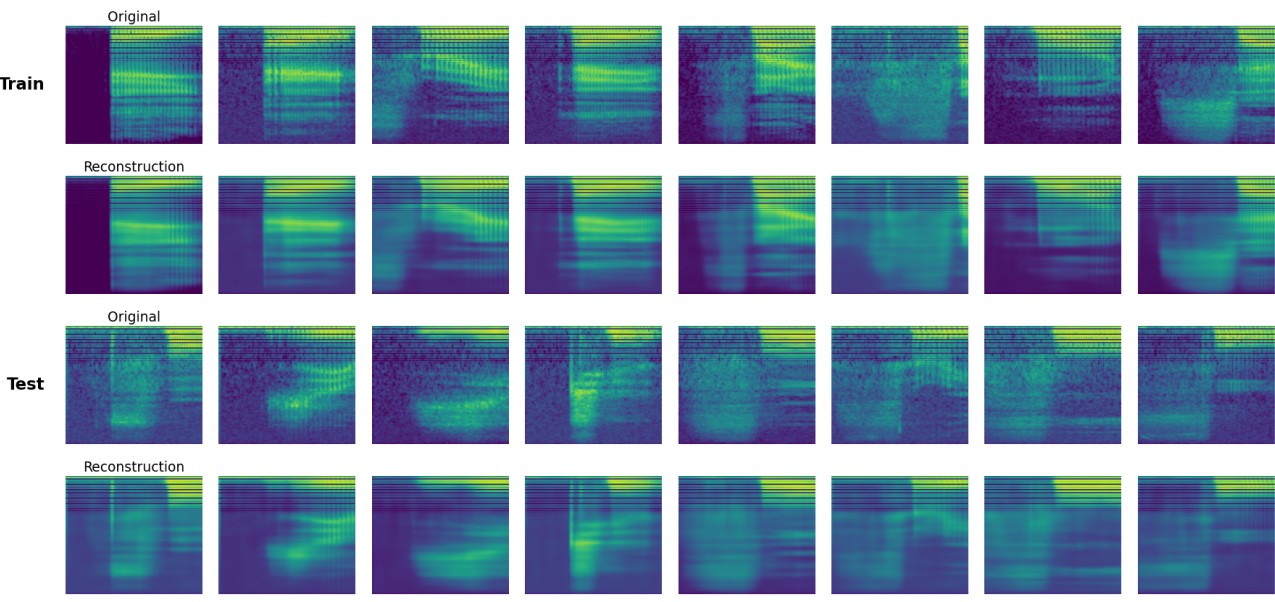

*Figure 11.* **Audio reconstruction samples from Audio MNIST.** The top row presents original test samples, the bottom its reconstructions from our pretrained Audio MNIST model (seed 0).

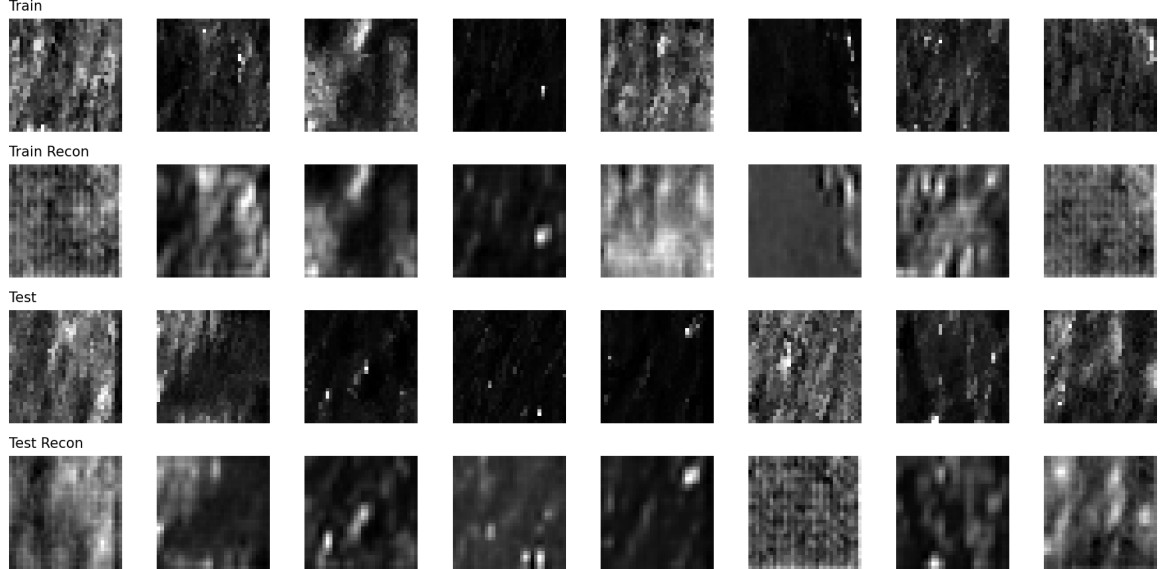

*Figure 12.* **Reconstruction of the SAR modality (channel 5).** Channel 5 contains the intensity of the refined Lee-filtered VH channel from Sentinel-1. Reconstructions are shown from the pretrained model (seed 0).

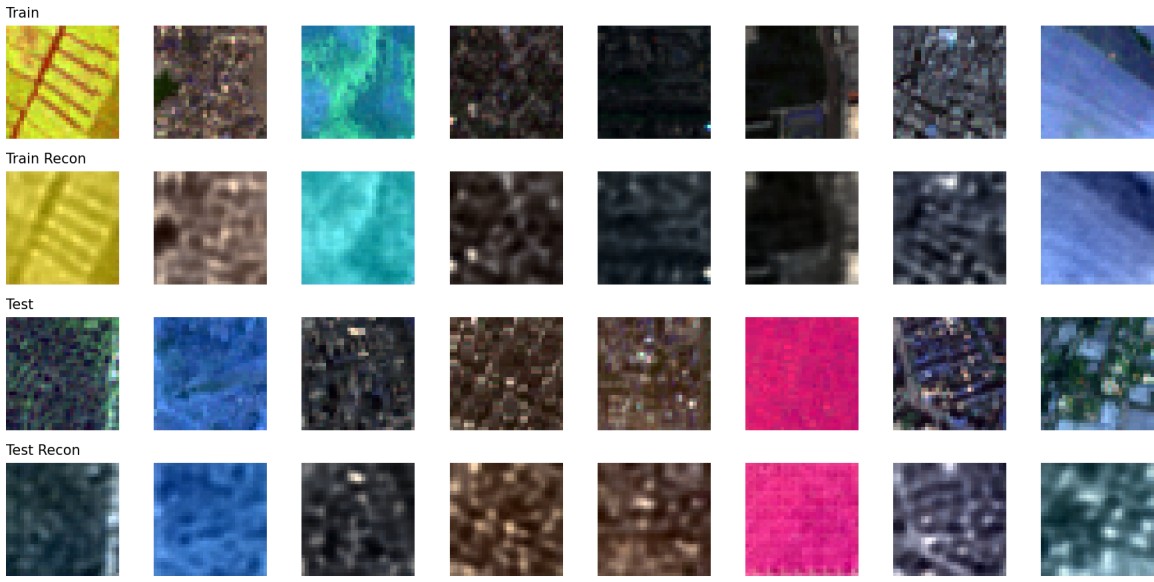

*Figure 13.* **Reconstruction of the RGB channels in the optical modality.** Reconstructions are shown from the pretrained model (seed 0).

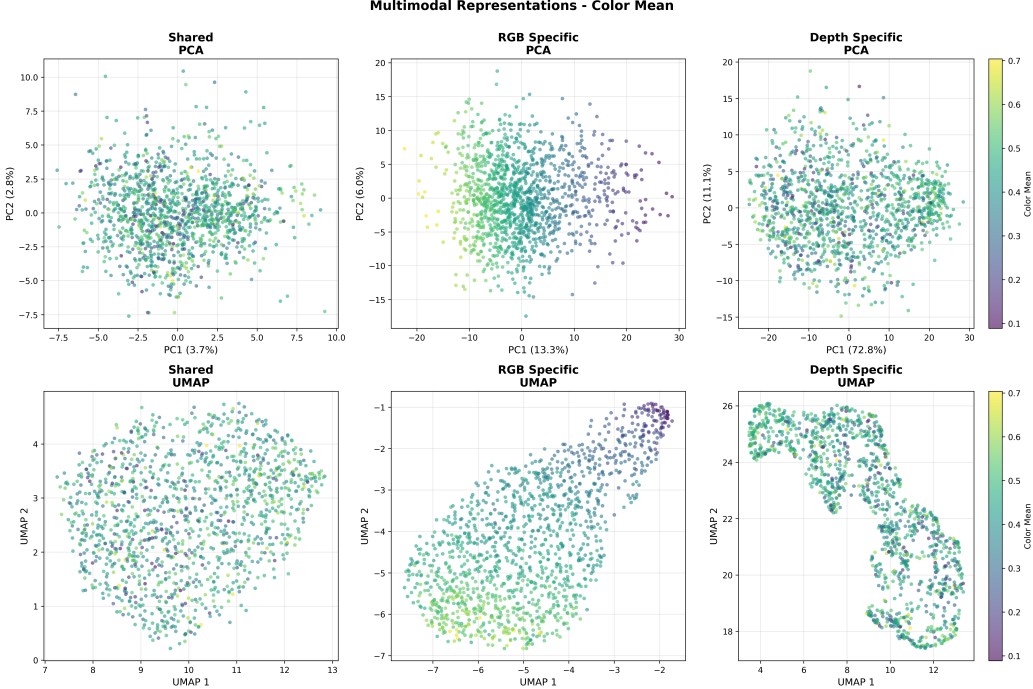

*Figure 14.* **Image color is decomposed into the image-specific representation.** Rows show PCA and UMAPs of each subspace (columns) from the train set. Color indicates the mean RGB values.

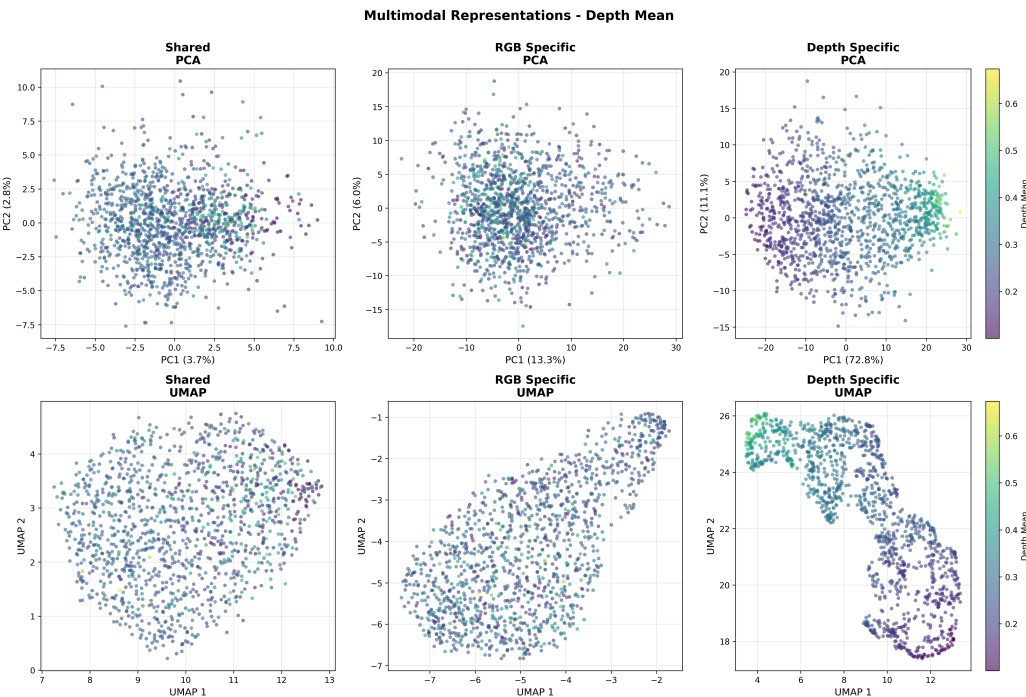

*Figure 15.* **Image depth is decomposed into the depth-specific representation.** Rows show PCA and UMAPs of each subspace (columns) from the train set. Color indicates the mean depth values.

