# OpenReview forum: "FiGuRO: Intrinsic Dimension Estimation for Multi-Modal Data"
_ICML.cc/2026/Conference — ICML 2026 regular_

### Official Review · Reviewer_ZRoj · 2026-03-06

**Soundness:** 3
**Presentation:** 3
**Significance:** 3
**Originality:** 3
**Overall Recommendation:** 5
**Confidence:** 3

**Summary:**

This paper introduces FiGuRO, a unified and adaptive framework for multimodal intrinsic dimensionality (ID) estimation. The goal is to train a single multimodal autoencoder that simultaneously estimates the modality-specific IDs and the shared ID across modalities. A key technical idea is to leverage the SVD of the projection (weight) matrix to obtain an effective-rank type criterion, which provides a direct way to infer ID and supports dynamic adjustment of the estimated dimensionalities during training. On simulation studies with known ground-truth IDs, FiGuRO yields more stable and more accurate ID estimates across modalities than strong competing methods. On real-world datasets without ground-truth IDs, the authors report good scalability and competitive downstream accuracy (often higher than unimodal baselines) with emerging disentanglement-like behavior, i.e. the model appears to capture modality-specific structure while effectively synthesizing shared information in a common subspace.

**Compliance With Llm Reviewing Policy:**

Affirmed.

**Final Justification:**

I appreciate the nuances and additional ablations the authors provided during the discussion period. All my main concerns have been addressed. I therefore raised Soundness and Presentation to 3, and my overall score to 5.

**Key Questions For Authors:**

- It would help to restructure the Related Work section so it reads more smoothly and is easier to navigate. Right now it feels a bit compressed. A clearer flow would be: state the shared goal of the methods, organize them into a few families and highlight key differences, then make explicit which gap remains and how the paper addresses it. It would also be worth spending a little more time on the baselines used in the main experiments (JIVE, AJIVE, SLIDE, ShIndICA): what each method is designed to capture, why it is an appropriate comparator here, and why these were selected instead of other reasonable alternatives.
- Please clarify what Acc_all refers to in Table 3.
- The experimental setup for competing methods should be stated more explicitly in the main paper. In particular, whether the default settings from the chosen implementations were used should be made clear. If not, the chosen settings should be specified and justified as fair for comparison to FiGuRO.
- Along the same lines, moving some additional ablations from the appendix to the main paper (Appendix D, Table 1) would strengthen the message. It would also help to place the sensitivity analysis next to the competing methods’ results, so it is clear that varying FiGuRO’s hyperparameters does not change the overall conclusions relative to the baselines. This would better support the claim that the gains are driven by the architectural design rather than a particular hyperparameter combination.
- The robustness analysis (Section 4.2) would be easier to follow if each robustness experiment linked to the specific appendix subsection where it is described. Right now, only Appendix B is referenced, which is long and difficult to navigate.
- The appendix contains a few standalone subsections that could be merged for readability. For example, there is a single “B.4.1 Distortion Metrics” under B.4 and a single “B.5.1 Classification Accuracy” under B.5; merging these into their parent sections would likely improve flow.

**Limitations:**

yes

**Strengths And Weaknesses:**

### Strengths
- While the ideas in the paper are not revolutionary per se (they mainly build on RRA and LoRA-style ingredients), the authors combine them into a relatively novel and meaningful unified framework that extends naturally to multiple modalities, without the tedious optimization challenges of probabilistic approaches or the expensive iterative search over latent dimensionality used in many existing methods.
- The work aligns with a current trend in machine learning toward multi-objective, multi-modal architectures. This direction is well-motivated in practice, since real-world data streams often come from heterogeneous sources. The central hypothesis is that learning from multiple sources jointly can enable more transferable knowledge and yield a more accurate multi-view of the underlying phenomenon, provided the modalities are not orthogonal to each other. Overall, the work pushes in a promising long-term direction.
- The experimental protocol is solid and exhaustive. The authors include a wide range of established and recent baselines (in both the main paper and the appendix), as well as extensive ablation studies covering the main hyperparameters of FiGuRO.

### Weaknesses
- (Presentation) The paper is doing several things at once, and the main thread sometimes gets buried in details. While the technical content seems solid, I found it harder than it should be to keep track of (i) what the core idea is, (ii) how it differs from the most relevant prior work, and (iii) which parts are essential versus “implementation choices.” A smoother high level narrative would make the contribution easier to absorb.
- (Significance) The paper would benefit from a clearer sense of “why these comparisons.” The experimental section reports strong results, but without enough context, it is hard to tell whether the gains come from the central design choice or from how methods are configured. This makes the contribution feel less grounded than it could be.
- (Significance) There is a lot of experimentation, but it does not yet translate into clear guidance for a reader who wants to use the method. I am left unsure about what settings are stable, what matters most, and how sensitive the approach is outside the curated benchmark setup.

---

> ### Author Rebuttal · Authors · 2026-03-30
>
> Thank you for your thorough review and positive feedback on technical solidity and exhaustive experimentation. We appreciate the feedback that our communication of the main contribution was at times not as clear as it could be. We implemented text edits to improve readability. Revised text is highlighted in *italic*. We use abbreviations “S” and “P” for section and paragraph, respectively.
>
> Weaknesses:
>
> 1. Presentation:
>     - i) The core idea: We edit S1P3 lines 56ff to improve clarity on our goal and contribution:
>         > ''Our approach estimates [...]. *We combined two powerful principles to create a method that approximates intrinsic dimensions of decoupled subspaces: Firstly, latent spaces are learned via low-rank decomposable layers using truncated Singular Value Decomposition* inspired by adaptive rank reduction (Mounayer et al., 2025) *and LoRA (Hu et al., 2021). Secondly, our algorithm to govern the rank optimization is based on Rate-Distortion Theory, using relative reconstruction fidelities of all modalities to decide when to increase or decrease subspace ranks. FiGuRO is the first method to achieve multi-modal ID estimation and information decoupling* in a single training pass."
>     - ii) Difference to prior work: In S2P1, we introduce advanced uni-modal ID estimation methods and discuss differences and shortcomings. In S3.1P3 and S3.2P1 we clarify the differences between ARR and our method. The focus of our work is on the multi-modal case, for which no explicit ID estimation techniques exist. We describe traditional multi-view decomposition methods and how they fall short in S2P1 (first two sentences).
>     - iii) Essential vs implementation choice: This is a great point. We add the following paragraph to S3.2 after equation 5:
>         > ''*The essential components of FiGuRO are the explicit subspace separation (in the multi-modal case), low-rank decomposable layers, and the bi-directional rank optimization logic described in Algorithm 1, including the use of a distortion metric $D$ and threshold $\lambda$. Using an interval with step size $\tau$ and patience $\pi$, the choice of distortion metric, and updating ranks based on cumulative energy threshold $\gamma$ are implementation choices.*"
>
> 2. Significance: Even though our focus is on multi-modal ID estimation, FiGuRO can also be applied to uni-modal data. As a result, our experimental order in S4 and S5 is the following: We introduce the datasets, evaluate our method's sensitivity to hyperparameters and data characteristics, and move on to both uni- and multi-modal ID estimation benchmarks in simulated data settings, as well as decoupling evaluation. Afterwards, to demonstrate downstream utility of the learned latent subspaces, we apply FiGuRO to real world datasets.
>
>     Gains: Our ablation study in Supplementary Table 3 and S5.2 demonstrates that gains stem from our algorithm and not just the extension of low-rank approximation to a multi-modal architecture.
>
> 3. Practical guidance: In S5.1 we report on the robustness of hyperparameters, noting that most hyperparameters did not have a large effect in the simulated setting, aside from a dependence on the distortion threshold $\lambda$. In Section 6, we suggest estimating ''bounds under low and high distortion budgets”. Our supplementaries further include an example notebook on how to use FiGuRO that will be made public on GitHub.
>
> Questions:
>
> 4. Related work: S2P1 discusses advanced uni-modal ID estimation methods. P2 introduces multi-view decomposition methods (lines 95ff) and defines the main gap in existing methods. We then move to briefly touch upon multi-modal representation learning and note that ID estimation has only been discussed for shared information. This is the best structure we can think of to include all information smoothly, but are open for any concrete suggestions. We revise S2P2 to name multi-modal methods explicitly.
>
> 5. $Acc_{all}$: We update the caption for Table 3 to clarify that it refers to the classification accuracy using concatenated features from all learned latent subspaces.
>
> 6. Experimental setup for baselines: Thank you for pointing this out. To improve clarity on implementations, we edit the last sentence in S4.3P2:
> > ''*We implemented these methods with default settings except for sparsity penalty in SLIDE and the rank grid in ShIndICA (tuning required for convergence). Details can be found* in Appendix B.9.”
>
> 7. Ablations and sensitivity analysis: We move the ablation study into the main text. The uni-modal benchmark table contains both the sensitivity and best results. The sensitivity analysis from Supplementary Table 1 was performed on simulation dataset A, and the benchmark on C, so we cannot compare these results directly.
>
> 8. Robustness analysis: Thank you for pointing this out. We will link each experiment in Section 4.2 individually.
>
> 9. Merging single subsections: This was a formatting mistake, thank you for noticing. We have corrected this.

---

> > ### Author Rebuttal · Reviewer_ZRoj · 2026-04-03
> >
> > Thanks for the clarifications. Most of my concerns are addressed, but I am still unsure about the role and impact of the distortion metric.
> >
> > While Appendix Fig. 2 is helpful and supports $R^2$ as a reasonable default on your simulated settings, I do not think it supports the stronger wording in the paper (S6P2) that FiGuRO is robust across distortion metrics. The figure suggests that $R^2$ is the most robust choice among the tested metrics, but other choices of $D$ can show clear spikes in error, and the conclusions are based on artificial simulations. More generally, performance depends on $D$ and on the simulation regime, so the claim should be softened or stated more precisely.
> >
> > FiGuRO also introduces a choice that some reconstruction-based baselines avoid. For example, the "loss cliff" baseline trains multiple autoencoders and uses the reconstruction loss itself as the notion of fidelity to pick the ID. In FiGuRO, the model is trained with a reconstruction loss, but rank updates are driven by a potentially different distortion metric $D$ (e.g., $R^2$). This decoupling is only lightly justified beyond the simulation study, and it raises a question about fairness of comparisons across methods if they are effectively using different notions of fidelity.
> >
> > I suggest softening the robustness claim and adding a short note that the estimated IDs depend on $D$ (and $\lambda$). If possible, a small check of a few $D$ choices on the real benchmark data would also help.
> >
> > Overall, I am leaning toward increasing my score after these clarifications.

---

> > > ### Author Response · Authors · 2026-04-04
> > >
> > > Thank you for this positive response. We are happy we addressed most of your concerns successfully and will gladly give further clarifications and supporting evidence on the distortion metric.
> > >
> > > **Softentening the robustness claim**:
> > >
> > > As per reviewer **aRsj**’s suggestion, we described our results on robustness more clearly and nuanced in S5.1 (please see our initial rebuttal for reviewer aRsj point 6).
> > >
> > > We will also edit S6P2 as you suggested to capture the dependence on the choice of distortion metric:
> > >
> > > > “[…] dependent on the model’s parameters, nonlinearity, *and appropriateness and choice of the distortion metric*. However, unlike prior neural estimators that proved unstable, extensive hyperparameter sweeps confirm FiGuRO’s robustness across diverse data characteristics *and relative stability over* distortion metrics.”
> > >
> > > **Decoupling training loss and distortion metric**:
> > >
> > > We appreciate your concern about the fairness in choosing different distortion metrics from training loss. While we understand how this could raise questions about baseline comparability, this separation is a deliberate design choice to separate training dynamics from the fidelity evaluation and allow for the use of more intuitive metrics. For example, using $R^2$ allows a user to set a threshold (e.g., $\lambda = 0.95$) that universally means "preserve 95% of the variance." This translates intuitively across different domains. We motivate this in our edits along our rebuttal for reviewer **aRsj** point 6.
> > > Importantly, regarding the fairness of comparisons across methods, our empirical evaluations demonstrate that the variation introduced by changing the distortion metric in FiGuRO is substantially smaller than the differences in estimates from different methodologies (such as the loss cliff baseline).
> > >
> > > **Sensitivity to the distortion metric in real data settings**:
> > >
> > > In order to address your recent question about the impact of the distortion metric and your previous question about the sensitivity of hyperparameters “outside the curated benchmark setup”, we set up a small sensitivity analysis on Audio MNIST. In this analysis, we test the effect on the ranks of varying distortion metric, distortion threshold, energy threshold, rank reduction frequency, and patience individually.
> > > The Explained Variance Score gave the lowest estimates, quickly followed by $R^2$ with an average difference of 2.9 in total ranks. RMSE gave the same total average score as $R^2$ and MSE gave estimates of on average 1.1 above $R^2$. This partially aligns with our results on simulated settings. The magnitude of rank changes are within the range of the other two relevant hyperparameters: distortion budget $\lambda$ and energy threshold $\gamma$.
> > > We include these results in an additional supplementary table and add the following text to S5.4P3:
> > >
> > > > *"A small sensitivity analysis to hyperparameters (Supplementary Table XX) revealed that FiGuRO was most sensitive to $\lambda$, the choice of distortion metric, and $\gamma$ (in this order). The sensitivity to $\gamma$ stemmed from an overestimation of ranks due to small $\gamma$ values not allowing for enough rank reduction given $\lambda$. As a result, we believe using $\gamma \geq 0.01$ is most robust. Average total ID estimates were lowest for Explained Variance Score and highest for MSE. The latter matches our previous robustness results on simulations.”*
> > >
> > > We hope these additional clarifications and supporting results fully resolve your remaining concerns and support your inclination to increase your score.

---

### Official Review · Reviewer_6Tr9 · 2026-03-12

**Soundness:** 4
**Presentation:** 4
**Significance:** 2
**Originality:** 3
**Overall Recommendation:** 5
**Confidence:** 4

**Summary:**

The authors stress that one must know the ID of data for interpretable and efficient representation learning. They introduce Fidelity-Rguided Rank Optimization (FiGuRO) to learn the ID of data via truncated singular value decomposition and an algorithm that grows or shrinks dimensionalities in various latent spaces. They claim that shared and private information is disentangled as a result of this process. FiGuRO outperforms existing ID estimation methods.

SOTA ID estimation is accomplished by contrastive models which adapt to shared ID as an emergent property – this is not computed explicitly. FiGuRO directly estimates IDs for each subspace by combining latent rank optimization with a distortion measure.
Rather than estimating the dimension of a latent variable using truncated SVD on latent batches, FiGuRO takes inspiration from LoRA to decompose weight matrices and thereby learn a global low-rank structure which does not depend on individual batches. After training a baseline, high-capacity autoencoder, FiGuRO begins to perform rank reduction according to the distortion budget lambda.
The method is evaluated on simulated data (3 broad datasets with various versions) and real-world data (audio MNIST, So2Sat, and NYU Depth V2). FiGuRO performs relatively well compared to baselines (e.g., AJIVE), especially for private modalities and in high dimensions. The predictions appear somewhat dependent on the distortion budget lambda, with the predictions understandably decreasing with increasing lambda.

**Compliance With Llm Reviewing Policy:**

Affirmed.

**Final Justification:**

The paper is well-written, addresses an important problem, and has findings that are substantiated by sufficient experiments. The authors addressed my concerns sufficiently in the rebuttal. This would be interesting to readers of ICML, and I believe this work should be accepted.

**Key Questions For Authors:**

What is the efficiency of the method compared with the baselines?

How does the method compare in ID estimation with aggregated Local Intrinsic Dimension (LID) estimators? There are strong diffusion baselines for LID estimation

**Limitations:**

No experiments on the efficiency of the method compared with baselines

Work does not discuss or include strong parametric (diffusion) baselines for local intrinsic dimension estimators [1][2]

[1] Kamkari, Hamidreza, et al. "A geometric view of data complexity: Efficient local intrinsic dimension estimation with diffusion models." Advances in Neural Information Processing Systems 37 (2024): 38307-38354.

[2] Yeats, Eric, et al. "A Connection Between Score Matching and Local Intrinsic Dimension." arXiv preprint arXiv:2510.12975 (2025).

**Strengths And Weaknesses:**

Strengths

Strong motivation and clear introduction

Sufficient coverage of related work for ID estimation

Section 3.2, paired with Fig. 1, is well-written and very helpful to understanding the method

Strong reproducibility information and the choices for experiments are well-explained


Weaknesses

It is not clear from the results that FiGuRO is a better choice than existing baselines

No experiments on the efficiency of the method compared with baselines

Work does not discuss or include strong parametric (diffusion) baselines for local intrinsic dimension estimators [1][2]


[1] Kamkari, Hamidreza, et al. "A geometric view of data complexity: Efficient local intrinsic dimension estimation with diffusion models." Advances in Neural Information Processing Systems 37 (2024): 38307-38354.

[2] Yeats, Eric, et al. "A Connection Between Score Matching and Local Intrinsic Dimension." arXiv preprint arXiv:2510.12975 (2025).

---

> ### Author Rebuttal · Authors · 2026-03-30
>
> We thank the reviewer for the detailed summary and are encouraged by their positive assessment of our motivation and methodology. We address questions and concerns regarding baselines, efficiency, and modern local ID estimators below. Revised text is highlighted in *italic*, novel results are in $\textcolor{blue}{\text{blue}}$.
>
> 1. Why FiGuRO is a better choice than existing baselines: In the uni-modal case, some baselines like RRA and lPCA achieve good estimates with optimized hyperparameters, but they are highly sensitive to the choice of their primary hyperparameters. In real-world settings where the ground truth ID is unknown, this sensitivity makes them difficult to use reliably. FiGuRO provides much tighter bounds across its parameter range.
> Our primary contribution however is estimating the complete multi-modal ID structure (shared and private). As demonstrated in Table 2, classical baselines struggle significantly with this task. Our ablation study (Supplementary Table 3) further shows that simply extending reductive methods like RRA to the same multi-modal architecture and training leads to rank collapse, whereas our adaptive algorithm approximates the true manifold structure well.
>
> 2. Computational efficiency: We agree that an efficiency comparison would enrich our work and provide a better understanding of differences in scalability. We have recorded wall times on consistent hardware and report approximate scaling complexities.
>     - We add a wall time column to Supplementary Table 2 and add a new **supplementary table to report wall times** for FiGuRO and multi-modal baseline runs for the simulation data as well as Audio-MNIST. While the traditional linear baselines are faster on our small-scale simulations, the relationship starts to shifts on Audio MNIST. FiGuRO completes in 60.5 minutes, which is ~4x slower than the fastest baseline SLIDE (14.4 minutes), but 8x faster than the slowest baseline ShIndICA (507.83 minutes). Audio MNIST is still a relatively small dataset. To provide a more thorough 'bigger picture' of scalability, we are currently running additional benchmarks on our largest simulation scenario (Dataset C: 30,000 samples, $2 \times 20,000$ features).
>     - Traditional multi-view methods typically rely on SVD of the full data matrices, scaling quadratically with $\max(N,d)$ (N being the number of samples and d being the ambient dimension of the full data). JIVE and ShIndICA additionally scale linearly with the number of iterations. For high-dimensional or very large datasets, this becomes computationally prohibitive. FiGuRO’s complexity scales only linearly with the number of samples N and the data dimension as part of the base model complexity (if assuming the embeddings are derived from an MLP base model, otherwise it scales with the base model complexity and bottleneck size $l$). But it also scales with the number of epochs.
>     - Once the additional runs on data C are in, we will add a sentence in Section 5.4 paragraph 2 summarizing wall times and scaling relationships. We would like to note, however, that even if FiGuRO is not faster than all baselines, it provides latent spaces that can integrate new data at fast inference speeds and a lot of downstream utility and performed much better than baselines at ID estimation and information decoupling.
>
> 3. LID and diffusion baseline: Thank you for pointing out this diffusion-based LID method and bringing recent relevant work to our attention.
>     - We have **integrated FLIPD [1] into our uni-modal benchmark**. As suggested in the paper, we implemented FLIPD in the latent space of the same pretrained encoder (latent dimension 1000) used for FiGuRO, ARR, and ARDVAE for the high-dimensional, uni-modal simulation benchmark in Supplementary Table 2. We increased the range of $t_0$ values to $[0.01, …, 0.7]$ as we observed that the predicted IDs were too large in the recommended range of $[0.01, …, 0.3]$. We add all architecture and training details in the Appendix. The best estimate was achieved at $t_0 = 0.7$ with values $\textcolor{blue}{\text{9.2 ± 0.4, 16.0 ± 0.5}}$. However, the sensitivity to $t_0$ was extremely high, with values from $\textcolor{blue}{\text{9.2 ± 0.4 to 998.1 ± 0.2}}$ for data $C_1$. Including FLIPD enriches our analysis, but since it is a local, uni-modal estimator, it does not change our core conclusions regarding global multi-modal ID estimation.
>     - We added this reference and the recent work by Yeats et al. (2025) to our related works (Section 2 paragraph 1 after the last sentence):
>     > ''*Another recent direction in neural ID estimation connects manifold dimension and score-based generative models. Kamkari et al. (2024) introduced FLIPD for efficient local ID estimation. Yeats et al. (2025) took this further and established a formal theoretical link, showing that the denoising score matching loss provides a lower bound for a manifold’s ID.*"

---

> > ### Author Rebuttal · Reviewer_6Tr9 · 2026-04-02
> >
> > Thank you for addressing my questions (1 & 2 in particular). The author's response to 2 is comprehensive and paints a more complete picture on the computational performance of the proposed method.
> >
> > For the result with 3, it appears that the diffusion model failed to fit the latent manifold (LID predictions that are near ambient dimension). This is to be expected if the diffusion model is implemented as an MLP. A diffusion transformer (DiT) architecture is necessary in this setting. Please make note of this - one can consider it a limitation of the diffusion-based LID estimators.
> >
> > I am leaning towards an increase to 5 (Accept). Thanks again for the response.

---

> > > ### Author Response · Authors · 2026-04-04
> > >
> > > Thank you very much for this positive response and the score increase. We are happy we addressed questions 1 and 2 successfully.
> > >
> > > We appreciate you highlighting the findings in Yeats et al. (2025) regarding the necessity of Diffusion Transformer (DiT) backbones for complex distributions. To clarify our previous results: We report the full range of dimension estimates for all tested $t_0$ values for transparency, since (as mentioned by Kamkari et al. (2024)) the automatic knee detection may fail for complex datasets. As per their suggestion, We then treated $t_0$ as a tunable hyperparameter and picked $t_0$ at which estimates were closest to the GT in Supplementary Table 2 as the “best” ($t_0$=0.7). We also ran the knee algorithm, which predicted an ID of 130.53 ± 0.58 for data $C_1$ (consistently best $t_0$=0.4 when tested across 50 values from 0.01 to 1.0). We have added both the hand-picked and the knee LID estimates to Supplementary Table 2. However, this value is still a lot higher than estimates from some other NN-based estimators and may be a result of the limitation mentioned by you.
> > >
> > > We add the following to our implementation description of FLIPD:
> > >
> > > > “We note that our decision to maintain a uniform MLP architecture to ensure a fair algorithmic comparison highlights a fundamental limitation of diffusion-based LID estimators: their high sensitivity to the backbone capacity. As recently demonstrated by Yeats et al. (2025), standard MLPs can underfit highly complex manifolds. This biases estimates toward higher dimensions. More expressive architectures such as Diffusion Transformers, are often necessary to reliably fit the distribution and extract the local intrinsic dimension.”
> > >
> > > And we add this to the results in Supplementary Table 2:
> > >
> > > > “Note: The overestimation observed for FLIPD illustrates their sensitivity to architectural capacity, a limitation of diffusion-based LID estimators. Standard MLPs can underfit complex manifolds and bias estimates toward the ambient dimension (Yeats et al., 2025).”

---

### Official Review · Reviewer_XKvv · 2026-03-13

**Soundness:** 3
**Presentation:** 3
**Significance:** 3
**Originality:** 3
**Overall Recommendation:** 4
**Confidence:** 3

**Summary:**

The paper studies intrinsic dimension estimation for multimodal data, aiming to explicitly estimate the dimensionalities of a shared latent subspace and modality-specific private subspaces. It proposes FiGuRO, a reconstruction-fidelity-guided dynamic rank optimization approach that adaptively selects the ranks of shared and private components. The paper further claims that shared/private disentanglement emerges naturally from this rank optimization process. The method is evaluated on synthetic and real datasets, as well as on fixed pretrained representation spaces.

**Compliance With Llm Reviewing Policy:**

Affirmed.

**Key Questions For Authors:**

1)Is the quantity estimated by FiGuRO intended to represent the intrinsic dimension of the data itself, or rather an effective dimension under a given model and distortion budget?
2)Could the authors provide further analysis of the sources of the key deviation cases in Table 2?
3)Given that real-world datasets do not have ground-truth intrinsic dimensions, are the real-data experiments primarily intended to support the practical utility of the method, or the correctness of its dimension estimates?

**Limitations:**

Yes

**Strengths And Weaknesses:**

Strengths
1）Tackles an important problem: explicitly estimating shared vs. modality-private intrinsic dimensions in multimodal data.
2）Solid evaluation mix: controlled synthetic setups plus real-world experiments.

Weaknesses
1）Rank ≠ intrinsic dimension (as stated).  The abstract treats FiGuRO’s learned rank as “intrinsic dimension,” but later sections admit it depends on model capacity, distortion metric, and budget. The paper should clearly frame this as an effective dimension under specified conditions, not an absolute property of the data.
2）Bias cases under-explained. Table 2 shows notable deviations (e.g., Bi1 shared ID 13.4 vs. GT 20). Underestimation is mentioned but not analyzed; the source (optimization, identifiability, metric choice, budget, model mismatch) needs clearer diagnosis.
3）Real-data results validate utility more than correctness. Without ground-truth IDs, the real-data experiments cannot verify that the estimated numbers are correct; “reasonable range” vs. prior unimodal estimates is weak evidence. This limitation should be stated more explicitly.

---

> ### Author Rebuttal · Authors · 2026-03-30
>
> We very much appreciate your positive feedback on our work, especially that we tackle an important problem and provide solid evaluation on synthetic and real data. Your comments on framing and diagnostic clarity are important, and we believe our answers and planned revisions will improve the paper’s positioning. Revised text is highlighted in *italic*.
>
> 1. You are correct, FiGuRO computes an effective dimension dependent on data, model capacity, and hyperparameters as global ID estimates. We have already acknowledged this dependency in our assumptions (Section 3.2 lines 171ff) and conclusion (Section 6 lines 436ff). But we agree it should be made clear from the start. We edit the following sentences:
> > ''[...], a framework for *approximating* the complete ID structure of uni- and multi-modal data *under constraints of model capacity and hyperparameters.*" (abstract)
>
>     > ''Our approach estimates the *effective* IDs for each subspace *under a number of constraints. We have combined* latent rank optimization [...]." (Section 1)
> 2. Diagnosis of underestimation in Table 2: Thank you for highlighting that we did not discuss this observation. The deviations in $B_{i1}$ shared and $B_{l}$ private subspace 1 could stem from the following 3 sources we identified:
>     - Generative redundancy: We looked into our data simulators and found that the ground truth variables have - in some cases - redundancies/correlations. This could explain the case of $B_{i1}$ shared.
>     - Subspace leakage (Identifiability): Predictability analysis shows that shared labels sometimes bleed into modality-specific subspaces and vice versa (see Supplementary Figures 5-6). This could explain the case of $B_{i1}$ shared.
>     - Metric bias: In our evaluation of distortion metrics, $R^2$ was selected for its stability. However, because rank is lower-bounded by 1, underestimation of the GT 5 is 'penalized' less in terms of optimization range than overestimation (which is upper-bounded by the initial rank of 25).
>     - Limitation: This underestimation could in part demonstrate that while disentanglement is a strong emergent property of FiGuRO, it is not a hard mathematical guarantee.
>     - We make the following edits to the main text:
>     > ''[...] albeit sometimes underestimating the true ID ([...]). *The $B_{i1}$ shared generative variables exhibited an effective rank of 6.61 in contrast to the 20 variables (Supplementary Table XX), and we also see some information leakage into the private subspaces for this case (Supplementary Figure 6).*" (Section 5.2)
>
>         > ''[...] *It is not guaranteed, as we saw in some failure modes in which dimensions were underestimated likely due to generative variable correlation, information leakage, or a conservative distortion metric. However, a high degree of* disentanglement occurs [...]." (Conclusion)
> 3. Real data experiments: As correctly noted, correctness cannot be verified without ground truth, which we do not have in real data. These experiments are primarily intended to demonstrate practical utility, specifically FiGuRO’s ability to serve as a scalable latent probe for large models and provide interpretable, compressed representations. Section 4.4 lines 255ff state that we evaluate the meaningfulness of the ID estimates and information decoupling. We see how some word choices could be confusing. We will rename Section 4.4 to ''*Application to* Real Data" and clarify the objective in the contribution (Section 6 paragraph 2 sentence 1: ''Our findings are further *supported by experiments* on real-world datasets [...]”. Section 5.4 paragraph 3 focuses on MNIST. Here, we have a lot of literature making estimates on the ID. So we wanted to use the 'reasonable range' of 7-25 as a sanity check for consensus. We hope this clarifies the purpose of the real world data experiments.

---

### Official Review · Reviewer_aRsj · 2026-03-15

**Soundness:** 2
**Presentation:** 2
**Significance:** 2
**Originality:** 2
**Overall Recommendation:** 3
**Confidence:** 4

**Summary:**

The paper aims at defining an adaptive method to determine and exploits the optimal Intrinsic Dimension (global) of the data in question via shaping the latent space in Auto Encoders.
The methods is heavily based on the idea presented in the ARR (cited) and introduces Rate Distortion Theory as a base for the adaptivity of the method.

**Compliance With Llm Reviewing Policy:**

Affirmed.

**Key Questions For Authors:**

* d is the main driver for the adaptation and as such is supposed to captured the complexity of the data. How can you justify your choices of such metrics?
* Rank reduction via t-SVD essentially relies on the projection over the fixed rank matrix manifold (constructed via rank decomposition) with approximation loss provided by the Eckart-Young theorem (1936). Why is this not mentioned as a formal statement rather than (poentially confusing and unclear) text?

**Limitations:**

Yes

**Strengths And Weaknesses:**

Strengths:
* The paper addresses an important fundamental problem in Machine Learning, that of discovering and exploiting the Intrinsic dimensionality (ID) of the data (rather information - or representation of-) of the problem in question.
* The method seeks a scalable technique that will be actionable at inference time

Weaknesses:
* The paper is heavily based of the ARR method that essentially proposes a *linear* rank reduction of the latent space (via t-SVD) as a (weak) way to discover the intrinsic dimensionality. The added contribution is essentially that of using Rate Distortion Theory (RDT) for the adaptive process, which is a limited empirical contribution.
*  The paper neglects all the line of research on ID estimation (outside deep learning). Worse, the paper assumes a *global* ID whereas a line of work seeks to estimate the *local* complexity of the data as local ID.
* The paper is unclear on several places:
  - The (adhoc) patience parameter is weakly defined in the supplementary material only
  - Algo 1 is not aligned with the text. In line 21 it is question of a cumulative value larger than $\gamma$, in the text it says "discarding dimensions with an energy [not cumulative] below $\gamma$". Also the equation for $E_j$ has no meaning.
  - The proof of convergence (A.1.6) is not convincing on the fact that the algorithm cannot oscillate. The patience parameter only prevents the algorithm to get stuck over a constant value.
  - The distortion metric should be discussed more clearly in the main text (instead of merely presented in B.4.1) as it is the main driver for the adaptation

---

> ### Author Rebuttal · Authors · 2026-03-30
>
> We thank the reviewer for their rigorous evaluation and technical insights, which have helped us strengthen the mathematical grounding of our work. We clarify below how FiGuRO diverges from prior work and detail the revisions implemented to address specific concerns. Revised text is highlighted in *italic*.
>
> Clarifications:
>
> 1. Novelty and relationship to ARR: While ARR provides inspiration for adaptive latent rank reduction, our work addresses the fundamental challenge of **multi-modal subspace ID estimation and decomposition**.
>     - Our primary contribution is quantifying and disentangling shared and modality-specific IDs. To our knowledge, FiGuRO is the first neural framework to provide these estimates.
>     - To achieve stable decomposition in the multi-modal setting, we introduced critical architectural changes that move beyond the original ARR logic: (1) weight-matrix decomposition for a batch-independent global structure; (2) bidirectional optimization allowing ranks to grow or shrink to satisfy distortion budgets (essential for multi-modal convergence); and (3) cumulative energy thresholds for scalable rank adjustment.
>     - Our ablation (Supplementary Table 3) shows ARR-style linear reduction universally collapses in multi-modal architectures, proving our fidelity-guided engine is essential.
> 2. Clarification on related works (ID estimation): We respectfully disagree with the reviewer’s assessment that we neglected non-deep learning ID estimation or ignored the distinction between global and local ID (LID). We have engaged with both lines of research throughout the paper.
>     - We compared FiGuRO against **ten traditional estimators including both global and local methods** (see Section 1 paragraph 2 lines 33ff, Supplementary Table 2, Appendix B.9.1).
>     - All multi-modal baselines are non-deep learning (Table 2, Section 2 paragraph 2 lines 95ff).
>     - While LID is valuable for point-wise complexity, our research objective is to learn global IDs and general decomposed subspaces from multi-modal data.
>
> We thank you for your insightful feedback on what is unclear and could be communicated better:
>
> 3. Patience: $\pi$ is designed to provide the model sufficient time to stabilize and optimize its weights following a change in latent dimensionality (Algorithm 1 line 26, Appendix C.1). We add to the end of Section 3.2 paragraph 1:
> > ''*An additional hyperparameter of FiGuRO is patience $\pi$. $\pi$ determines after how many steps $\tau$ without rank changes the algorithm stops.* The full procedure is detailed in Algorithm 1."
> 4. We appreciate pointing out the misalignment between text and Algorithm 1. We exchange the sentence in question (last sentence in Section 3.1 paragraph 3, line 146ff):
> > ''*While Mounayer et al. (2025) used a unit step of 1 to reduce the ranks, a cumulative energy threshold $\gamma$ of the singular values S is a more flexible choice (Zhang et al., 2023): Dimensions above index $\min (k: \sum_j^k \mathbf{E}_j \geq 1 - \gamma)$ with the energy [...] are discarded all at once.*"
> 5. We appreciate the reviewer’s rigorous check of our theoretical guarantees and thank them for pointing out a weakness in Theorem A.1.6. We revise the proof in Theorem A.1.6:
> > ''*Together with the discrete rank updates and a monotonically tightening bound $[k_{min}, k_{max}]$, continuous weight optimization ensures that reconstruction fidelity for any fixed rank is non-decreasing over training time. This contractive dynamic ensures the model cannot cycle indefinitely between failure states and sets for the minimum stable rank.* The patience parameter $\pi$ […].”
> 6. Distortion metric: We acknowledge that the justification for our distortion metric fell short in the main text. We make the following edits to clarify our choice of $R^2$:
> > Section 3.2: ''We mainly use the coefficient of determination [...] as our fidelity metric. *R2 is scale-invariant,* which allows us to define the minimum acceptable fidelity as [...]. *This makes the use of FiGuRO with different data and loss functions more intuitive.*"
>
>     > We also move our results from C.1 to the end of Section 5.1: ''*Investigating the robustness of different distortion metrics to nonlinearities and sparsity in the data revealed that $R^2$ was more robust to sparsity than MSE and RMSE, and more robust to nonlinearities than the Explained Variance Score.*"
>
> To answer your remaining questions:
>
> 7. Choice of D: Please see our earlier response in point 6.
> 8. We appreciate your suggestion to define our rank-reduction mechanism more precisely. We edit Section 3.1 accordingly:
> > ''Building on truncated Singular Value Decomposition (SVD), *the unique optimal rank-$k^\*$ approximation of $Z$ is given by* $Z \approx Z^{(k^\*)} = U^{(k^\*)}S^{(k^\*)}V^{T(k^\*)}$*, as established by the Eckart-Young-Mirsky Theorem (Eckart & Young, 1936; Mirsky, 1960).*"
>
> We hope that our clarifications and revisions convince you of the relevance and soundness of our work.

---

> > ### Author Rebuttal · Reviewer_aRsj · 2026-04-04
> >
> > I thank the authors for the answers to my comments, the adaptation they propose improve the clarity f the man text.
> > However, and also supported by other reviews, the paper remains a LINEAR dimension (aka rank) estimation technique and does not provide any certainty that the value thus estimated can be referred to as Intrinsic Dimension, which by definition is Intrinsic to the data and is not a mere estimate of the number of dimensions required to represent it. This is only true for FLAT spaces.
> > This deep discrepancy (or lack of guarantee) remains problematic in my view. At the light of the discussion and the proposed adaptation of the paper, the title of the proposal is admittedly too bold of a statement.
> > Despite the value of the content, this creates a confusion between what the paper is meant to contain and its actual content.

---

> > > ### Author Response · Authors · 2026-04-05
> > >
> > > We thank the reviewer for acknowledging that we successfully addressed concerns from the original review, clarified misunderstandings about the novelty of the method and discussion of related work, and that our revisions improved the clarity of the main text.
> > >
> > > The reviewer’s new concern seems to be based on reviewer XKvv’s review concern 1, that the rank does not directly equate to ID. In our response to reviewer XKvv, we have addressed this and ensured that our paper clearly states that we are approximating the ID by estimating effective dimensions under constraints of model capacity already in the abstract (and in introduction and conclusion) to minimize ambiguity. Please see our rebuttal to reviewer XKvv point 1 for the exact edits.
> > >
> > > Additionally, the reviewer suggests that because our rank estimation is linear, it only applies to flat spaces. We respectfully point out that this overlooks the role of the non-linear encoder and projection layers in FiGuRO’s architecture. While the rank reduction at the bottleneck itself is linear, the mapping from the ambient data space (or precomputed embeddings) to the latent space is not. According to the manifold hypothesis, a sufficiently expressive non-linear encoder learns to "unroll" or flatten a curved data manifold into a Euclidean latent space [1,2]. Therefore, estimating the linear rank of this learned latent space is a valid mechanism for estimating the intrinsic dimension of the original, non-linear manifold [3]. We also provided empirical proof of this in our simulations, where FiGuRO successfully recovered the ID of data generated with complex non-linearities, a task where strictly linear methods fail.
> > >
> > > We hope that given our detailed clarifications and text improvements, the reviewer will reconsider their score.
> > >
> > > References:
> > >
> > > [1] Basri, N. & Jacobs, D.W. (2017). Efficient Representation of Low-Dimensional Manifolds using Deep Networks. International Conference on Learning Representations.
> > >
> > > [2] Psenka, M. et al. (2024). Representation Learning via Manifold Flattening and Reconstruction. Journal of Machine Learning Research.
> > >
> > > [3] Jing, L., Zbontar, J., LeCun, Y. (2020). Implicit Rank-Minimizing Autoencoder. Advances in Neural Information Processing Systems.

---

### Decision · Program_Chairs · 2026-04-30

**Decision:**

Accept (regular)

**Comment:**

The reviewers appreciated the motivation and significance of the problem addressed in the paper and highlighted the strong evaluation across synthetic and real-world multi-modal experiments, comparing to numerous baselines. The authors demonstrate the utility of the approach in downstream tasks on real-world data. The method is also able to estimate effective dimensionality in unimodal settings. The design by which disentanglement of shared and private information arising as an emergent property is an interesting contribution of interest to the community.

Concerns about the method being true intrinsic dimension estimation of the data vs effective dimension have been addressed by the authors with a promise to emphasize in the text that FiGuRO computes an *effective dimension* dependent on data, model capacity, and hyperparameters as global ID estimates. This and other revisions for clarity that arose in the discussion (e.g., diagnosis of underestimation, role and impact of $R^2$, ) should be incorporated in the final manuscript. Concerns regarding efficiency of the method compared with baselines were also resolved, and results should be added to the paper. As a minor comment bold/underline should be added for all rows of table 3.